# Starvation-induced proteasome assemblies in the nucleus link amino acid supply to apoptosis

Maxime Uriarte [1,2], Nadine Sen Nkwe [2,3], Roch Tremblay [2,3], Oumaima Ahmed [2,3], Clémence Messmer [1,2], Nazar Mashtalir [4,5], Haithem Barbour [2,6], Louis Masclef [1,2], Marion Voide [2,3], Claire Viallard [2,7], Salima Daou [8], Djaileb Abdelhadi [1,2], Daryl Ronato [9,10], Mohammadjavad Paydar [11], Anaïs Darracq [2,3], Karine Boulay [2,7], Nicolas Desjardins-Lecavalier [2], Przemyslaw Sapieha [2,12,13], Jean-Yves Masson [9,10], Mikhail Sergeev [2,7], Benjamin H. Kwok [7,11], Laura Hulea [2,7], Frédérick A. Mallette [2,7], Eric Milot [2,7], Bruno Larrivée [2,7], Hugo Wurtele [2,7] & El Bachir Affar [2,7✉]

Eukaryotic cells have evolved highly orchestrated protein catabolic machineries responsible for the timely and selective disposal of proteins and organelles, thereby ensuring amino acid recycling. However, how protein degradation is coordinated with amino acid supply and protein synthesis has remained largely elusive. Here we show that the mammalian proteasome undergoes liquid-liquid phase separation in the nucleus upon amino acid deprivation. We termed these proteasome condensates SIPAN (Starvation-Induced Proteasome Assemblies in the Nucleus) and show that these are a common response of mammalian cells to amino acid deprivation. SIPAN undergo fusion events, rapidly exchange proteasome particles with the surrounding milieu and quickly dissolve following amino acid replenishment. We further show that: (i) SIPAN contain K48-conjugated ubiquitin, (ii) proteasome inhibition accelerates SIPAN formation, (iii) deubiquitinase inhibition prevents SIPAN resolution and (iv) RAD23B proteasome shuttling factor is required for SIPAN formation. Finally, SIPAN formation is associated with decreased cell survival and p53-mediated apoptosis, which might contribute to tissue fitness in diverse pathophysiological conditions.

[1] Department of Biochemistry and Molecular Medicine, University of Montréal, H3C 3J7 Montreal, QC, Canada. [2] Maisonneuve-Rosemont Hospital Research Center, Montréal, QC H1T 2M4, Canada. [3] Molecular Biology Programs, University of Montreal, Montréal H3A 0G4 QC, Canada. [4] Department of Pediatric Oncology, Dana-Farber Cancer Institute and Harvard Medical School, Boston, MA 02215, USA. [5] Broad Institute of MIT and Harvard, Cambridge, MA 02142, USA. [6] Biomedical Sciences Programs, University of Montréal, Montréal H3C 3T5 QC, Canada. [7] Department of Medicine, University of Montréal, Montréal H3C 3J7 QC, Canada. [8] Lunenfeld-Tanenbaum Research Institute, Sinai Health System, Toronto, ON M5G 1X5, Canada. [9] CHU de Québec Research Center, Oncology Division, 9 McMahon, Québec City, QC G1R 3S3, Canada. [10] Laval University Cancer Research Center, Québec City, QC G1V 0A6, Canada. [11] Institute for Research in Immunology and Cancer (IRIC), University of Montréal, Montréal, QC H3T 1J4, Canada. [12] Department of Ophthalmology, University of Montréal, Montréal, QC, Canada. [13] Department of Neurology-Neurosurgery, McGill University, Montréal, QC, Canada. ✉email: el.bachir.affar@umontreal.ca

Protein degradation and subsequent recycling of amino acids is fundamental for normal cell physiology. The autophagy system targets large macromolecule complexes, protein aggregates and organelles, all of which are first engulfed within double membrane-delimited structures and subsequently delivered to the lysosome for degradation[1-3]. On the other hand, the proteasome catalyzes the degradation of proteins that are in surplus, improperly folded or unwanted at a given time or in a specific subcellular location[4-6].

The proteasome is an evolutionarily conserved protein degradation machinery that generally recognizes substrates that are polyubiquitinated through the concerted action of E2 ubiquitin-conjugating enzymes and E3 ubiquitin ligases. The 26S proteasome is composed of two sub-complexes, the 20S cylinder-like catalytic particle (CP) and the 19S regulatory particle (RP) (Fig. 1a). The CP contains the proteases with CASPASE-like, trypsin-like and chymotrypsin-like activities that are responsible for substrate degradation into small peptides[7,8]. This particle is the target of the widely used proteasome inhibitors (e.g., MG132 and Bortezomib)[7,8]. The RP is also a large multi-protein complex that binds the CP to assemble a competent proteasome. The RP is responsible for the recognition and unfolding of polyubiquitinated proteins, as well as their translocation inside the CP. The CP can also associate with other regulatory complexes including the homoheptameric ring-shaped 11S complex composed of PSME3 (PA28γ or REGγ), which targets proteins for ubiquitin-independent degradation (Fig. 1a)[9].

While a large body of findings have elucidated how the proteasome regulates diverse cellular processes, including the identification of a wide spectrum of its substrates, much less is known about how the proteasome is regulated. Nonetheless, ample evidence now indicated that highly conserved mechanisms, e.g., transcriptional feedback mechanisms and post-translational modifications, regulate the abundance of proteasome subunits and their assembly into a degradation competent complex (recently reviewed by Rousseau et al.[6]).

Several studies have also enquired to determine how protein degradation by the proteasome is coordinated with protein synthesis and metabolic demands. It was originally found that, following amino acid deprivation in mammalian cells, proteasome activity is required to replenish the cellular amino acid pool, thus ensuring protein synthesis[10]. Indeed, proteasome inhibition in yeasts and mammalian cells results in amino acid depletion, which triggers cell death, indicating the critical role of the proteasome in maintaining amino acid homeostasis[11]. On the other hand, ostensibly conflicting results exist on the link between the mechanistic target of rapamycin (mTOR) kinase, which promotes anabolism and protein synthesis, and the signaling pathways that coordinate proteasome function. TOR signaling appears to play an important role in negatively regulating proteasome abundance in yeast and mammals[12,13]. Inhibition of TORC1 induces a transient increase in the abundance of proteasome subunits and their assembly[12]. Interestingly, the activation of proteasomal degradation upon mTOR inhibition is accompanied by a rapid increase of K48-linked ubiquitination and the degradation of long-lived proteins[13,14]. However, overactivated mTOR signaling was also shown to increase proteasome-mediated proteolysis[15]. While the reasons behind these discrepancies remain unclear, these studies provided evidence for the intricate relationships between cell growth signaling and protein degradation.

Another potentially important determinant in amino acid homeostasis is how the subcellular localization of the proteasome is modulated under conditions of nutrient deprivation. In the budding yeast, *Saccharomyces cerevisiae*, the 26S proteasome is relocated from the nucleus to the cytoplasm during quiescence or carbon starvation[16]. The cytoplasmic proteasome assembles a large condensate termed Proteasome Storage Granule (PSG), and

these structures are rapidly dissipated when cells resume proliferation[16]. A genetic screen in yeast revealed that several signaling pathways directly or indirectly contribute to the assembly of PSG[17]. Interestingly, while nitrogen starvation induces proteasome degradation by autophagy, a process termed proteaphagy, glucose starvation induces the formation of cytoplasmic PSG. It was concluded that PSG protects the proteasome from autophagy, thus promoting yeast cell viability following periods of carbon starvation[18]. Remarkably, carbon deprivation also induces the formation of large cytoplasmic PSG-like structures in *Arabidopsis Thaliana*[18]. In contrast, nitrogen starvation in this organism induces proteaphagy, a process that requires a multivalent binding of RPN10 subunit of the proteasome with ubiquitin and the ubiquitin-like protein, Autophagy-related protein 8 (ATG8)[19]. These findings further highlight the evolutionary conservation and functional importance of proteasome delocalization and regulation mechanisms.

In animal species, including mammals, the mechanisms that coordinate the subcellular localization of the proteasome in response to nutrient deprivation are incompletely understood. The cytoplasmic proteasome was shown to be partly degraded by the autophagy machinery following nutrient starvation[20]. Indeed, amino acid starvation induces proteasome recognition by the ubiquitin-associated domain of p62/SQSTM1 and subsequent engulfment by the autophagy system in the cytoplasm[20]. However, it remained unclear how the nuclear proteasome is regulated in response to nutrient deprivation. In this study, we show that the mammalian proteasome undergoes liquid–liquid phase separation (LLPS) in the nucleus in response to amino acid starvation. We present a characterization of these proteasome condensates, which we termed SIPAN (starvation-induced proteasome assemblies in the nucleus). We show that these structures are associated with amino acid deprivation-mediated cell death.

## Results

**Nutrient deprivation in mammalian cells results in the localization of proteasome components into nuclear foci.** In yeast, the 26S proteasome particle translocate to the cytoplasm and form proteasome storage granules (PSG), in response to carbon starvation or quiescence entry[16,18], but whether the mammalian proteasome is also subjected to a similar regulation has remained unknown. We used primary human fetal lung fibroblasts IMR90 as a model for normal diploid mammalian cells which were incubated in Hank's Balanced Salt Solution (HBSS), containing 1 g/L of glucose as the only nutrient (HBSS hereafter). While nutrient-deprived IMR90 cells showed a rapid loss of 4EBP1 phosphorylation[21], which is indicative of cell starvation, no major changes were observed on the abundance of proteasome subunits or accessory factors (Fig. 1a, b). However, a noticeable decrease could be observed, at later time points, for certain proteasomal factors, e.g., PSMB7 and PSMD14, possibly reflecting autophagy consumption of the cytoplasmic proteasome, as previously reported[20]. In addition, no signs of cell death were detected after 12 hrs of treatment, as indicated by the normal appearance of cells (Supplementary Fig. 1a), absence of subG1 apoptotic cell population (Supplementary Fig. 1b), and absence of apoptotic cleavage of PARP1 and CASPASE-3 (Supplementary Fig. 1c). Similar cellular response to nutrient deprivation was observed in HCT116 colon carcinoma cells with no apparent apoptotic cleavage of CASPASE-3 or PARP1 (Supplementary Fig. 1d). Using these two cell models, we initially conducted immunostaining at 6 or 8 hrs post-nutrient starvation and revealed that PSMD4 subunit of the RP localizes in spherical nuclear foci in IMR90 and HCT116, respectively (Fig. 1c). Importantly, hyperosmotic stress was recently shown to induce proteasome foci

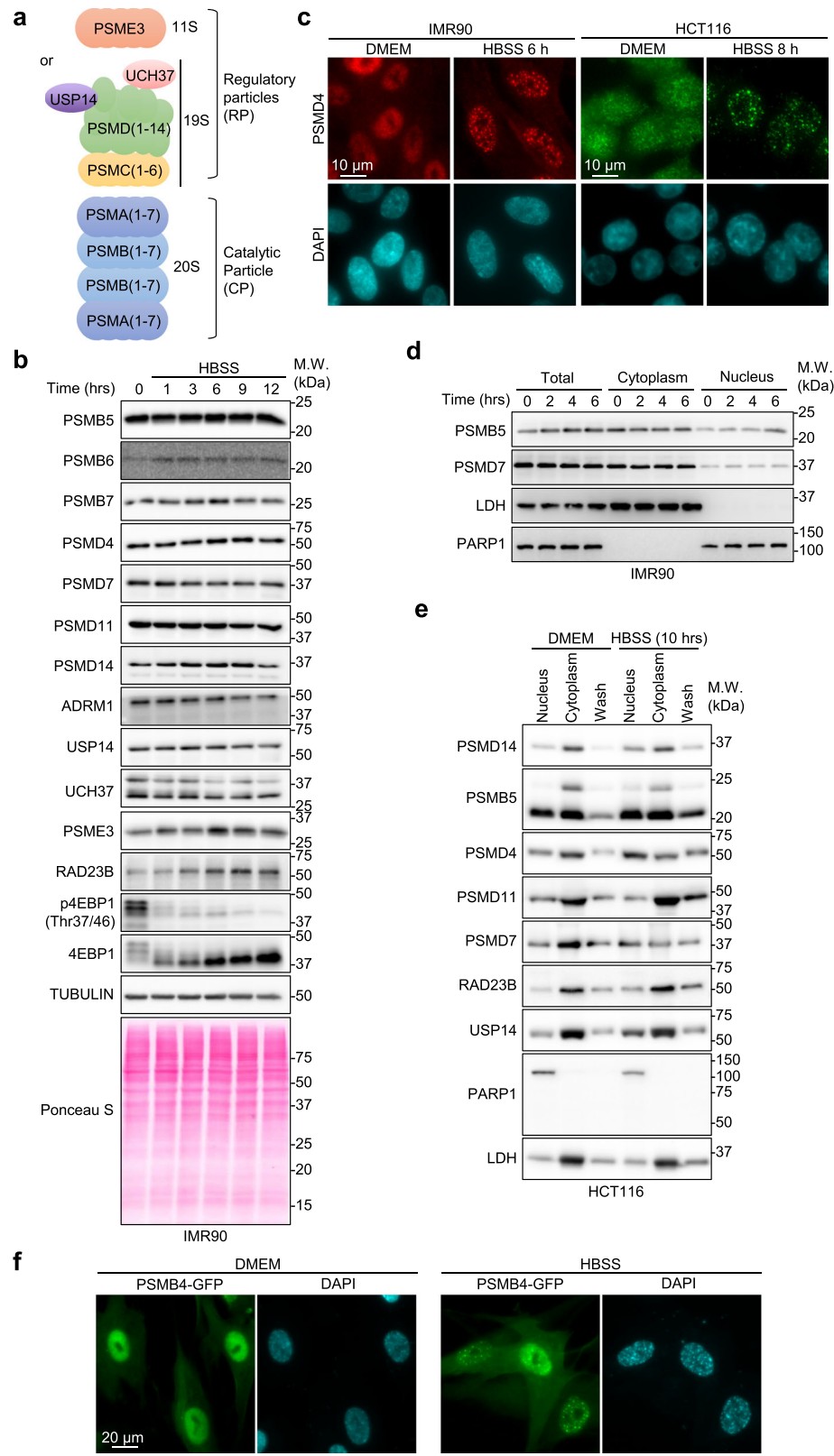

formation in the nucleus[22]. However, HBSS osmolality was around 280 mOsm/kg, corresponding to normal values of cell culture media. In addition, no significant changes of HBSS osmolality was observed under conditions of HBSS treatment for both cell types used (Supplementary Fig. 1e, f).

Next, we isolated cytoplasmic and nuclear fractions of IMR90 and HCT116 cells, following incubation in HBSS, and found that,

in contrast to yeast[16,18], no manifest translocation of proteasome components from the nucleus to the cytoplasm was noticed (Fig. 1d, e). Similar observations were also made in IMR90 cells stably expressing PSMB4-GFP, which showed foci formation following HBSS with no manifest change in its nucleocytoplasmic distribution (Fig. 1f). Thus, in mammalian cells, proteasome components are localized into nuclear foci in response to nutrient

**Fig. 1 Nutrient starvation induces the formation of proteasome foci in the nucleus of mammalian cells. a** Schematic representation of the mammalian proteasome composed by the 20S catalytic particle (CP) and 19S or 11S regulatory particles (RP). **b** Protein levels of proteasome components and other factors following nutrient deprivation in IMR90 cells. Cells were incubated in HBSS solution and harvested at different time points for immunoblotting. 4EBP1 phosphorylation is included as a control for starvation (representative from 2 independent experiments). **c** IMR90 and HCT116 cells form nuclear foci following nutrient deprivation. IMR90 and HCT116 cells were incubated in HBSS for 6 hrs or 8 hrs, respectively, and harvested for immunostaining (representative from 3 independent experiments). **d** Fractionation of IMR90 and immunoblotting for components of the proteasome. IMR90 cells were fractionated by hypotonic lysis and cytoplasmic and nuclear fractions were used for immunoblotting. LDH and PARP1 proteins were detected as markers of cytoplasmic and nuclear fractions, respectively (representative from 2 independent experiments). **e** Fractionation of HCT116 cells and immunoblotting for components of the proteasome. HCT116 was fractionated by quick lysis with 0.1 % NP-40 detergent and nuclear and cytoplasmic fractions were obtained. The wash fraction corresponds to resuspension of the nuclear pellet in detergent-free buffer followed by an additional centrifugation. LDH and PARP1 were detected as markers of cytoplasmic and nuclear fractions, respectively (representative from 2 independent experiments). **f** Subcellular localization of PSMB4-GFP following incubation of IMR90 cells in HBSS. IMR90 cells were transduced with lentivirus particles expressing PSMB4-GFP. Following four days post-infection, cells were incubated in HBSS for 6 hrs and harvested for fluorescence microscopy (representative from 3 independent experiments).

deprivation. Moreover, the nuclear and cytoplasmic proteasomes appear to be subjected to distinct regulatory events.

**Interdependency of catalytic and regulatory particles in proteasome nuclear foci formation upon nutrient starvation**. We first probed multiple components of the proteasome and found that several subunits of the CP, e.g., PSMB1, PSMB2, PSMB4, PSMB6, and PSMB7, co-localize with PSMD11 or PSMD4, components of the RP, following incubation of IMR90 in HBSS (Fig. 2a, Supplementary Fig. 2a). C-terminal GFP-tagged versions of PSMB4, PSMB5, PSMD12, or PSMD14, all localize in nuclear foci, following incubation of IMR90 cells in HBSS (Fig.1f and Supplementary Fig. 2b). These results suggest that proteasome nuclear foci formed in response to nutrient deprivation contain fully assembled 26S proteasome particles.

Next, we sought to test whether CP and RP are independently assembled in proteasome foci upon nutrient starvation. Following siRNA-mediated depletion of PSMB5, PSMB6 or PSMB7 components of the CP, we observed reduced accumulation of the RP components, PSMD4, PSMD7, or the CP component PSMB4, in nuclear foci (Fig. 2b, c). Conversely, depletion of PSMD4, PSMD7, PSMD11, or PSMD14 components of RP results in reduced assembly of PSMB4 in nuclear foci. Similar results were obtained following HBSS treatment of IMR90 cells stably expressing PSMB4-GFP (Supplementary Fig. 2c). Efficient depletion of proteasomal proteins was validated by immunoblotting (Supplementary Fig. 2d). Interestingly, PSME3 proteasome activator complex also localizes in nuclear foci with PSMD7 following incubation of IMR90 cells in HBSS (Fig. 2d). Additionally, we found that while depletion of components of the CP or the 19S RP results in the loss of PSME3 foci (Fig. 2e), no significant effect was observed on either PSMD4 or PSMB4 foci following PSME3 depletion (Fig. 2f, Supplementary Fig. 2e). Thus, following nutrient deprivation, both 20S CP and 19S RP are required for the assembly of intact proteasome particles in nuclear foci.

**Starvation-induced proteasome assemblies in the nucleus (SIPAN) are a general response of mammalian cells to nutrient deprivation**. Nuclear proteasome foci induced by nutrient deprivation, as detected with PSMD14 or PSMD4 antibodies, do not correspond to known nuclear foci, condensates or bodies such as PML bodies (PML staining), nucleoli (Fibrillarin staining), nuclear speckles (SC35 staining) or DNA double strand break foci (53BP1 staining) (Fig. 3a). Confocal microscopy indicated that PSMB4-GFP foci are preferentially located in low-density chromatin regions (Fig. 3b). This was confirmed by transmission electronic microscopy, which also revealed that proteasome foci are membrane-less nuclear structures (Fig. 3c).

It was recently shown that, under hyperosmotic stress conditions, proteasome undergoes LLPS and form nuclear foci that target ribosomal proteins[22]. However, under conditions of nutrient deprivation, the stress-sensitive nucleolar protein NPM1 is not re-localized to the nucleoplasm (Supplementary Fig. 3a). Moreover, neither endogenous nor overexpressed ribosomal proteins are enriched in nutrient deprivation-associated proteasome foci (Fig. 3d, Supplementary Fig 3b–d). Thus, in contrast to hyperosmotic stress-induced proteasome foci[22], nucleolar stress is not associated with proteasome foci formation upon nutrient deprivation. Next, we found that PSMB4 foci are not induced when IMR90 cells are exposed to oxidative stress ($H_2O_2$), heat shock (45 °C), genotoxic stress (ionizing radiation or UVC light) or hypoxia (1% $O_2$) (Fig. 3e). However, only proteasome inhibitors consistently provoked the formation of nuclear PSMD4 foci, but in a very small percentage of cells. Relevant signaling factors and cell survival were assessed to control for treatment efficacy (Fig. 3f, Supplementary Fig. 3e).

We then sought to evaluate the presence of proteasome foci in diverse mammalian cell types upon nutrient deprivation. We found that nuclear foci of PSMD4 or PSMD7 are found, with different frequencies, in several other normal, immortalized or tumoral mammalian cells (Fig. 4a and Supplementary Fig. 4). Proteasome foci are also observed in mouse embryonic stem cells (mESC) and differentiated 3T3L1 mouse adipocytes upon nutrient starvation (Fig. 4b, c). Interestingly, we noted that several cancer cells, i.e., T47D, PC3, MIA PaCa-2, have reduced ability to form proteasome foci following nutrient deprivation (Supplementary Fig. 4). Thus, we concluded that proteasome foci are a general response of mammalian cells to nutrient deprivation and might be modulated during oncogenic transformation. We termed the above-described nuclear structures: Starvation-Induced Proteasome Assemblies in the Nucleus (SIPAN).

**SIPAN are highly dynamic, undergo fusion events, and are reversible**. To investigate the dynamics of SIPAN, we surveyed SIPAN formation in IMR90 cells and found that these structures form as rapidly as 2–3 hrs with a maximum of cells harboring PSMD4 foci at 6–10 hrs post-nutrient deprivation (Fig. 5a). The percentage of cells with SIPAN, their signal intensity and number per cell all progressively increased during nutrient starvation (Fig. 5a–c). While signal intensity of SIPAN reaches a plateau, their average number per cell decrease at later time points, suggesting SIPAN fusion or resorption (Fig. 5b–c). To further investigate this, we conducted live imaging using IMR90 cells stably expressing PSMB4-GFP (Supplementary Movie 1, 2). Live-cell SIPAN have an apparent average size of 0.3–0.4 μm² (Fig. 5d) and can indeed undergo fusion events (Fig. 5e–g, Supplementary Movie 3). Further analysis of SIPAN mobility showed that these structures have a mean square displacement of 0.2 μm²/min

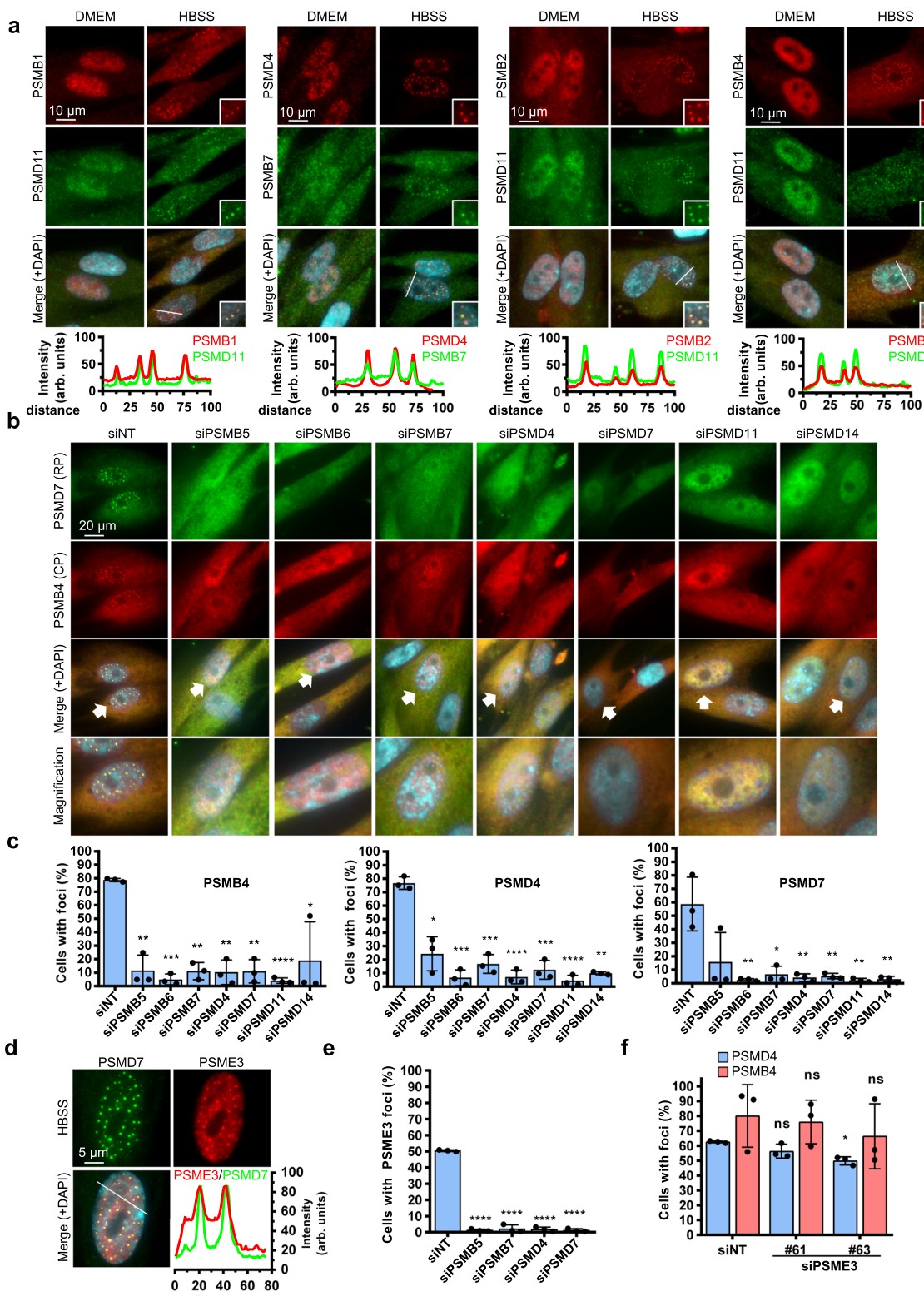

(Fig. 5h–j), three-fold higher for that reported for 53BP1 DNA damage foci[23].

To further determine whether SIPAN are reversible, we first induced their formation by depriving cells of nutrients and then replenished the cells with fresh culture medium. We observed that SIPAN, detected by PSMD4 immunostaining, dissipate within 60 min after the addition of complete culture medium (Fig. 5k–m and Supplementary Fig. 5a). Similar results were

obtained for GFP-tagged PSMB4 component of the CP (Supplementary Fig. 5b). PSME3 proteasome activator particle also dissipates following addition of culture medium (Fig. 5n, Supplementary Fig. 5c). As disassembly of multi-protein complexes might involve the AAA$^{+}$-type ATPase VCP/p97[24], we blocked this enzyme and determined the impact on SIPAN resolution. Nutrient-deprived IMR90 cells, with preformed SIPAN, were treated with various VCP/p97 inhibitors in normal

**Fig. 2 Proteasome integrity is required for foci formation upon nutrient deprivation. a** Immunostaining of proteasome components in IMR90 human fibroblasts showing that component of the CP and RP co-localize in nuclear foci following 6 hrs incubation in HBSS. The green (endogenous PSMD11 or PSMB7), red (endogenous PSMB1, PSMB2, PSMB4, or PSMD4), and blue (DAPI) signals were merged to indicate foci localization within the nucleus. The inset at the bottom right of each picture corresponds to a magnification of a small portion of the nucleus as indicated. Graphs in the bottom represent relative signal intensity of the indicated line for each component of the proteasome (representative from 3 independent experiments). **b** siRNA-mediated depletion of components of the RP or CP abolishes PSMD7 or PSMB4 foci formation. IMR90 cells were transfected twice with the indicated siRNAs. After three days, cells were treated with HBSS for 6 hrs and harvested for immunostaining of endogenous PSMB4 and PSMD7 proteins. The images at the bottom represent magnification of the merge as indicated by white arrows (representative from 3 independent experiments). **c** Cell counts of PSMD4, PSMB4, or PSMD7 foci following depletion of individual components of the proteasome ($n = 3$ independent experiments). **d** Immunostaining of PSME3 in IMR90 cells showing that this subunit of the 11S RP localizes with PSMD7 in nuclear foci following 6 hrs incubation in HBSS (representative from 5 independent experiments). **e** siRNA-mediated depletion of components of the RP or CP abolishes PSME3 foci formation. IMR90 cells were transfected with siRNAs of components of the RP or CP. After 3 days, cells were incubated in HBSS for 6 hrs and harvested for immunostaining of PSME3 ($n = 3$ independent experiments). **f** Depletion of PSME3 by siRNA does not affect PSMD4 or PSMB4 foci formation. IMR90 cells were transfected with siRNAs of components of the RP or CP. After 3 days, cells were incubated in HBSS for 6 hrs and harvested for immunostaining of PSME3 ($n = 3$ independent experiments). Data in **c**, **e**, **f** represent mean ± SD. *$P < 0.05$; **$P < 0.01$; ***$P < 0.001$; ****$P < 0.0001$; ns: not significant; 2-sided unpaired Student's $t$-test with Welch correction (**c**, **e**, **f**). Source data are provided as a Source data file.

---

culture medium for 1 h. We found that VCP/p97 chaperone is not required for SIPAN resolution (Fig. 5o). As expected, VCP inhibition results in cell death at later time points, indicating inhibitor efficacy under these experimental conditions (Supplementary Fig. 5d).

**SIPAN result from liquid–liquid phase separation (LLPS).** We sought to determine how SIPAN respond to abrupt physico-chemical changes of the cellular environment. SIPAN were assembled by incubation of IMR90 cells in HBSS and were subsequently subjected to other treatments. Interestingly, while SIPAN dissipate within 2 min upon incubation in 0.01% of Triton X-100, DNA damage-induced 53BP1 foci are resistant to concentrations of this detergent of up to 1% (Fig. 6a and Supplementary Fig. 6a, b). The nuclear staining of PSMD4 does not decrease during the initial time of detergent treatment, suggesting that SIPAN are dissipated in the nucleus as opposed to being expelled from this compartment (Fig. 6a and Supplementary Fig. 6a). Similar results are obtained with digitonin (Supplementary Fig. 6c), a mild detergent that permeabilizes cellular membranes with minimal effects on nuclear membrane[25]. Moreover, within the same period post-HBSS treatment, we did not observe a noticeable diminution in the signal of RNA Polymerase II following incubation with low concentration of digitonin (Supplementary Fig. 6d). Next, we conducted live-cell imaging of PSMB4-GFP and found that, upon treatment with 0.03% Triton X-100, SIPAN fluorescence signals become diffuse before reduction of PSMB4-GFP overall intensity and entry of propidium iodide (P.I.) into the nucleus (Supplementary Fig. 6e, Supplementary Movie 4). These data suggest that SIPAN continuously depend on physico-chemical determinants of the nucleus and cytoplasm that quickly dissipate upon discrete changes in the composition of the nucleocytoplasm.

A fundamental characteristic of LLPS is a thermodynamic equilibrium that can shift towards assembly or disassembly depending on the cellular environment[26]. First, we treated IMR90 cells with 1,6-hexanediol, an aliphatic alcohol, which disrupts weak hydrophobic interactions and LLPS[27], and found that this treatment causes rapid SIPAN dissolution (Fig. 6b, Supplementary Movie 5). Next, we wanted to determine whether SIPAN assembly/disassembly could be influenced by salt concentration, without plasma membrane permeabilization. We induced SIPAN in IMR90 cells, which were subsequently incubated in various detergent-free solutions for 2 min. We found that hypotonic treatments induce quick SIPAN dissipation (Fig. 6c and Supplementary Fig. 6f). Changing the pH of the hypotonic buffer from 6.8 to 8.8 had no impact on SIPAN dissolution, while

supplementing this minimal buffer with 100 mM or 400 mM of NaCl maintained SIPAN assembled (Fig. 6c, Supplementary Fig. 6g). Interestingly, live-cell imaging indicated that SIPAN dissipation is followed by quick recovery in the original locations, when hypotonic buffer is supplemented with 200 mM NaCl (Fig. 6d, Supplementary Fig. 6h and Supplementary Movie 6). Importantly, no staining of DNA with P.I. was observed, indicating that plasma membrane integrity is not compromised during the course of these treatments (Fig. 6d). Of note, a similar behavior of foci dissipation upon incubation in hypotonic buffer conditions is also observed for PML bodies, which also result from LLPS[28] (Supplementary Fig. 6g). Overall, these results indicate that SIPAN are highly sensitive to the physico-chemical environment of the cells and their assembly is governed, at least partly, by hydrophobic interactions.

To further investigate SIPAN dynamics, we conducted fluorescence recovery after photobleaching (FRAP) experiments in HCT116 cells stably expressing PSMB4-GFP. 53BP1-GFP and GFP-H2A, which are tightly associated to DNA[29,30], were included for comparison with DNA damage foci or high-density chromatin domains, respectively. We found that PSMB4-GFP fluorescence recovery is very rapid, while little or no apparent recovery of fluorescence was detected for 53BP1-GFP or histone GFP-H2A, respectively (Fig. 6e, Supplementary Movie 7–9). Taken altogether, these results indicate that SIPAN result from LLPS, and that these structures can be dynamically modulated in response to extracellular cues.

**Ubiquitin dynamics drive SIPAN formation.** We sought to determine whether ubiquitin signaling is involved in SIPAN formation. No manifest accumulation of ubiquitin conjugates is observed over time by immunoblotting in IMR90 or HCT116 following incubation in HBSS (Fig. 7a, Supplementary Fig. 7a). Interestingly, SIPAN are enriched with K48-conjugated ubiquitin indicating the presence of ubiquitinated proteins (Fig. 7b, c and Supplementary Fig. 7b). In contrast, while SUMO formed foci in untreated cells that likely correspond to PML bodies[31,32], we did not observe accumulation of this protein in SIPAN (Supplementary Fig. 7c). These results suggest that discrete ubiquitination events might play a role in SIPAN dynamics. This prompted us to determine whether increasing the pool of ubiquitinated proteins promotes SIPAN formation. We treated IMR90 with HBSS and MG132 for 3 hrs, time at which, little foci are formed following nutrient deprivation only. Indeed, inhibition of proteasome activity with MG132 accelerates SIPAN formation, as both foci number and intensity were increased upon nutrient starvation (Fig. 7d–f). In addition, inhibition of UAE1 ubiquitin-

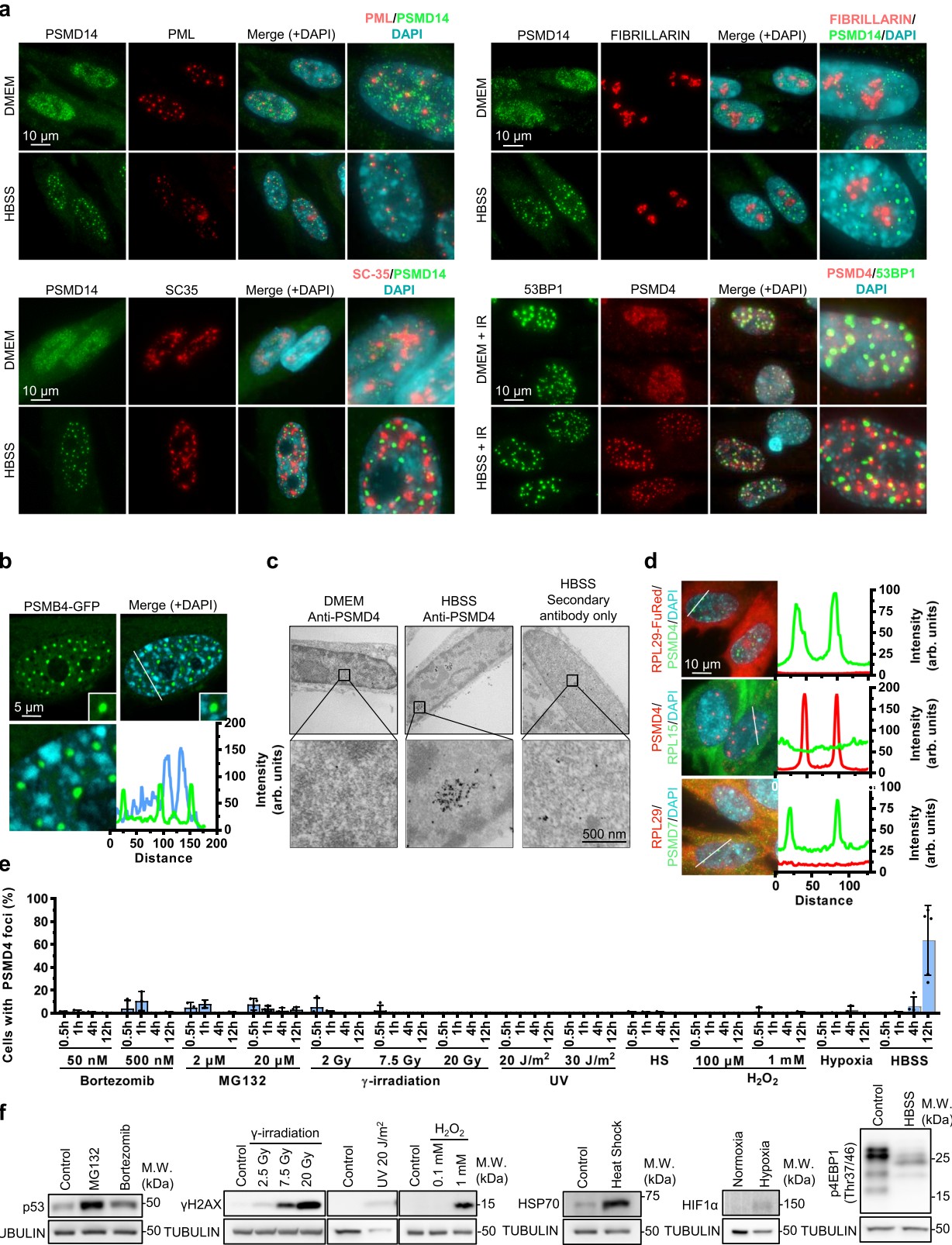

activating enzyme with TAK-243 prevents SIPAN formation (Fig. 7g, h, Supplementary Fig. 7d). Interestingly, continuous ubiquitination is required for SIPAN maintenance, as UAE1 inhibition results in SIPAN dissolution under conditions of nutrient deprivation (Fig. 7g, i). As expected, UAE1 inhibition does not impact SIPAN dissolution following replenishment of nutrient-deprived IMR90 cells with normal culture medium

(Fig. 7g, j). Next, we reasoned that ubiquitin removal from SIPAN might, however, be required for SIPAN resolution upon incubation in normal culture medium. Inhibition of the proteasome-associated deubiquitinases, ubiquitin C-terminal hydrolase 5 (UCHL5) and ubiquitin-specific peptidase 14 (USP14), using the small molecule b-AP15[33], prevents SIPAN resolution (Fig. 7g, k). On the other hand, proteasome inhibition does not impact

**Fig. 3 SIPAN are induced by metabolic stress and do not correspond to previously known nuclear structures. a** PSMD11 or PSMD4 proteasome foci do not correspond to any known nuclear foci, structure or bodies including PML bodies (PML staining), nuclei (Fibrillarin staining), nuclear speckles (SC35 staining) or DNA double strand break foci (53BP1 staining). Stainings were conducted following 6 hrs incubation in HBSS (representative from 3 independent experiments). **b** Confocal microscopy showing PSMB4-GFP foci in low-density chromatin regions (representative from 3 independent experiments). **c** Transmission electronic microscopy in conjunction with colloidal gold-based immunodetection of PSMD4. Following incubation of IMR90 cells in HBSS for 6 hrs, cells were fixed and processed for immunodetection and electronic microscopy analysis. PSMD4 condensates are membrane-less and located in regions with reduced chromatin density (representative from 2 independent experiments). **d** Ribosomal proteins do not co-localize with proteasome foci. Merges from Supplementary Fig. 3b–d. Graphs at the right represent relative signal intensity of the indicated line for each component of proteasome (representative from 2 independent experiments). **e, f** Proteasome foci are not observed in response to other stress conditions. **e** IMR90 cells were treated with various chemical or physical agents and endogenous PSMD4 was detected by immunostaining at the indicated times ($n = 3$ independent experiments). **f** Western blot analysis of proteins following treatments of cells shown in (**e**) to ensure treatment efficacy (representative from 2 independent experiments). Data in (**e**) represent the mean ± SD. Source data are provided as a Source data file.

SIPAN resolution following replenishment of normal culture medium (Fig. 7g, k). Thus, we concluded that continuous ubiquitination and deubiquitination cycles are critical for SIPAN dynamics.

**RAD23B is required for SIPAN formation**. Based on the data described above, we rationalized that ubiquitin-binding proteins, and notably proteasome shuttling factors, might be involved in SIPAN formation. We found that siRNA depletion of several ubiquitin-binding proteins and shuttling factors[34–37], notably RAD23B, inhibit SIPAN formation (Fig. 7l, Supplementary Fig. 7e). We validated the effect of RAD23B depletion using additional siRNAs targeting other regions of RAD23B mRNA (Fig. 7m). As expected, depletion of RAD23B does not affect the levels of proteasome components (Supplementary Fig. 7f). Endogenous RAD23B localizes in SIPAN (Fig. 7n). We also expressed RAD23B by lentiviral transduction in IMR90 and found that this factor localizes in SIPAN (Supplementary Fig. 7g). Of note, while depletion of RAD23A does not prevent SIPAN formation, this factor also localized in SIPAN (Fig. 7l, Supplementary Fig. 7h). Interestingly, deletion of the gene encoding the RAD23B orthologue in yeast[38], RAD23, also compromises the assembly of RPN5-GFP (PSMD12) into PSG, suggesting that the role of RAD23B in proteasome phase separation might be conserved throughout evolution (Fig. 7o, p).

Multivalent interactions ensured by intrinsically disordered regions and/or structured domains are key determinants in LLPS[39,40]. RAD23B is known to interact with the proteasome[41], and contains an ubiquitin-like domain (UBL) and two ubiquitin-binding motifs, UBA1 and UBA2, which can engage in multivalent interactions (Fig. 8a)[38,42]. Bioinformatics analysis of human RAD23B and yeast RAD23 showed that intrinsically disordered regions are mainly located between functional domains, while maximal hydrophobicity is found within UBL and UBA domains (Fig. 8a, Supplementary Fig 8a). As SIPAN appear to depend on hydrophobic interactions, we tested the requirement of the above-mentioned domains for their formations. We expressed several mutants lacking key domains of RAD23B and found that the UBL and UBA domains are required for SIPAN formation (Fig. 8b, c and Supplementary Fig. 8b, c). We subsequently purified human RAD23B from bacteria and found that this factor undergoes LLPS in the presence of Ficoll 400 molecular crowding agent, as the protein mixture became turbid and liquid droplets could be readily observed (Fig. 8d–f). In contrast, BSA did not undergo LLPS in the same conditions. RAD23B LLPS depends on protein and crowder concentration, and is induced by other molecular crowding agents (Fig. 8g, h and Supplementary 8d). Of note, removal of N-terminal His tag on RAD23B has little or no impact on protein droplet formation (Supplementary Fig. 8e). Consistent with its ability to undergo phase separation in vitro, RAD23B droplets are disrupted by 1,6

hexanediol (Fig. 8i, Supplementary Fig 8f). Significantly, live imaging indicates that RAD23B droplets undergo fusion events in vitro (Fig. 8j, Supplementary Movie 10). Finally, we found that deletion of RAD23B domains, notably UBA2 reduced RAD23B ability to undergo phase separation (Fig. 8k, l).

**Deprivation of non-essential amino acids is responsible for SIPAN formation**. To further dissect the mechanism that govern SIPAN formation, we supplemented the nutrient-free HBSS solution with specific nutrients and found that addition or removal of glucose or pyruvate does not significantly impact SIPAN formation in IMR90 cells (Fig. 9a–c). In contrast, addition of fetal bovine serum (FBS) or amino acid mixture inhibit SIPAN formation (Fig. 9a–c). Preserving amino acid pools by inhibiting residual protein synthesis with cycloheximide (CHX), during incubation in HBSS, also blocks SIPAN formation (Fig. 9d–f). Conversely, preventing amino acid recycling by blocking autophagy with chloroquine or 3-methyladenine accelerates SIPAN formation (Fig. 9d–f, Supplementary Movie 11). Of note, mTOR is inhibited during nutrient deprivation and further blocking of this pathway with rapamycin or torin1, does not affect SIPAN formation following nutrient deprivation (Fig. 9d, e). Moreover, no formation of SIPAN was detected following mTOR inhibition in complete culture medium (Supplementary Fig. 9a, b). These results also suggest that SIPAN formation is either independent of mTOR signaling or that inhibition of this pathway is not sufficient to licence their formation. Next, we deconvolved the amino acid mixture and treated IMR90 cells with HBSS supplemented with 1 mM of individual amino acids. Interestingly, we found that most non-essential amino acids (NEAA), rather than essential amino acids (EAA), strongly prevent SIPAN formation (Fig. 9g). Similar results were observed on SIPAN resolution, following amino acid replenishment in IMR90 cells (Fig. 9h). More pronounced effects were noticed with higher concentrations of amino acids (Supplementary Fig. 9c, d). Live-cell imaging of IMR90 expressing PSMB4-GFP further confirmed the rapid resolution of SIPAN following addition of NEAA (Supplementary Movie 12). We concluded that deprivation of amino acids, and especially non-essential amino acid, is a major determinant of SIPAN formation and resolution in mammalian cells.

**RAD23B and PSME3 provide a link between amino acid supply, SIPAN formation, and apoptosis**. In yeast, PSG promote viability and cell fitness following carbon starvation[18]. To provide insights into the significance of SIPAN formation in mammalian cells, we first sought to determine the state of proteasome activity upon amino acid deprivation. We were unable to extract SIPAN to conduct in vitro activity studies, as these foci are quickly dissipated in the nucleus upon detergent treatment or hypotonic cell lysis (Fig. 6a, c, d). We, therefore, preformed SIPAN in IMR90

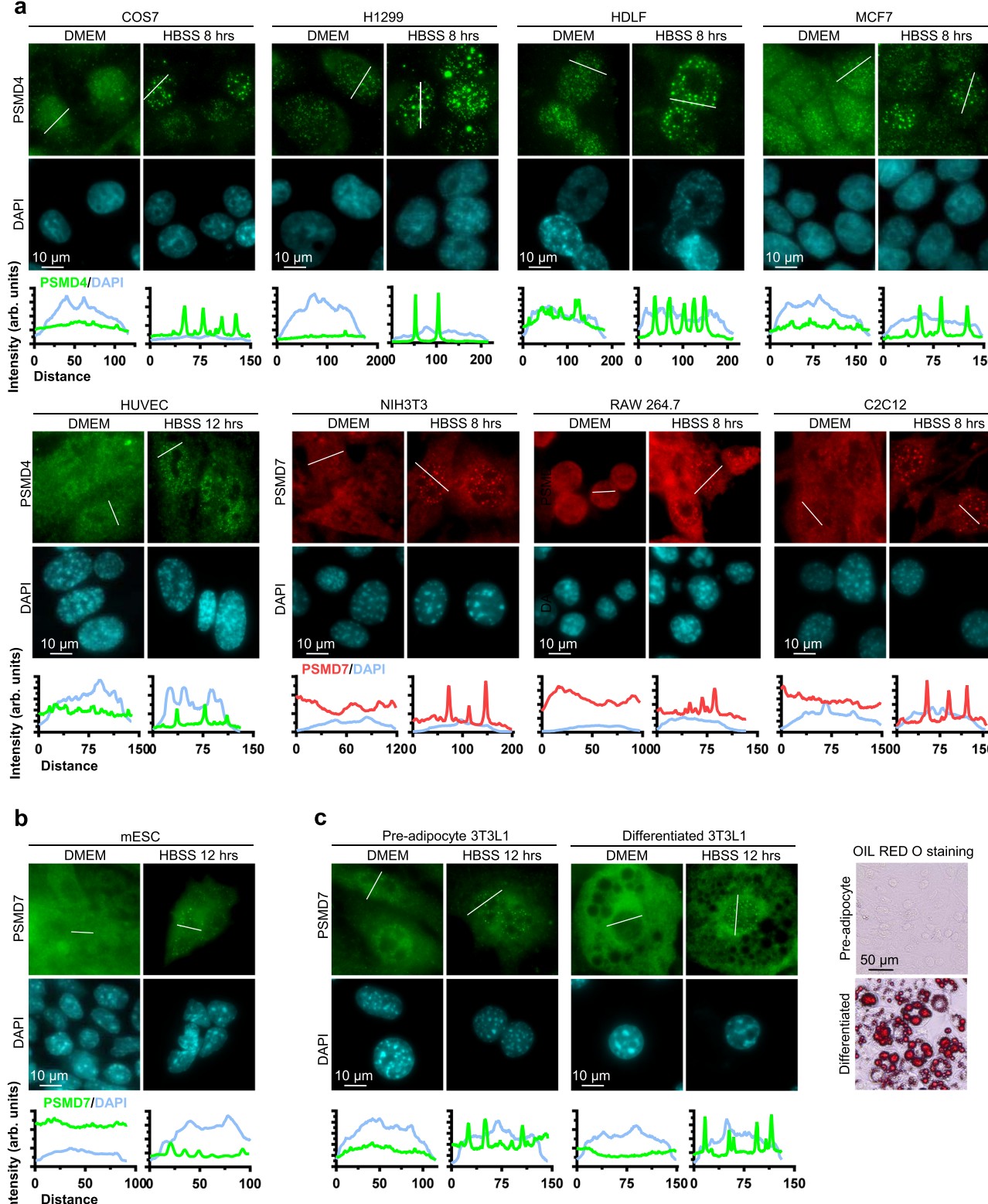

**Fig. 4 SIPAN are a general phenomenon common to many mammalian cell types. a**, **b** Immunostaining of endogenous PSMD4 or PSMD7 in diverse cell types, showing that these proteins localize in nuclear foci following incubation in HBSS. SIPAN are observed in normal primary cells (e.g., HDLF: primary human lung fibroblasts, HUVEC: human endothelial cells, mESC: mouse embryonic stem cell); immortalized cells (e.g., NIH3T3: mouse embryonic fibroblasts, C2C12: mouse myoblast cell line, 3T3L1: mouse preadipocytes); transformed (e.g., Cos7: Transformed monkey kidney cells, RAW264.7: Abelson murine leukemia virus transformed macrophage) and tumoral cells (MCF7: human breast cancer, H1299: human non-small cell lung cancer). **c** Immunostaining of PSMD7 in mouse pre-adipocytes 3T3L1 and differentiated adipocytes showing that this protein localizes in nuclear foci following incubation in HBSS. Right panel, Oil Red O staining was conducted to control for adipocyte differentiation (representative images from 3 independent experiments).

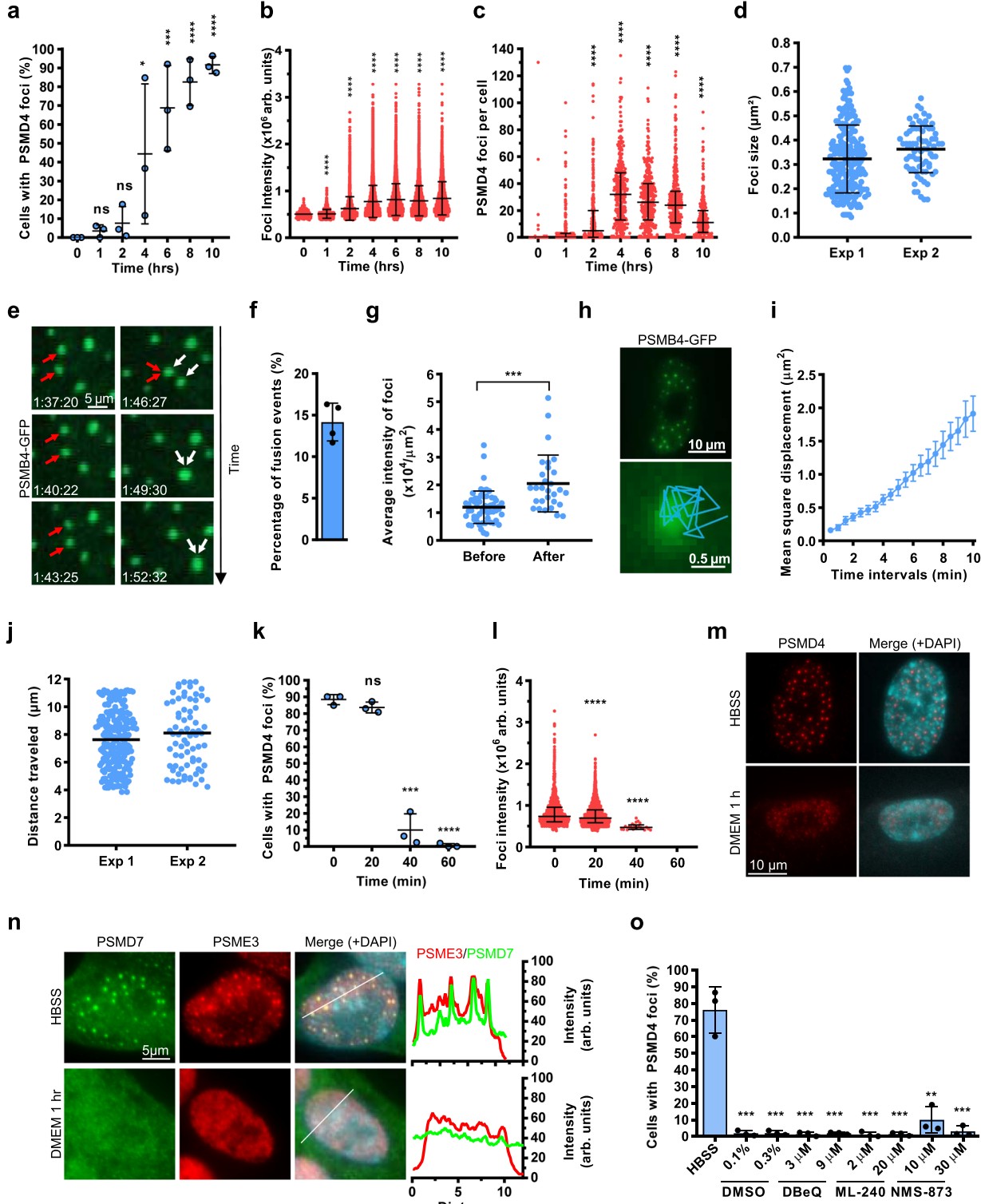

cells by incubation in HBSS for 8 hrs followed by treatment with MG132 in the same solution for 1 hr. Following proteasome inhibition, we observed increased levels of several short-lived nuclear stress-associated transcription factors, including p53, C-FOS, and C-JUN, suggesting that nuclear proteasome is still active under conditions of SIPAN formation (Fig. 10a). Next, we used a Me4BodipyFL-Ahx3Leu3VS (Proteasome Activity Probe)[43], which accumulates in SIPAN after hydrolysis, and this also suggested that the proteasome is active in SIPAN (Fig. 10b).

To further define the biological significance of SIPAN, we investigated the link between SIPAN formation and cell viability. Treatment of IMR90 cells with chloroquine or 3-MA, which promotes SIPAN formation, diminishes cell survival following incubation in HBSS (Supplementary Fig. 9e, f). In contrast, treatment with CHX, which dampens SIPAN formation, protects cells from loss of viability relative to treatment with HBSS alone (Supplementary Fig. 9e, f). As expected, the CASPASE inhibitor Z-VAD protects IMR90 cells from undergoing apoptosis following

**Fig. 5 Rapid dynamics of SIPAN assembly and resolution. a–c** Kinetics of SIPAN formation in IMR90 fibroblasts. **a** Cells were incubated in HBSS and harvested for immunostaining with PSMD4 antibody. Cells with more than 10 foci were counted ($n = 3$ independent experiments). **b** Signal intensity of SIPAN. Images from control and starved cells were used to estimate SIPAN signals (representative from 3 independent experiments). Arbitrary units (arb. units). **c** Quantification of the number of SIPAN per cell at different time points post-starvation (representative from 3 independent experiments). **d** Quantification of the size of the traced PSMB4-GFP foci ($n = 2$ independent experiments). **e** Time lapse from live-cell imaging indicating SIPAN fusion in vivo. IMR90 cells expressing PSMB4-GFP were incubated in HBSS and used for live imaging. Two fusion events are indicated by arrow of different colors (representative from 3 independent experiments). **f** Determination of the frequency of PSMB4-GFP foci fusion events over 1 hr (representative from 2 independent experiments). **g** Comparison of the average intensity of PSMB4-GFP foci in the corresponding cells, before and after fusion events. Note the increase in average intensity after fusion (representative from 2 independent experiments). **h** A representative image of a 10-min mobility trace of a PSMB4-GFP focus (representative from 2 independent experiments). **i** A mean-square displacement measurement plot of PSMB4-GFP foci is represented (representative from 2 independent experiments). **j** Quantification of the distance traveled by PSMB4-GFP foci over 10 min in two independent experiments. Each data point in the distribution plot represents a traced focus ($n = 2$ independent experiments). **k–m** SIPAN are reversible. SIPAN formation is induced for 8 hrs and then IMR90 cells were replenished with normal culture medium and harvested for immunostaining. **k** Cells with more than 10 foci were counted ($n = 3$ independent experiments). **l** Signal intensity of SIPAN from nutrient-starved cells and nutrient-replenished cells (representative from 3 independent experiments). **m** Cell nucleus showing dissipation of PSMD4 signals after medium replenishment (representative from 3 independent experiments). **n** Resolution of PSME3 foci after nutrient replenishment. IMR90 were incubated 6 hrs with HBSS and then incubated with culture media for 1 hr (representative from 3 independent experiments). **o** VCP chaperone is not required for SIPAN resolution. IMR90 cells were treated with HBSS solution for 6 hrs and then treated with various inhibitors at the indicated concentrations in complete medium for 1 hr ($n = 3$ independent experiments). Cells with more than 10 foci are counted. Data represent mean ± SD (**a, d, f, g, i, j, k,o**) or median with interquartile range for one representative experiment (**b, c, l**). *$P < 0.05$; **$P < 0.01$; ***$P < 0.001$; ****$P < 0.0001$; ns: not significant; one-way ANOVA with Holm-Sidak's (**a, k**) or Kruskal–Wallis test with Dunn's test (**b, c, l**) or 2-sided unpaired Student $t$-test (**g, o**). Source data are provided as a Source data file.

---

nutrient deprivation (Supplementary Fig. 9e, f). NEAA pool, which prevents SIPAN formation, also increases cell viability upon incubation in HBSS; whereas EAA pool decreases cell viability (Supplementary Fig. 9g, h). We then used several individual EAA (lysine, arginine, and tryptophan), or NEAA (serine, glycine, glutamine, asparagine, and alanine) whose presence in HBSS has no noticeable effect or inhibit SIPAN formation, respectively (Fig. 9g, h, Supplementary Fig. 9c, d). We found that while individual EAA decrease cell viability, individual NEAA have either no effect or significantly increase cell viability (Supplementary Fig. 9g, h). Importantly, we found that depletion of RAD23B or PSME3 results in increased cell viability following amino acid depletion, as determined by MTT reduction assay and crystal violet staining (Fig. 10c; Supplementary Fig. 10a). Immunoblotting detection of PARP1 apoptotic cleavage and FACS analysis for SubG1 cell population also indicated that depletion of RAD23B or PSME3 protects from cell death induced by amino acid depletion (Fig. 10d, e). Similar results were obtained in HDLF, another normal human fibroblast cell line (Supplementary Fig. 10b–d). Finally, depletion of RAD23A, which has no impact on SIPAN assembly, does not promote cell survival in IMR90 or HDLF cells (Fig. 7l; Supplementary Fig. 10e, f). Based on these results altogether, we concluded that SIPAN formation is associated with amino acid starvation-induced cell death.

Next, we sought to investigate several factors suspected to be involved in nutrient deprivation-induced cell death and found that the tumor suppressor p53 as well as several of its pro-apoptotic target genes including BIM, PUMA, BAX and NOXA, are upregulated following amino acid depletion (Fig. 10f). Notably, NOXA has been previously involved in metabolic stress responses[44,45]. Depletion of RAD23B or PSME3 prevents NOXA upregulation (Fig. 10g, Supplementary Fig. 10d). Moreover, depletion of p53 or NOXA in IMR90 or HDLF promote cell survival upon amino acid starvation, as determined by MTT, phase contrast microscopy, and immunoblotting for PARP1 cleavage (Fig. 10h–j, Supplementary Fig. 10g–k). These results provide further evidence for the link between the proteasome regulators RAD23B and PSME3 and p53/NOXA-mediated apoptosis in response to amino acid deprivation.

Based on the results shown above and taking into account that certain cancer cells showed reduced ability to form SIPAN (Supplementary Fig. 4), we rationalized that oncogenic

transformation and/or tumor environment, both of which induce profound metabolic changes[46], might further select for cells with reduced ability to form SIPAN. Thus, we induced oncogenic transformation of IMR90 cells by inhibiting p53, through MDM2 overexpression, with concomitant expression of two collaborating oncogenes, E1A and RAS$^{G12V}$ [47]. The resulting transformed cells were further injected into immunodeficient mice and the tumors were isolated, and cells dissociated and cultured (Fig. 10k). Using normal, transformed (IMR90$^{Trans}$) and tumoral (IMR90$^{Tumor}$) cells, we then analyzed SIPAN formation and survival upon amino acid deprivation. We found that oncogenic transformation results in reduced ability of the cells to form SIPAN (Fig. 10l). Additionally, passage of transformed cells through tumors further decreased the ability of the cells to form SIPAN (Fig. 10l, Supplementary Fig. 10l). Moreover, propidium iodide incorporation-based cell viability assay indicated that transformed and tumoral IMR90 cells have increased cell survival ability in response to nutrient deprivation (Fig. 10m, n). These results provide further evidence for a possible role for SIPAN as a defense mechanism against malignant transformation.

## Discussion

In this study, we demonstrate that the mammalian proteasome undergoes LLPS upon amino acid deprivation, resulting in the formation of nuclear membrane-less organelles we termed SIPAN. SIPAN assemble in nuclear regions of low chromatin density and the proteasome within these structures is labeled with a fluorescent activity probe, suggesting that SIPAN are subnuclear sites of active proteolysis. SIPAN undergo fusion events, can be dissipated and reassembled, and rapidly exchange components with the nuclear milieu, reflecting their fast turnover rates and highly dynamic nature. Moreover, SIPAN are nuclear sub-structures common to mammalian cells of various origins. Recently, it was also shown that, under hyperosmotic stress conditions, the proteasome undergoes LLPS and form nuclear foci that target ribosomal proteins[22]. While, no ribosomal proteins were observed in SIPAN, these data altogether indicate that LLPS-mediated proteasome foci formation is a major determinant of mammalian stress responses. Indeed, SIPAN are associated with nutrient deprivation-induced cell death (Fig. 10o).

We uncovered a previously unappreciated link between proteasome dynamics and mammalian cell metabolism. Interestingly,

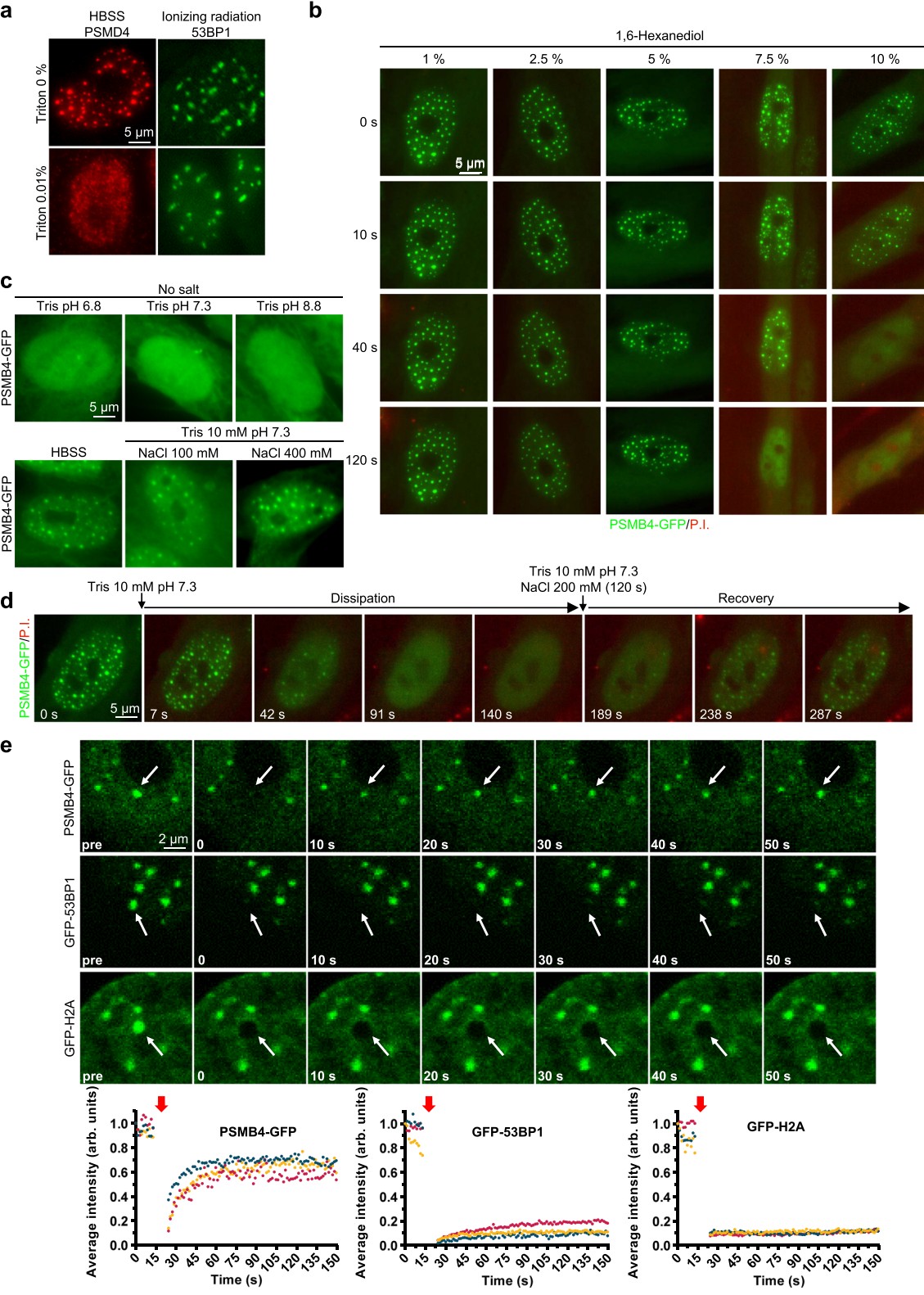

our results highlight significant evolutionary divergence in the responses of mammalian and yeast proteasomes to nutrient deprivation. (i) SIPAN form in the nucleus, whereas yeast proteasome is exported from the nucleus and form PSG structures in the cytoplasm[16,18]. (ii) SIPAN are formed following amino acid, but not glucose deprivation, while the opposite situation occurs in yeast[18]. (iii) Our data suggest that proteasome foci promote cell death in mammalian cells; however, the reverse outcome occurs in yeast, i.e., PSG improve cell fitness and viability[18]. (iv) Finally, while SIPAN contain conjugated ubiquitin, PSG contain free ubiquitin[17]. The reasons that can explain these differences between mammalian and yeast proteasome structures are currently unknown, but we emphasize that yeast and high-order metazoan present both common (conserved) and distinct metabolic responses to nutrient deprivation[48]. Wild yeast can face dramatic changes in their environment and need to adapt quickly

**Fig. 6 SIPAN are highly dynamic. a** SIPAN dissipate rapidly following incubation with very low concentration of Triton X-100 detergent. IMR90 cells with preformed proteasome foci were treated with Triton X-100 in HBSS for 2 min and then harvested for PSMD4 immunostaining. For 53BP1 foci, IMR90 cells were treated with ionizing radiations for 4 hrs and then used for detergent treatment in HBSS for 2 min before immunostaining. **b** Time lapse from live-cell imaging indicating SIPAN dissipation following incubation with 1,6-hexanediol. **c** SIPAN dissipate following incubation in hypotonic buffers. IMR90 cells expressing PSMB4-GFP were incubated in HBSS for 6 hrs and then treated as indicated before fixation and fluorescence microscopy. **d** Time lapse from live-cell imaging indicating SIPAN dissipation and recovery following incubation in Tris 10 mM pH 7.3 followed by NaCl 200 mM in Tris 10 mM pH 7.3, respectively. IMR90 cells expressing PSMB4-GFP were incubated in HBSS and used for live-imaging. **e** Fluorescence recovery after photobleaching (FRAP) of SIPAN. Bleaching of PSMB4-GFP foci indicate that SIPAN appear rapidly in their original foci following bleaching, while low or no recovery of fluorescence was detected for 53BP1 foci or histone H2A nuclear domains. The red arrows indicate the time of laser bleaching (graphs are from 3 independent experiments and the images are representative from 3 independent experiments). Source data are provided as a Source data file.

to maintain survival. In contrast, most of the cells of high-order multicellular organisms are in stable tissular microenvironments, with a generally stable and constant supply of nutrients. In addition, the utilization of nutrients in higher eukaryotes, such as mammalians, depends on growth factor signaling. Moreover, in these organisms, nutrient deprivation is usually a condition of tissue or organ stress[49]. On the other hand, common features unify the responses of the yeast and mammalian proteasomes to nutrient deprivation. The formation of SIPAN or PSG is reversible, as these structures dissipate quickly when nutrients are replenished. SIPAN and PSG contain ubiquitin, and the mammalian ubiquitin-binding factor RAD23B and its yeast orthologue RAD23 are important regulators of their formation. Thus, RAD23B or RAD23 provide a link between ubiquitin signaling pathways and the dynamics of SIPAN or PSG. Nonetheless, we cannot exclude, at this point, that PSG and SIPAN are unrelated structures in terms of composition, dynamics and functional significance.

One explanation of SIPAN formation is that, under conditions of nutrient deprivation, increase of E3 ubiquitin ligase activity and/or decreased deubiquitinase activity results in the accumulation of a subset of ubiquitinated substrates. This, in turn, triggers proteasome LLPS, through interaction with RAD23B and other ubiquitin receptors. Consistent with this notion, we found that (i) proteasome inhibition accelerates SIPAN formation, (ii) inhibition of ubiquitination, using UAE1 inhibitor, prevents SIPAN formation and maintenance without altering their resolution upon amino acid replenishment, and (iii) proteasome deubiquitinase inhibition prevents SIPAN resolution. Thus, our studies altogether support a model whereby K48 chain linked ubiquitination promotes LLPS of the mammalian proteasome. Interestingly, a parallel can be made with K63-linked polyubiquitin chains which drives LLPS of the p62 scaffold protein resulting in autophagosome formation and selective autophagy[50,51].

Mechanistically, we found that the multiple RAD23B domains are required for SIPAN formation. The multivalent interactions ensured by UBL and UBA domains might link ubiquitinated substrates to the proteasome and thereby drive LLPS in the nucleus. Nonetheless, we cannot exclude that a combination of domain-mediated interactions and weak interactions involving non-organized regions act in a concerted manner to promote proteasome LLPS. It is noteworthy that another proteasome shuttling factor, UBQLN2, with similarities to RAD23B, also uses its multiple domains to undergo LLPS and association with stress granules in the cytoplasm[52]. However, in this case, ubiquitin was found to inhibit LLPS mediated by UBQLN2. Further studies are required to determine how ubiquitin promotes or inhibits LLPS.

Protein ubiquitination per se is not sufficient to trigger SIPAN formation, as proteasome inhibition alone, under conditions of nutrient availability, barely induces SIPAN formation. On the other hand, PSME3, which activates proteasome in ubiquitin-independent manner, is also recruited and assembled in SIPAN.

Thus, ubiquitin-independent signals are also needed to licence SIPAN formation. Importantly, our results indicate that the cellular levels of amino acids, and notably NEAA, are key determinants in SIPAN assembly and disassembly. Interestingly, NEAA are actively involved in intermediate carbon metabolism, nucleotide metabolism and signaling processes[53,54], and their presence might favor the formation of metabolic intermediates that might act as hydrotropes, thereby preventing LLPS[55].

What is the biological significance of LLPS-mediated nuclear proteasome in mammalian cells? Our data suggest that SIPAN are disadvantageous for the fitness of individual cells. It is possible that, in multicellular organisms, this response has evolved for the benefit of tissues and organs rather than that of individual cells. In multiple pathological conditions including wound, organ injury, and tissue ischemia, cells experience drastic nutrient deprivation that could trigger SIPAN formation and eventually cell death. For instance, SIPAN, by promoting cell death upon nutrient deprivation, might contribute to tissue and organ homeostasis by decreasing competition between cells for nutrients. In addition, release of constituents by dying cells might be beneficial for surrounding cells during periods of nutrient deprivation. This response might further contribute to tissue repair and homeostasis.

Our data also suggest a link between SIPAN and tumor development/progression. First, certain cancer cells including T47D, PC3, MIA PaCa-2, have reduced ability to form SIPAN. Second, nutrient deprivation induces p53-dependent apoptosis with a notable upregulation of the p53 target gene *NOXA*; and inhibition of RAD23B and PSME3 prevents p53/NOXA upregulation and apoptosis. Third, we also found that oncogenic transformation of normal human fibroblasts results in reduced cell ability to form SIPAN and resistance to apoptosis induced by nutrient deprivation. Based on these results, SIPAN formation might ensure tumor suppression by preventing the survival and propagation of cells that have been exposed to extreme metabolic changes, which, in turn, can induce genetic or epigenetic changes that might promote oncogenic transformation. Interestingly, transformed cells that were subjected to tumor formation have a highly reduced ability to form SIPAN, suggesting that further cell selection might take place during tumor progression. Indeed, nutrient scarcity within the tumor microenvironment, i.e., poorly vascularized regions, is known to impose a selective pressure that can further impact on cancer cell progression. For instance, in pancreatic ductal adenocarcinoma, a poorly vascularized tumor type, it was revealed that certain NEAA, and particularly glutamine and serine, are depleted in tumors, while EAA levels are increased[56]. Glutamine, which has the strongest effect on SIPAN formation, is often depleted in the central region of solid tumors compared to other amino acids[57]. Clearly, further manipulation of metabolic and signaling pathways that link proteasome LLPS and amino acid sensing is expected to provide additional insights into the role of SIPAN in physiology and human disease.

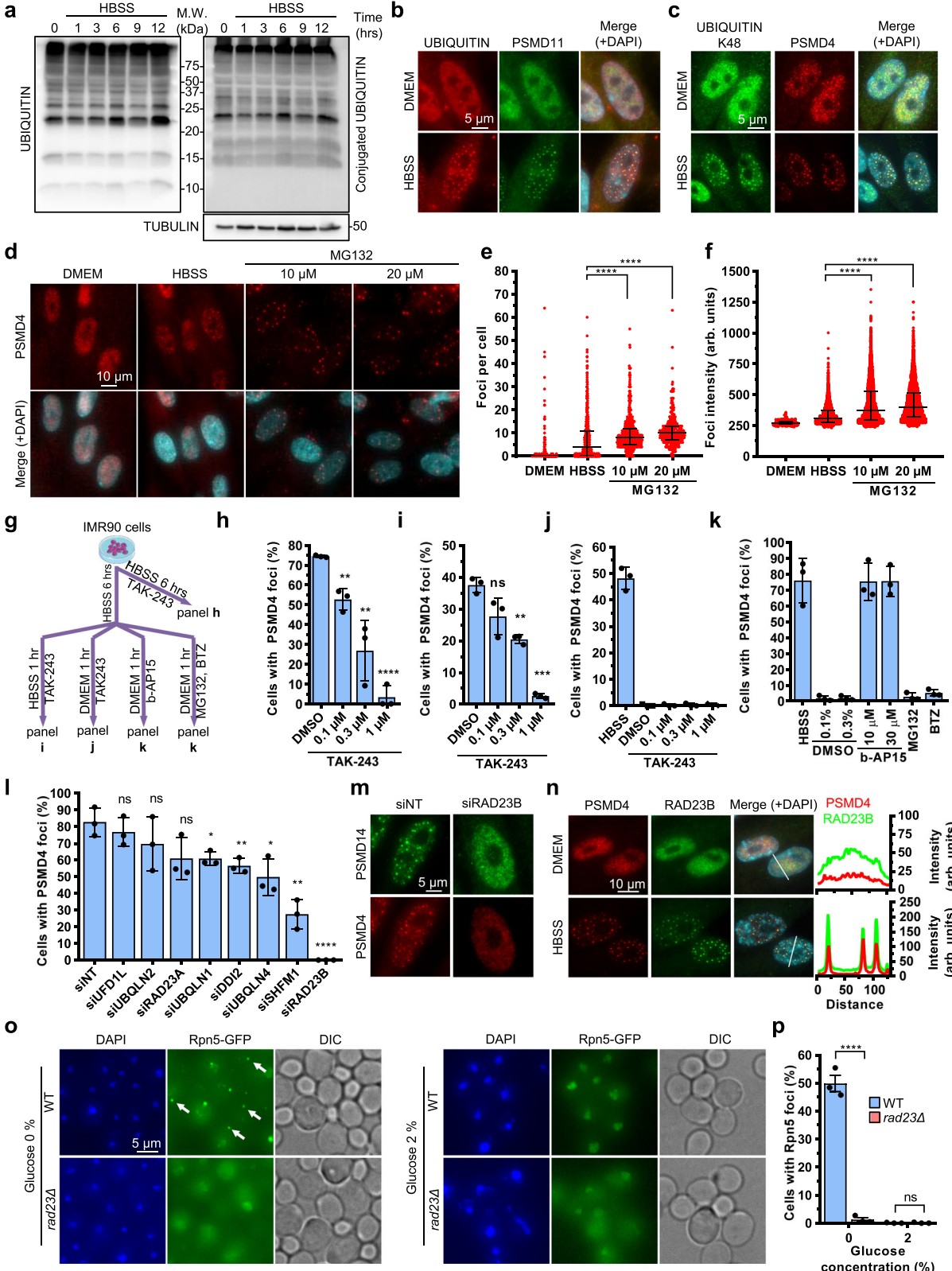

## Methods

**Plasmids.** The cDNAs of human *PSMB4*, *PSMB5*, *PSMD14*, and *PSMD12* genes were generated in a modified pBluescript using gene synthesis (Biobasic). The constructs were then subcloned as a fusion with the *GFP* gene and recombined using gene LR-clonase in lentiviral expression vectors as previously described[58]. SiRNA-resistant constructs for human *RAD23B* and its corresponding mutants *RAD23B* ΔUBL, *RAD23B* ΔUBA1, *RAD23B* ΔUBA2, *RAD23B* ΔUBA1/UBA2, and *RAD23B* ΔSTI were generated using gene synthesis (Biobasic) and then

recombined into lentiviral expression constructs. PAX2 (#35002) and pMD2G (#12259) lentiviral packaging plasmids were from Addgene. Human histone *H2A* was generated by gene synthesis in pENTR gateway plasmid and then recombined into pDEST *GFP* plasmid. For bacterial expression of RAD23B, His-tagged human *RAD23B* was generated by recombination of pBluescript into pDEST-His expression vector. pcDNA5-FRT/TO-*eGFP-53BP1* (Addgene #60813) is used to express GFP-53BP1.

**Fig. 7 RAD23B is required for SIPAN formation. a** Levels of ubiquitin and conjugated ubiquitin following nutrient deprivation in IMR90 cells. Cells were incubated in HBSS solution and harvested at different time points for immunoblotting with anti-ubiquitin or anti-conjugated ubiquitin FK2 antibodies (representative from 2 independent experiments). **b, c** Ubiquitin and K48-conjugated ubiquitin co-localize with SIPAN. Immunostaining of PSMD11 or PSMD4 and ubiquitin or K48 ubiquitin chains following nutrient starvation. IMR90 cells were incubated in HBSS for 6 hrs and harvested for immunostaining as indicated (representative from 3 independent experiments). **d** MG132 treatment increase SIPAN intensity and promote their formation. IMR90 cells were incubated in HBSS for 3 hrs in the presence or absence of MG132, and cells were harvested for immunostaining (representative from 3 independent experiments). Foci per cell (**e**) and foci intensity (**f**) were measured and represented by violin plot (representative from 3 independent experiments). **g** Schematic representation of treatment experiments with TAK-243, MG132, Bortezomib (BTZ) and b-AP15 inhibitors. **h** E1 ubiquitin-activating enzyme is required for SIPAN formation. Cells were incubated with TAK-243 E1 inhibitor for 6 hrs in HBSS ($n = 3$ independent experiments). **i** Continuous ubiquitination is required for SIPAN stability. Pre-formed SIPAN were treated with TAK-243 in HBSS for 1 hr ($n = 3$ independent experiments). **j** Ubiquitination is not required for SIPAN resolution. IMR90 cells were treated with TAK-243 for 1 hr in complete media ($n = 3$ independent experiments). **k** Deubiquitination is required for SIPAN resolution. Cells were incubated in HBSS solution for 6 hrs and then treated with b-AP15 deubiquitinase inhibitor, MG132, or BTZ in normal culture medium for 1 hr ($n = 3$ independent experiments). **l** RAD23B and other ubiquitin receptors are required for SIPAN formation. Following siRNA depletion of several ubiquitin-binding proteins and shuttling factors, IMR90 cells were incubated in HBSS for 6 hrs and harvested for immunostaining for SIPAN formation ($n = 3$ independent experiments). **m** Validation of RAD23B using additional siRNAs. Following siRNA transfection, IMR90 cells were incubated in HBSS for 6 hrs and harvested for immunostaining for SIPAN formation (representative from 3 independent experiments). **n** RAD23B is assembled in SIPAN following nutrient starvation. IMR90 cells were incubated in HBSS for 6 hrs and harvested for immunostaining for endogenous RAD23B and PSMD4 (representative from 3 independent experiments). **o, p** In yeast, RAD23 is important for PSGs formation under conditions of carbon depletion ($n = 3$ independent experiments). Data in graphs **h, i, j, k, l, p** represent the mean ± SD. 2-sided unpaired Student's *t*-test (**h, i, l, p**). Data in **e, f** represent the median with interquartile range for one representative experiment. 2-sided Mann–Whitney test (**e, f**). *$P < 0.05$; **$P < 0.01$; ***$P < 0.001$; ****$P < 0.0001$; ns: not significant. Source data are provided as a Source data file.

For the bacteria production of RAD23B without tag, 6His-TEV-RAD23B construct was generated by inserting the DNA sequence for the TEV cleavage site into pDEST-RAD23B. Site-directed mutagenesis was performed using primers TEV_RAD23B_F with TEV_RAD23B_R for RAD23B. Successful insertion of the TEV site was confirmed using DNA sequencing.

**Cell culture.** Human primary lung fibroblasts IMR90 and HDLF cells, MCF7 human breast cancer cell line, human non-small cell lung carcinoma NCI-H1299, AT3 mouse androgen-independent prostate cancer cells, RAW 264.7 murine macrophage cell line, HCT116 human colon cancer cell line, HeLa human cervical cancer cell line, T47D human breast cancer cell line, MDA-MB-231 human breast cancer cell line, PC3 human prostate cancer cell line, MIA PaCa-2 human pancreatic cancer cell line, LLC mouse Lewis lung carcinoma cell line, murine C2C12 myoblasts, 3T3-L1 mouse preadipocytes, NIH3T3 mouse fibroblasts were cultured in Dulbecco's modified Eagle's medium (DMEM) supplemented with 10% fetal bovine serum (FBS), 1% L-glutamine and 1% penicillin/streptomycin. Transformed monkey kidney cells Cos-7, human embryonic kidney HEK293FT cells were cultured in DMEM supplemented with 5% new born calf serum. Mouse embryonic stem cells (mESCs) were cultured on gelatin-coated plates in DMEM medium containing 15% FBS, 1% L-glutamine, penicillin/streptomycin, 0.1 mM β-mercaptoethanol, 0.1 mM Non-essential amino acids, 1 mM sodium pyruvate and 1000 U/ml of leukemia inhibitory factor (LIF) (Life technologies). HUVEC cells were cultured in EndoGRO basal medium (Millipore) supplemented with EndoGRO-VEGF Supplement (Millipore, containing 5 ng/mL recombinant human VEGF, 5 ng/mL recombinant human EGF, 5 ng/mL recombinant human FGF basic, 5 ng/mL recombinant human IGF-1, 50 μg/mL ascorbic acid, 1 μg/mL hydrocortisone hemisuccinate, 0.75 U/mL heparin sulfate, 1 % mM L-glutamine, 2 % FBS, 100 U/mL penicillin, 100 μg/ml streptomycin. Confluent 3T3TL1 were incubated in DMEM differentiation medium supplemented with 10% bovine serum, 1% glutamine, 1% penicillin/streptomycin, 1 μM dexamethasone, 1 μg/ml insulin, and 500 μM isobutylmethylxanthine (IBMX). Two days post-induction, culture medium was changed for DMEM medium supplemented with 10% FBS, 1% glutamine, 1% penicillin/streptomycin, and 1 μg/ml insulin. Media were changed every 2 days and cells were harvested at the indicated time points.

**Nutrient deprivation and chemical treatments.** Following cell culture medium removal, cells were washed three times with PBS (137 mM NaCl, 2.7 mM KCl, 10 mM Na$_2$HPO$_4$, 1.76 mM KH$_2$PO$_4$, pH 7.4) and then incubated in HBSS (137 mM NaCl, 5.4 mM KCl, 0.34 mM Na$_2$HPO$_4$, 1 g/L glucose, 0.44 mM KH$_2$PO$_4$, 1.3 mM CaCl$_2$, 0.81 mM MgSO$_4$, 4.17 mM NaHCO$_3$) supplemented with 10 mM HEPES pH 7.3 and 1% penicillin/streptomycin. HBSS was omitted from glucose or supplemented with other nutrients as indicated in figure legends. For UV treatments, IMR90 cell monolayers grown on coverslips in 6-well culture plates were washed twice with PBS and incubated in 2 ml of PBS, followed by UVC irradiation with a Philips G25T8 germicidal lamp at fluency of 0.2 J/m$^2$/s for a total dose of 20 or 30 J/m$^2$. H$_2$O$_2$ was diluted in culture medium at 1 mM and 100 μM. Hypoxia was conducted by incubating IMR90 with 1% of O$_2$ using a standard hypoxia chamber. Genotoxic stress was induced by treating cells with IR at 2, 2.5, 7.5, and 20 Gy using a Cesium-137 source (Gamma Cell, Atomic energy Canada). Heat shock was conducted by incubating the cell culture plates at 45 °C for 30 min and

then at 37 °C for the indicated time points. Chemical inhibitors and other reagents used are listed in supplementary Table 1.

**Cell permeabilization.** IMR90 cells were incubated in HBSS or treated with 7.5 Gy ionizing radiations in culture medium and then treated with Triton X-100 (Sigma, X100–1GA) or Digitonin (Sigma, D-5628) at the indicated concentrations and then harvested at the indicated time points for fixation and immunostaining. For live-cell imaging, IMR90 cell permeabilization was conducted in HBSS containing 0.03% Triton X-100 and 50 μg/ml propidium iodide.

**Cell transfection and lentivirus transduction.** HCT116 cells were transfected with expression plasmids using lipofectamine 2000. Two to three days after the transfection, cells were harvested for immunoblotting or treated with HBSS and used as indicated. For lentivirus production, expression constructs were transfected in HEK293FT cells, in combination with lentivirus packaging constructs and lentivirus particles were harvested several times post-transfection. IMR90 or HCT116 cells were infected once or multiple times with lentivirus suspension. Two days later, cells were selected with puromycin for 2 days and then used as indicated. siRNA oligonucleotides targeting various factors were purchased from Sigma-Aldrich. IMR90 cells were transfected using Lipofectamine RNAi max with 200 pmol of individual or pooled siRNAs. Three days post-transfection, cells were treated as indicated or directly harvested for immunoblotting or immunostaining. siRNA oligonucleotides used are listed in Supplementary Table 2.

For the electroporation of IMR90 cells, 2 million cells were resuspended in electroporation buffer (5 mM KCl; 15 mM MgCl$_2$; 120 mM Na$_2$HPO$_4$/NaH$_2$PO$_4$ pH 7.2; 25 mM sodium succinate; 25 mM manitol) and electroporation was done using X-100 program on Amaxa™ Nucleofector™ II Device (Lonza).

**Immunoblotting and antibodies.** Total cell extracts were prepared following cell lysis in 25 mM Tris pH 7.3 and 2% sodium dodecyl sulfate (SDS). Cell extracts were boiled at 95 °C for 10 min and then sonicated. Total protein concentration was determined using the bicinchoninic acid (BCA) protein quantification assay. The samples were diluted in Laemmli buffer. SDS-PAGE and the immunoblotting were done as we previously described[59]. The chemiluminescence band signals were acquired with a LAS-3000 LCD camera and analyzed with a MultiGauge software (Fuji, Stamford, CT, USA) or with Azurec600 chemiluminescence Imaging System. The antibodies and dilutions used are listed in Supplementary Table 3.

**Preparation of nuclear and cytoplasmic cell fractions.** IMR90 cells were incubated in HBSS and fractionated with a hypotonic lysis buffer as previously described, but with some modifications[60]. Briefly, IMR90 cells were washed once with cold PBS and then twice with hypotonic buffer (10 mM KCl, 10 mM Tris-HCl pH 7.3, 1.5 mM MgCl$_2$, 1 mM β-mercaptoethanol). The cells were scraped with the same hypotonic buffer containing 1 mM PMSF and protease inhibitors cocktail, and then left on ice for 5 min. The cells were then lysed using a dounce homogenizer and a fraction of the sample is collected as a total cell extract. The remaining samples are centrifuged for 15 min at 1,500 × g. The supernatant (cytoplasmic fraction) is harvested and mixed with 2% SDS. The pellet is washed with a hypotonic solution supplemented with 100 mM KCl and then centrifuged for 5 min at 1,500 × g. The supernatant is discarded and the pellet (nuclear fraction)

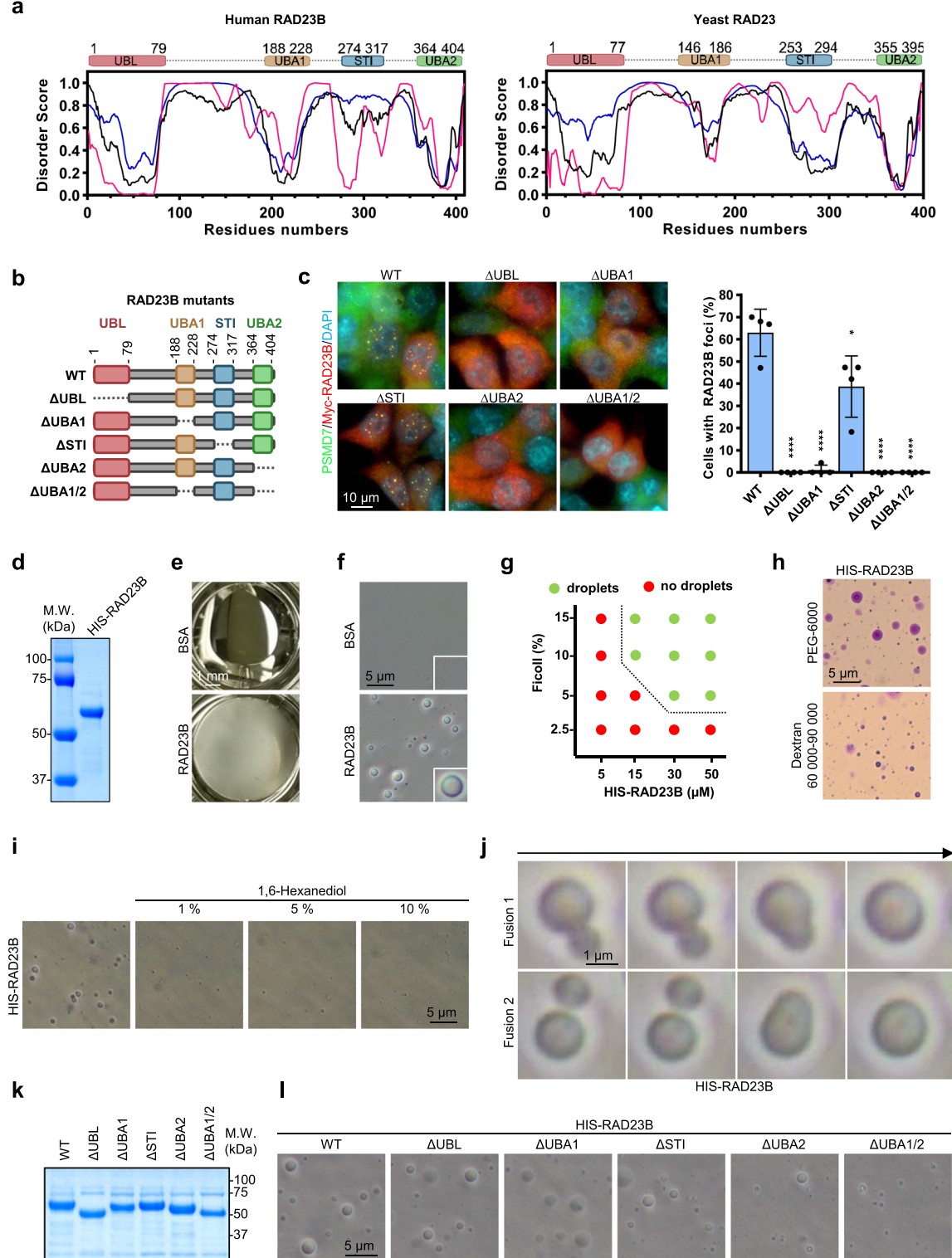

is resuspended in a volume of hypotonic buffer corresponding to that of the cytoplasmic fraction, and then mixed with 2% SDS. All samples were then used for immunoblotting. HCT116 cells were treated with HBSS and nuclear and cytoplasmic fractions were prepared by incubating the cells for 1 min in 10 mM Tris pH 7.3 containing 100 mM KCl, 1 mM β-mercaptoethanol, 1× anti-protease, 1 mM PMSF, and 0.1% NP-40. Cells were centrifuged at 3,500 × g for 1 min and the pellet fraction was washed once in detergent-free buffer. All samples were then used for immunoblotting.

**Colloidal gold-based immunodetection and transmission electronic microscopy.** This procedure was performed based on previous protocols[61], with the

following modifications. Cells were fixed for 30 min in 4% paraformaldehyde (PFA) in 0.1 M cacodylate buffer pH 7.2 (Tcaco), washed twice in Tcaco buffer and then once in PBS. Cells were permeabilized in PBS containing 0.2% de NP-40 for 10 min and non-specific sites were saturated with PBS containing 0.04% NP-40 and 10% FBS for 30 min (blocking buffer). Cells were then incubated for 3 hrs at RT with anti-PSMD4 primary antibody in blocking buffer. Cells were then washed 6 times, 5 min each, with the blocking buffer and incubated with anti-mouse IgG nanogold antibody (Nanoprobes, NY, USA), diluted 1/100 in the blocking buffer, followed by several washes in the blocking buffer. Coverslips were then incubated in 2% glutaraldehyde in PBS for 10 min followed by a 10 min incubation in silver enhancement solution (HQ Silver enhancement kit, Nanoprobes, NY, USA) at room temperature. Samples were post-fixed with 1% OsO4 in Tcaco buffer for

**Fig. 8 RAD23B mediates SIPAN formation. a** Analysis of domains and intrinsically disordered regions of RAD23 proteins. Prediction of order/disorder propensity of human RAD23B (left panel, uniprot #P54727) and yeast RAD23 (right panel, uniprot #P32628) based on their protein sequences. Disorder scores were calculated using PONDR-FIT (green), VSL2B (blue), and VLXT (magenta). **b, c** RAD23B lacking critical domains failed to assemble in SIPAN. **b** Schematic representation of RAD23B mutants. **c** Following lentiviral infection, IMR90 cells were incubated in HBSS for 6 hrs and harvested for immunostaining for Myc-RAD23B (anti-Myc) and PSMD7. Right, Estimation of the number of cells with SIPAN is indicated (representative from 4 independent experiments). **d**–**f** RAD23B undergoes LLPS in vitro. **e** Purified RAD23B in 50 mM HEPES pH 7.2 and 100 mM NaCl was mixed with Ficoll 400 and phase separation was visually observed, as RAD23B solution become turbid immediately after solution mixing (representative from 3 independent experiments). **f** RAD23B droplets were observed by bright-field microscopy. A portion of RAD23B and Ficoll 400 mixture was added on a microscope slide and a coverslip was applied on the solution near the edge of another coverslip to create space and allow liquid movement (representative from 3 independent experiments). **g** Effects of Ficoll and protein concentration on His-RAD23B droplet formation. Different concentrations of His-RAD23B were mixed with various concentration of Ficoll (representative from 3 independent experiments). **h** RAD23B was mixed with PEG 6000 or Dextran 60,000–90,000 and was added on a microscope slide and covered with a coverslip. Crystal violet was used to stain the droplets (representative from 3 independent experiments). **i** Inhibition of weak hydrophobic interactions alters RAD23B droplet formation. His-RAD23B was mixed with different concentrations of 1,6-hexanediol and analyzed for droplet formation (representative from 3 independent experiments). **j** RAD23B droplet fusion events during phase separation in vitro. Fusion events were detected by microscopy (representative from 3 independent experiments). **k** Coomassie showing purification of RAD23B protein and its corresponding mutants. **l** His-RAD23B mutants lacking critical domains were analyzed for droplet formation. Fewer droplets are observed with UBA1/2 mutant of His-RAD23B (representative from 3 independent experiments). Data in **c** represent mean ± SD. *P < 0.05; **P < 0.01; ***P < 0.001; ****P < 0.0001; ns: not significant; 2-sided unpaired Student's t-test is used. Source data are provided as a Source data file.

10 min, stained *en bloc* with 1% uranyl acetate for 5 min. Cells were dehydrated in graded series of ethanol and scrapped off the plates in ethanol and pelleted. The pellets were embedded in Epon[62]. Ultrathin sections of the samples were obtained using a Reichert Ultracut ultramicrotome and mounted on naked nickel grids. Sections were stained with lead citrate and examination was performed with a Philips CM100 transmission electron microscope. Electron micrographs were captured using an AMT XR80 digital camera (Advanced Microscopy Techniques, Corp. MA, USA).

**Immunofluorescence**. The immunostaining was conducted as previously described[60]. Briefly, culture medium was removed and cells were directly fixed in PBS containing 3 % PFA for 20 min. For antigen retrieval, the samples were incubated in sodium citrate Buffer (10 mM sodium citrate, 0.05% Tween 20, pH 6.0) and heated for 30 s in the microwave. The cells were then washed three times to remove sodium citrate buffer. Cells were permeabilized by incubation in PBS 0.5% Triton. Non-specific sites were blocked for 1 hr using PBS containing 0.1% Triton X-100 and 10% FBS. The coverslips were then incubated with primary antibodies for 3 hrs at room temperature or overnight at 4 °C. After three washes, cells were incubated with secondary anti-mouse Alexa Fluor® 594 (1/1,000), anti-mouse Alexa Fluor® 488 (1/1,000), anti-rabbit Alexa fluor® 488 (1/1,000) or anti-rabbit Alexa Fluor® 594 (1/1,000) antibodies for 1 hr. Nuclei were stained with 4′,6-diamidino-2-phenylindole (DAPI) during secondary antibodies incubation. Cell samples were mounted on a stage of an Olympus BX53 (Olympus Corp., Japan) upright microscope equipped with an Olympus UPlan SApo ×60/1.35 NA oil immersion objective. The corresponding fluorescence cubes (DAPI-1160B, GFP-3035C and Texas-4040C; Semrock Inc, USA) were used to efficiently reflect the excitation wavelengths and pass the emission wavelengths into the CCD camera detection channel. The images were acquired using a monochromatic Peltier cooled 1.4 megapixel CCD Olympus XM10 (Olympus Corp., Japan) CCD camera controlled by the Olympus CellSens software or using a DeltaVision Elite system (GE Healthcare) with z-stacking. Gamma, brightness, and contrast were adjusted on displayed images using the CellSens software. The collected EPI-fluorescence images were processed using WCIF-ImageJ program (NIH).

**EPI-fluorescence and bright field live-cell microscopy**. Cells stably expressing PSMB4-EGFP were grown on Mattek glass bottom petri dishes (Coverslip thickness No. 1.5, Mattek Corp., Ashland, MA). Cells were rinsed with PBS and then treated with HBSS with or without 100 μM chloroquine. The samples were then mounted on a motorized stage of a Zeiss AxioObserver.Z1/7 inverted microscope equipped with a live-cell incubator and a CO₂ module. Brightfield (DIC) and EPI-fluorescence images were collected using a Zeiss Plan-Apochromat ×63/1.4 NA oil immersion objective lens. GFP was excited with a CoolLED pE-300 lite LED light engine, and the corresponding fluorescence cube (49002-ET – EGFP/FITC/Cy2, Chroma, Bellows Falls, VT) was used to efficiently reflect the excitation wavelengths and pass the emission wavelengths into the CCD camera detection channel. DIC optics was utilized to enhance specimen detail of brightfield images. The images were acquired every 5 min using a monochromatic Zeiss AxioCam MRm R3 CCD camera (with 1× camera adapter) and Zen Blue v2.6 software. Multiple stage positions were collected using a motorized scanning stage. A combined focus strategy (via Zen's Tiling module assisted with a software autofocus) was utilized to maintain the focal plane over time. The acquisition settings were kept constant for all imaging experiments so that valid comparisons could be made between measurements from different datasets. Acquisition parameters were set within the linear range of the CCD camera detection. Z-series were displayed as maximum Z-projections via the Zen's Supplementary Depth of Focus image processing module.

**Automated quantification of SIPAN**. Images were analyzed using a custom Python 3.6 program. Briefly, images from cells grown in complete medium were first used to assess background foci intensity. Nuclei were segmented using DAPI staining channel images and images in the other channel (containing SIPAN) were processed using a Python implementation of a band pass filter algorithm (coded by K. Smith and M. Kilfoil) based on bpass.pro (http://www.physics.emory.edu/faculty/weeks/idl/kit/bpass.pro), which was originally developed by J. Crocker and D. Grier[63]. Images were then segmented using a local thresholding algorithm (from the Sci-Kit Python library) to identify regions of elevated signal intensity (i.e., foci). Pixel intensity values were extracted from these foci, and the average signal intensity and standard deviation was calculated. These values were used to assign a threshold intensity for subsequent analyses, which was set as average background intensity value + 2*SD. The number of foci per nuclei, as well as their intensity, was then calculated from all images (including those originating from cells grown in complete medium) using a similar Python code, although in this case, foci that were of intensity < than (background intensity value + 2*SD) or smaller than 3 by 3 pixels were discarded (to remove background noise).

**SIPAN counting and tracking**. Z-stack images were acquired at 0.26 μm intervals covering a range of 4–5 μm. Foci tracking was performed manually using ImageJ (NIH) and the foci mobility was corrected by any positional shifting due to the movement of the whole cell during the course of image recordings. The number of foci was evaluated using automatic particle counting option in ImageJ. Microsoft Excel and Kaleida Graph (Synergy) were used for data analysis and presentation and Student's t-test was used for statistical analysis. Mean-square displacement was calculated by the following equation as previously described[64].

$$MSD(\Delta t) = \frac{1}{n}\sum_{i=1}^{n} Di(\Delta t)^2 \qquad (1)$$

Where

$$Di(\Delta t) = \sqrt{((x_t^i - x_t^{GC}) - (x_{t-\Delta t}^i - x_{t-\Delta t}^{GC}))^2 + ((y_t^i - y_t^{GC}) - (y_{t-\Delta t}^i - y_{t-\Delta t}^{GC}))^2} \qquad (2)$$

For image presentations, 2D-maximum intensity projected images were generated using the ZEN blue software.

For the determination of the size and the average intensity of foci, the objects (foci) were identified automatically using MetaMorph (Molecular Devices). The size (area) of the identified objects and the integrated intensity of the identified objects was automatically measured. The foci average intensity was then calculated from their integrated intensity and size as below

$$Average\ intensity = \frac{Integrated\ intensity\ of\ foci}{Area\ of\ foci} \qquad (3)$$

**Fluorescence recovery after photobleaching (FRAP) experiments**. PSMB4-GFP expressing HCT116 cells were cultured in 35 mm Mattek chambers and foci formation was induced by incubation in HBSS for 6 hrs prior to the imaging experiments. HCT116 cells transiently expressing 53BP1-GFP were exposed to gamma irradiation for 4 hrs to induce the formation of DNA double-strand breaks/repair 53BP1 foci. HCT116 transiently expressing human histone GFP-H2A were

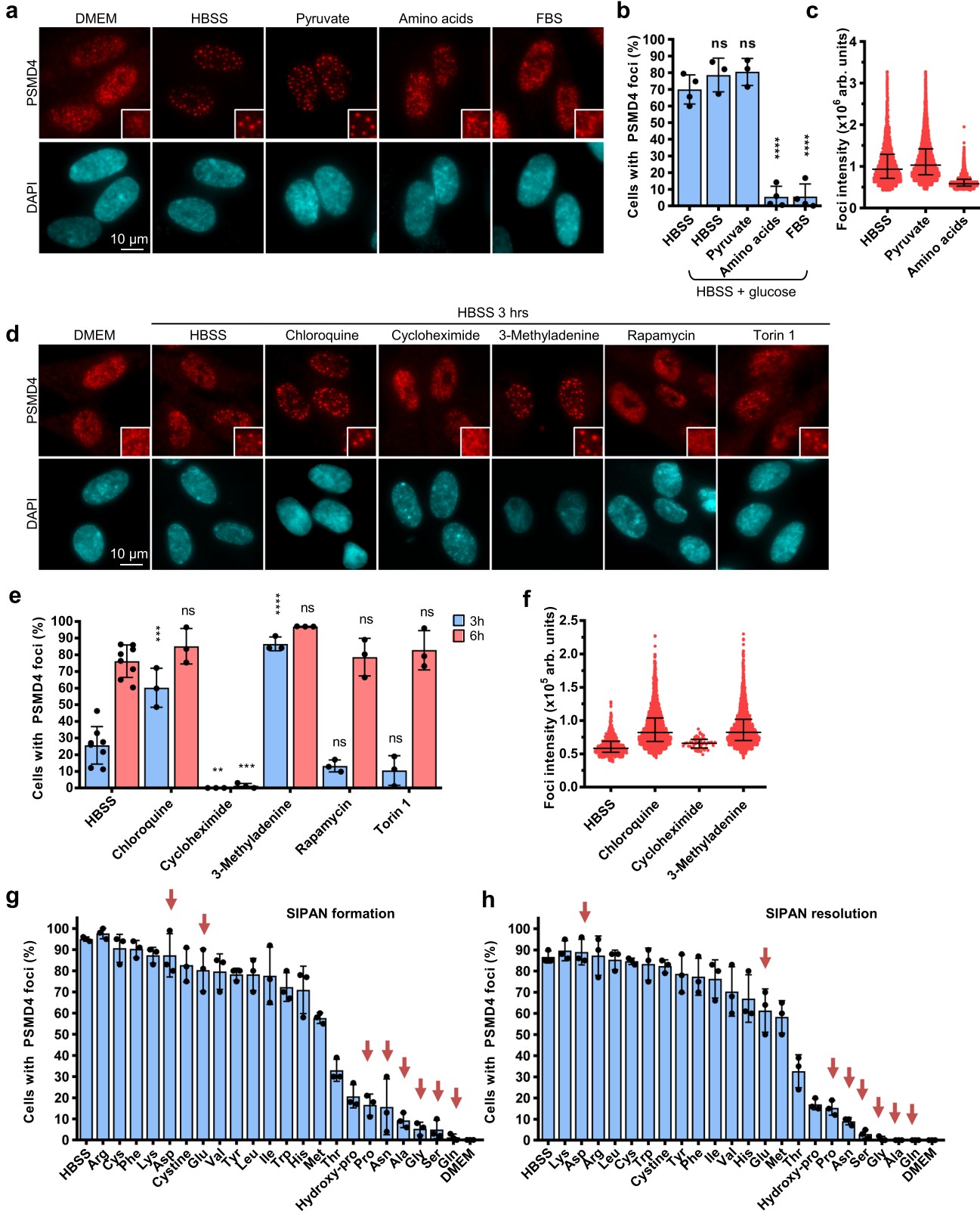

directly used. The samples were then mounted on a Prior ProScan III motorized stage of an Olympus IX81 inverted microscope. The FRAP experiments were performed on an Olympus FluoView FV1000 laser scanning confocal (CLSM) system equipped with spectral detectors. Regions of interest (ROI) from individual cells in the FluoView software were chosen (10 × 10 pixels with 1.03 μm/pixel) for the FRAP experiments. The 488 nm line of the Argon laser was used for both imaging (attenuated by 95%) and photobleaching (full 100% power output) of GFP in combination with a PLAPON ×60/1.4 NA OSC oil immersion lens. The DM405/

488 polychroic mirror was used to efficiently reflect the excitation wavelength and pass the emission wavelengths into the corresponding detection channel. Prebleach images (10 frames) were taken prior to FRAP activation for normalization of the data. Photobleaching of EGFP was generated by scanning the selected ROIs for 10 s (dwell time of 2 μs/pixel). Acquisition parameters (laser intensity, pixel dwell time, photomultiplier tube (PMT) voltage/gain, confocal pinhole aperture) were set within the linear range of the PMT detection. The acquired fluorescence recovery

**Fig. 9 Exhaustion of non-essential amino acid is responsible for SIPAN formation. a–c** Addition of amino acids, but not glucose or pyruvate, prevents SIPAN formation in IMR90 cells. Cells were incubated with various nutrients in HBSS solution and harvested after 6 hrs for immunostaining (**a**) and count (**b**) ($n = 3$ independent experiments). **c** Measured foci intensity is represented as violin plots (representative from 3 independent experiments). **d–f** Availability of amino acids regulates SIPAN formation. Cells were incubated with various inhibitors in HBSS solution and harvested after 3 hrs or 6 hrs for immunostaining (**d**) and count (**e**). Inhibition of autophagy by chloroquine accelerates SIPAN formation while inhibition of protein synthesis by cycloheximide inhibits SIPAN formation. Blocking mTOR pathway, with Rapamycin or Torin 1, does not affect SIPAN formation following nutrient deprivation ($n = 3$ independent experiments). **f** Measured foci intensity is represented as violin plot (representative from 3 independent experiments). **g** Non-essential amino acids (NEAA) completely prevented SIPAN formation. IMR90 cells were incubated with individual amino acids in HBSS solution and harvested after 6 hrs for immunostaining ($n = 3$ independent experiments). **h** NEAA promote SIPAN resolution. SIPAN formation is induced and then cells were replenished with fresh medium containing individual amino acids and harvested after 2 hrs for immunostaining ($n = 3$ independent experiments). Red arrows represent NEAA (**g, h**). Data in **b, c, e, f, g, h** represent mean ± SD. *$P < 0.05$; **$P < 0.01$; ***$P < 0.001$; ****$P < 0.0001$; ns: not significant; 2-sided unpaired Student's $t$-test used in (**b**) and 2-sided ANOVA in (**e**). Source data are provided as a Source data file.

---

curves were generated using the FluoView software. The values were obtained from the analysis of three independent experiments

**MTT assay.** IMR90 and HDLF cells were plated in 24-well culture plates and incubated in HBSS with treatments as indicated in figure legends. For siRNA experiment, the same number of cells (80,000 cells) were plated and treated with HBSS for 48 hrs for IMR90 or 24 hrs for HDLF. Medium was removed and replaced with complete medium containing 200 μg/ml of MTT (3-(4,5-dimethyl-thiazol-2-yl)-2,5-diphenyltetrazolium bromide, Bioshop, MTT222.1). Cells were then incubated at 37 °C for 3 hrs. Cells were washed once with PBS and DMSO was added to extract the formazan product. The absorbance was measured at 490 nm using a microplate reader (Biotek Instruments).

**FACS analysis.** Cells were washed with PBS and harvested by trypsinization. Cells were centrifuged, washed once with PBS and fixed with 75% cold ethanol overnight. Cells were centrifuged and resuspended in PBS containing 100 μg/ml RNase A and incubated at 37 °C for 30 min. To stain DNA, propidium iodide (P.I.) was added to the cell suspension at 50 μg/ml final concentration. DNA content of cells was acquired with BD (Becton Dickinson) FACS Calibur: 2 lasers, 4 detectors and analyzed using a FACScan flow cytometer fitted with CellQuestPro and FlowJo software (BD Biosciences). An example of gating strategy was provided as Supplementary Fig. 11.

**Colony forming assays.** IMR90 cells were plated in 35 mm culture dishes at the same density. Cells were treated with HBSS or chemicals as indicated. For the siRNA experiments, transfected IMR90 cells were plated in 6 cm dishes and incubated in HBSS for 48 hrs. Cells were then changed to normal culture medium and allowed to recover for 1 week. For all experiments, cells were fixed in PBS containing 3% PFA for 20 min. Then, cells were washed once with PBS and stained with 0.5% crystal violet for 30 min and then washed several times with water.

**RAD23B and RAD23 protein sequence analysis.** Analysis of domains and disordered regions of RAD23 proteins were conducted using PONDR-FIT algorithm[65], PONDR-VLXT and PONDR-VSL2[66]. The value of 0.5 in the Y-axis is considered as a threshold. Residues with a score above and below 0.5 are predicted to be disordered and ordered respectively. Hydrophobicity calculation was assessed using Kyte-Doolittle hydropathy algorithm[67]. The highest values indicate the hydrophobic amino acids along the sequence.

**Purification of human RAD23B.** Expression constructs for wild type His-RAD23B and its corresponding mutants were transformed into BL21-CodonPlus-RIL competent cells. Cells were grown at 37 °C and then treated with 400 μM Isopropyl β-d-1-thiogalactopyranoside (IPTG) to induce RAD23B protein production. Then, cells were harvested and centrifuged at $1,000 \times g$ for 15 min at 4 °C. The bacteria were washed with cold PBS, centrifuged and frozen on dry ice as pellets. The cell pellets were lysed in 50 mM Tris-HCl pH 8.0, 500 mM NaCl, 3 mM β-mercaptoethanol, 1 mM PMSF and 1× protease inhibitors (Sigma-Aldrich). The bacteria suspensions were sonicated and the resulting lysates centrifuged at $27,000 \times g$ for 20 min. Supernatants were incubated with Ni-NTA Agarose resin (Invitrogen, R901-15) overnight at 4 °C. Then, the resin was washed 5 times with 20 volumes of 50 mM Tris-HCl pH 8.0, 500 mM NaCl, 3 mM β-mercaptoethanol, 1 mM PSMF, 1× protease inhibitors, 20 mM imidazole and transferred into a Bio-Spin Disposable Chromatography column (Bio Rad, 731-1550). Proteins were eluted 5 times with 50 mM Tris-HCl pH 8.0, 500 mM NaCl and 200 mM imidazole, 3 mM β-mercaptoethanol and 1 mM EDTA and proteins were used subsequently for phase separation or temporarily stored at 4 °C.

For the production of RAD23B without His-tag, the elution of His-TEV-RAD23B was dialysed in 20 mM Tris-HCl pH 7.5 for 1 hr at 4 °C. The His tag was cleaved using TEV protease (New England Biolabs, P8112S) according to the manufacturer's protocol. The protein samples were incubated again with Ni-NTA Agarose resin (QIAGEN) for 5 hrs at 4 °C and then transferred into a Bio-Spin

Disposable Chromatography column. The elution was collected and the beads were washed three times with three volumes of the wash buffer. Proteins were immediately used for phase separation or temporarily stored at 4 °C.

**RAD23B liquid phase separation and droplet fusion in vitro.** His-RAD23B protein was concentrated 5 times and the elution buffer was changed with a phase separation buffer containing 50 mM HEPES pH 7.2 and 100 mM NaCl using Amicon Ultra-0.5 mL Centrifugal Filters 10 KDa (Millipore, UFC501024). For the in vitro assay, an equal volume of His-RAD23B and Ficoll 400 (Sigma, F4375, 300 mg/ml prepared in 50 mM HEPES pH 7.2 and 100 mM NaCl buffer) were mixed at room temperature to initiate droplet formation. The droplets were deposited on a glass slide in a chamber formed of two overlapping coverslips to allow liquid movements to observe fusion events. PEG 6000 (Alfa Aesar, A17541) or Dextran 60,000–90,000 (ICN, 101513) crowding agents prepared in 50 mM HEPES pH 7.2 and 100 mM NaCl buffer were used at 10 % or 5 % final concentration, respectively. Crystal violet (0.5% w/v) prepared in 25% methanol is occasionally used to stain droplets (1 μl of crystal violet for 20 μl of droplets solution).

**Yeast strains and growth conditions.** Yeast cells were generated and propagated using standard yeast genetics methods. The genotype of the yeast strains used in this study are BY4741 MATa *his3Δ1 leu2Δ0 met15Δ0 ura3Δ0 RPN5-GFP::HIS3MX* (from Life Technologies Yeast GFP collection, catalog #: 95702) and BY4741 MATa *his3Δ1 leu2Δ0 met15Δ0 ura3Δ0 RPN5-GFP::HIS3MX rad23Δ::KanMX* (generated for this study using standard yeast genetics methods). For carbon starvation, cells were grown to saturation in synthetic complete medium containing 2 % glucose. Cells were diluted in the same medium and allowed to grow to the exponential phase overnight. Cells were then washed once in synthetic complete medium without glucose and resuspended in the same medium at a density of 0.1 OD (630 nm). Cells were incubated in this medium for 24 hrs before harvest and fluorescence microscopy observations as described[68]. Briefly, yeast cells were fixed for 15 min in 0.1 M of potassium phosphate buffer (pH 6.4) containing 3.7% formaldehyde. Fixed cells are collected by centrifugation at $1,000 \times g$ for 2 min, resuspended in 50 mM Tris pH 7.5 and stored at 4 °C. Prior microscopy analysis, fixed cells were layered on a concanavalin A-coated 15-well slides and incubated with the antifade mounting medium containing DAPI (VECTASHIELD®). Images were taken by fluorescence microscopy using a ×60.

**Measurement of osmolality.** Osmolality of conditioned HBSS recovered from treated cells was measured with micro-osmometer Fiske® 210 (Advanced Instruments Inc.) according to the manufacturer recommendations. Calibration was done using calibration standard (Fiske 50 mOsm/kg #00A005 and 850 mOsm/kg #00A085).

**Oncogenic transformation and tumor generation.** Primary IMR90 fibroblasts were transformed following transduction by *MDM2*, *E1A* and *RAS^{V12G}* by retroviral infection as previously established[69]. Transformed cells were trypsinized, counted and then resuspended in culture media supplemented with Matrigel. $3 \times 10^6$ cells were subcutaneously injected (0.1 ml) using a 21-gauge needle in the right flank of 6-week aged athymic nude mice (JAX002019). All animal studies were approved by the Animal Care Committee of the research center of the Maisonneuve Rosemont Hospital and in agreement with the Canadian Council on Animal Care guidelines.

Tumor size was determined by measuring the length and width of the tumor using a caliper. At the end of the experiment, animals were euthanized with $CO_2$/ isoflurane influx. Tumors were collected 4–6 weeks post-injection to isolate and culture tumor cells. For cell isolation, tumors are minced into 1–2 mm pieces using a scalpel. Tumor pieces were incubated at 37 °C for 30 min in digestion media composed of DMEM/F12 media (Wisent), 0.5 ml collagenase (Bioshop), 100 μg/ml DNase (NEB), 2 % FBS, 1/100 Penicillin/Streptomycin. After incubation, tumor pieces were digested in prewarmed trypsin for 3–5 min and then with complete F12 media supplemented with DNase. Cell mixture was filtered using 40 μm cell strainer and isolated cells were cultured for subsequent assays.

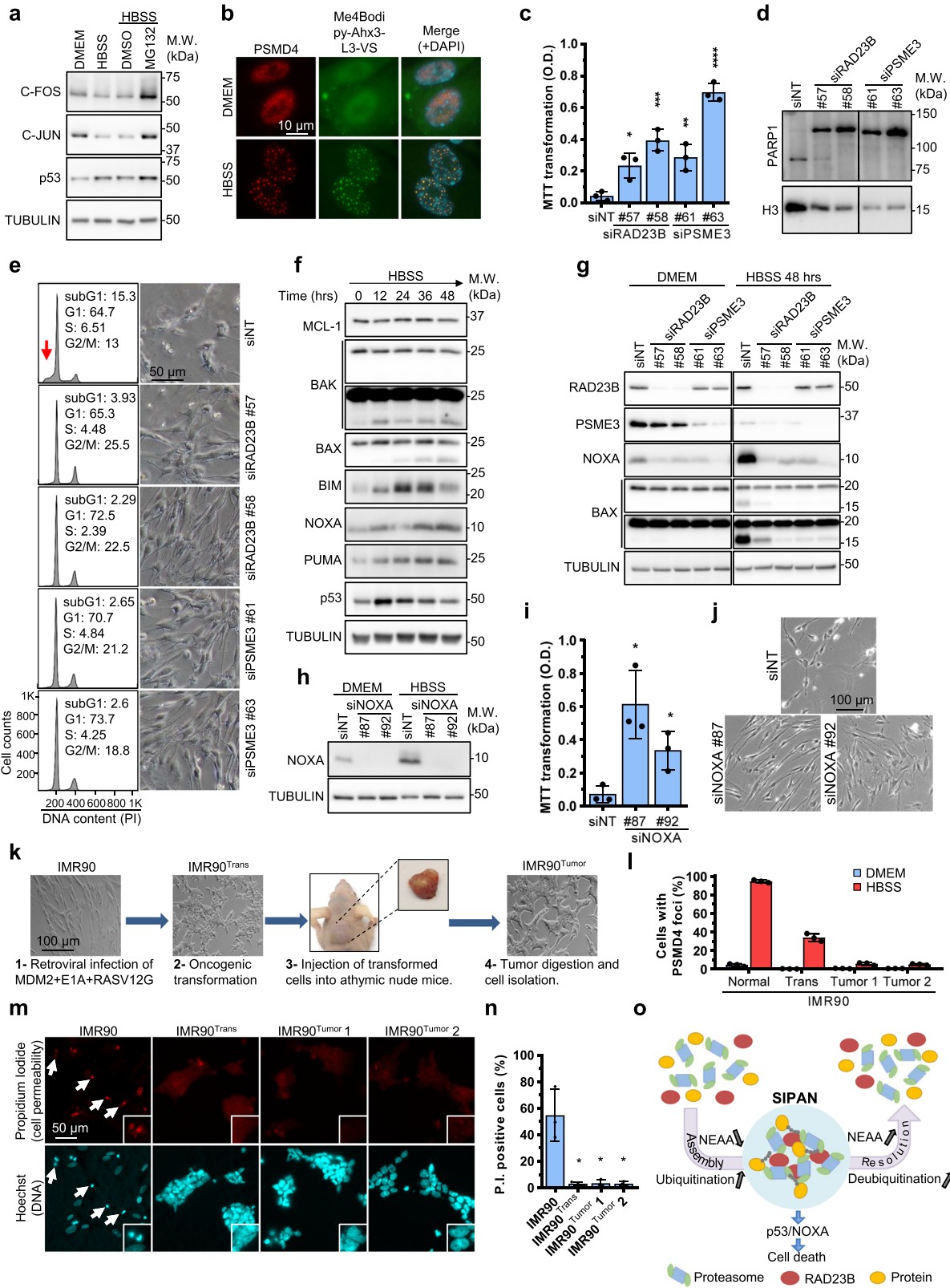

**Analysis of cell death using propidium iodide/Hoechst 33342**. After 96 hrs incubation with HBSS, IMR90, and IMR90^tumor cells were incubated for 1 hr with 1 μg/ml propidium iodide and 1 μg/ml Hoechst 33342. Cells were washed, fixed with PBS containing 3% PFA and analyzed by fluorescence microscopy as described above.

**Real-time quantitative PCR**. Expression levels of mRNAs encoding protein shuttling factors siRNA were analyzed as described before[70]. The mRNA levels

were normalized to PGK1 expression. The primers used are listed in Supplementary Table 4.

**Statistics and reproducibility**. Statistical analysis was carried out with Prism 8 (GraphPad). Data are represented as mean ± SD or median with interquartile. When applicable, appropriate statistical tests are used as described in figure legends. Briefly, unpaired 2-sided Student's *t*-test with or without Welch's correction or 2-sided

**Fig. 10 Inhibition of SIPAN formation is associated with apoptosis. a** Proteasome is active under nutrient starvation. IMR90 cells were incubated in HBSS for 8 hrs and then treated with MG132 for 1 hr and harvested for immunoblotting (representative from 3 independent experiments). **b** SIPAN are catalytic active. IMR90 cells were treated with HBSS for 6 hrs and then incubated with 1 μM of Me4Bodipy-Ahx3-L3-VS proteasomal activity probe for 1 hr. Cells were then harvested for immunostaining (representative from 3 independent experiments). **c–e** Inhibition of RAD23B or PSME3 result in increased cell survival following nutrient deprivation. Three days following siRNA transfection, IMR90 cells were incubated in HBSS for 48 hrs and harvested for MTT viability assay (**c**), Western blotting (**d**), phase contrast imaging or FACS analysis (**e**) ($n = 2$ independent experiments). Red arrow in (**e**) represents subG1 apoptotic cell population. **f** Induction of pro-apoptotic factors during HBSS treatment. IMR90 cells were treated with HBSS and harvested at the indicated times for western blotting (representative from 3 independent experiments). **g** Depletion of RAD23B or PSME3 cells protects from cell death induced by nutrient starvation. IMR90 cells without RAD23B or PSME3 were treated with HBSS for 48 hrs and harvested for immunoblotting for pro-apoptotic proteins (representative from 3 independent experiments). **h–j** Depletion of NOXA protects from cell death induced by nutrient deprivation. IMR90 were depleted of NOXA and treated with HBSS for 48 hrs and harvested for western blotting (**h**), MTT assay (**i**) and phase contrast (**j**) (representative from 3 independent experiments). **k** Schematic representation of the procedure for the generation of IMR90 cells transformed or derived from tumors. **l** Primary, transformed or tumoral IMR90 cells were treated with HBSS for 8 hrs and harvested for immunostaining for PSMD4 (representative from 3 independent experiments). **m, n** Cells were treated with HBSS for 96 hrs and stained with Hoechst (DNA) and propidium iodide for imaging (**m**) and counting of dead cells (**n**) ($n = 3$ independent experiments). **o** Model of SIPAN formation and function. Amino acids exhaustion promotes foci formation in the nucleus with RAD23B as a driver. SIPAN formation is associated with cell death. Data in **c**, **i**, **l**, **n** represent the mean ± SD. *$P < 0.05$; **$P < 0.01$; ***$P < 0.001$; ****$P < 0.0001$; ns: not significant; 2-sided unpaired Student's $t$-test used in (**c**, **n**). Source data are provided as a Source data file.

Mann–Whitney test was used to compare two groups. For multiple groups, Sidak's multiple comparisons for one-way ANOVA or Kruskal–Wallis test with Dunn's multiple comparisons were used. All experiments were done at least three times independently unless indicated if different. All $p$-value are present in Source data file.

**Reporting summary**. Further information on research design is available in the Nature Research Reporting Summary linked to this article.

## Data availability

There are no applicable accession codes, unique identifiers, or datasets that could be publicly available. Source data are provided in the Supplementary Information/Source data file. Additional data that support the findings of this study are available from the corresponding author (El Bachir Affar) upon reasonable request. All unique materials are readily available from the authors. Source data are provided with this paper.

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

## Acknowledgements

We thank Diana Adjaoud for technical assistance, Dr. Diane Gingras for help with electronic microscopy and Dr. Alain Verreault for yeast strains. This work was supported by grants from the Canadian Institutes of Health Research (CIHR) (MOP159539) to E.B.A., CIHR (GER-163050) to K.B. and F.A.M., the Natural Sciences and Engineering Research Council of Canada (NSERC) (DG 2015–2021) and The Cancer Research Society (2015–2017) to E.B.A., NSERC (DG 2019–2024) to H.W., CIHR (MOP126009) to E.M., the CIHR (Foundation grant to J.-Y.M.). E.B.A., L.H., B.L., and H.W are scholars of the Fonds de la Recherche du Québec-Santé (FRQ-S). J.-Y.M. is a FRQS Chair in genome stability. F.A.M. holds the Canada Research Chair In Epigenetics of Aging and Cancer. K.B. is the recipient of a Cole Foundation post-doctoral fellowship. N.K. and O.A had PhD scholarships from the FRQ-S and the Cole Foundation. L.M. had a PhD scholarship from the FRQ-S.

## Author contributions

M.U. designed, performed, analyzed experiments, performed statistics, and contributed to manuscript writing N.S.N. performed electronic microscopy and experiments related to treatments with individual amino acids and stress-inducing agents. N.M. made the original observations. H.B. performed cell fractionation and testing on cell types. L.M. performed qPCR experiments. S.D. conducted protein sequence analysis and helped with critical suggestions. M.S. helped with microscopy experiments. C.V., P.S., and B.L. helped with in vivo experiments. L.H. helped with metabolism analysis. M.P. and B.H.K. performed video analysis for the characterisation of foci dynamics. C.M., D.A., and M.V. performed experiments. N.D.L. and H.W. helped with foci count and intensity analysis. A.D. and E.M. performed flow cytometry experiments. K.B., O.A., and F.A.M. performed cell transformation and O.A. generated transformed cells and conducted tumor xenograft experiments. H.W. and R.T. performed yeast experiments. E.B.A. designed, performed experiments, analyzed data, and wrote the manuscript. D.R. and J.Y.M. expressed, purified, and performed cleavage of His-tag of RAD23B for in vitro assay. All authors contributed on concepts and data analysis and approved the final manuscript.

## Competing interests

The authors declare no competing interests.
