## [Peer Review File · Nature Communications]

Starvation-induced proteasome assemblies in the nucleus link amino acid supply to apoptosisReviewers' comments:

Reviewer #1 (Remarks to the Author):

The proteasome plays a pivotal role in cellular protein quality systems by degrading injured harmful proteins, thereby proteasome regulations are critically important for the survival of stressed cells. However, the spatiotemporal regulation of proteasomes, especially upon stress conditions, is not fully understood. In this study, Uriarte et al. discovered that the formation of novel proteasome-positive nuclear foci (termed SIPAN by authors) that have liquid droplets-like properties upon amino acid deprivation. Also, the authors showed that SIPAN colocalizes with ubiquitin conjugates and the formation of SIPAN requires RAD23B, a proteasome shuttling factor, and PSME3, an alternative proteasome activator in the nuclear proteasome. Interestingly, in vitro experiments showed recombinant RAD23B itself can form condensates by liquid-liquid phase separation (LLPS). As one of the biological consequences of SIPAN formation, it seems that SIPAN facilitates cell death induction upon amino acid depletion. The findings are novel and will be of wide interest for those in the fields of protein degradation as well as LLPS. However, there are several technical concerns and unanswered questions. First, the LLPS of RAD23B may be an artifact of using a high concentration of crowding reagents (Fig. 4), and may mislead the interpretations of other results. Second, although RAD23B and PSME3 were identified, the current manuscript lacks mechanistic insight for the SIPAN formation. The authors should address these main issues to make this suitable for publication in Nature Communications.

Major points:

1. I have concerns about the in vitro LLPS of RAD23B (Fig. 4). First, the condition of the experiment so limited and the concentration of crowding reagents is very high. Perhaps, 15% Ficoll would induce LLPS of most proteins. Can you make phase diagrams showing how easily RAD23B can form droplets, e.g. concentrations of RAD23B vs crowding reagent? Second, although His6 tag is small, His tag may affect chemical features of the protein of interest in many cases. So, His tags would be removed prior to any in vitro LLPS experiment. Third, domain analysis of RAD23B is also necessary for in vitro LLPS as in Fig. 2. Also, given that SIPAN includes ubiquitylated proteins and that the UBA domains are required for SIPAN formation, I wonder if ubiquitin conjugates are required for the phase separation. Can the authors test co-phase separation of K48-linked ubiquitin chain and RAD23B.

2. The study lacks mechanistic insights for SIPAN formation. Amino acid depletion induces LLPS of proteasomes in a RAD23B-dependent manner, but what is the direct trigger of the formation of SIPAN? In my view, it is much likely that ubiquitin chains are also involved in the SIPAN formation. To clarify this, the authors need to test the effect of TAK-243, a highly effective UAE1 inhibitor, on the SIPAN formation. If the TAK-243 inhibits the SIPAN formation or induces the SIPAN dissolution, the authors need to investigate the possible ubiquitylated substrates that trigger the LLPS of the proteasome. Although there is no apparent change in cellular ubiquitylation levels, amino acid depletion might induce

ubiquitylation of certain nuclear substrates. Another possibility is the ubiquitylation of proteasome itself as reported by Cohen-Kaplan et al (PNAS 113, E7490-E7499, 2016). Ideally, the authors should perform an MS-based ubiquitylome analysis between normal and amino-acid depleted conditions.

3. SIPAN facilitates apoptosis induced by amino acid depletion, but its molecular mechanism was not described. Is apoptosis induction a consequence of the sequestration of the proteasome? Or is SIPAN a site for degradation of apoptosis-related factors? The authors showed a link between SIPAN and apoptosis, but it is too descriptive and lacks the physiological function of SIPAN.

4. The authors showed that PSME3 is a component of SIPAN required for the SIPAN formation. PSME3 is a ubiquitin-independent activator of the CP, and the authors are discussing ubiquitin-independent proteasomal degradation in SIPAN. However, this alternative activator can form the RP-CP- PSME3 hybrid proteasome. To clarify this, the authors need to test whether the PSME3-positive foci can form in the RP subunit knockdown cells. With the same approach, it would be interesting to analyze the functional relationship between RAD23B and PSME3, because a single depletion of either RAD23B or PSME3 is sufficient to suppress apoptosis. These analyses would add the formation mechanism and function of SIPAN.

Minor points:

1. 1,6- hexanediol (1,6-HD) is a problematic molecule as it is known to lead to membrane integrity problems (Kroschwald et al, Matters 3, e201702000010, 2017). The authors use 10% 1,6-HD, a very high concentration, to disrupt SIPAN (Fig. 2k). The authors need to retest at 1% or lower concentration.

2. Can the authors show the diameter and circularity of SIPAN? This may be useful to think of LLPS.

3. Fig. 3c. Although bAP-15 is originally reported as an inhibitor of proteasomal deubiquitinases, b-AP15 inhibits degradation of proteasome substrates broadly. To evaluate correctly, the authors need to test general proteasome inhibitors and compare them with that of b-AP15. Also, I am curious if proteasome activity is required for SIPAN resolution upon amino acid replenishment.

4. Fig.3d. The authors showed a mini siRNA screening to pick up the essential factor for the SIPAN formation. Curiously, although RAD23A has the same domain structure with RAD23B, only the RAD23B depletion drastically reduced the SIPAN formation. Can you explain why RAD23B, but not RAD23A, is essential for the SIPAN formation? Also, the authors discuss "SHFM1 and UBQLN3 provide a link between ubiquitin signaling and...", but, SHFM1 (also known as DSS1 or SEM1) is a component of a lid subcomplex of the proteasome.

5. Fig.3h. WB is needed to confirm the proper expression of RAD23B mutants, because previous studies showed UBA deletion causes RAD23 degradation by the proteasome (Heinen et al, Nat Commun 2, 191, 2011). Also, the authors discuss that the RAD23B LLPS is induced by multivalent interactions between

UBL and two UBA domains, but it is unlikely because the UBL domain is used for the recruitment of the proteasome.

6. I cannot see some WB panels (Fig 5a and 5h). Please fix it.

Reviewer #2 (Remarks to the Author):

In this manuscript, Uriarte and coworkers investigate the formation and function of SIPAN, starvation-induced proteasome-containing nuclear bodies, in mammalian cells. SIPAN forms under nutrient deprivation, but not other cellular stresses, and requires components of both proteasomal catalytic and regulatory particles. SIPAN have characteristics of membraneless compartments that form via LLPS. They assemble and dissipate relatively quickly following introduction and removal of stress, appear spherical in shape, and undergo fusion and rapid exchange with the surroundings. Since the authors found conjugated Ub in SIPAN, they hypothesized that Ub-binding shuttle factors might affect SIPAN assembly and showed that knocking down Rad23B, among others to a lesser effect, inhibits SIPAN formation. Importantly, the UBL and UBA domains of Rad23B are essential for SIPAN assembly, suggesting that Rad23B's recruitment of ubiquitinated substrates to the proteasome leads to SIPAN (my interpretation, not the authors). Interestingly, Rad23B can undergo LLPS in vitro. The authors further found the presence of non-essential amino acids to inhibit SIPAN formation and correlated SIPAN with cell death from amino acid-induced starvation. These bodies operate differently than PSGs, proteasome storage granules. Still, the function of these SIPAN bodies is not entirely clear, and the state of the proteasome (active or not) remains to be examined in SIPAN bodies. This work is important for the LLPS field, suggesting that proteasome LLPS is important for cell function under certain stress conditions.

The authors did a tremendous amount of work with great controls. The data are good. The manuscript is well written, although the introduction seemed to have been rushed. The manuscript is suitable for a publication in Nature Comm after the comments below are addressed.

Comments

1. Please include scale bars in all figures, especially Fig 3i-j (and also a timescale for Fig 3j). Additionally, please include in vitro LLPS buffer conditions in Figure 3 legend.

2. The authors have some data that might imply that ubiquitinated substrates are needed for the formation of these SIPAN bodies. It would be great if they can show definitively that polyub chains are needed. How about inhibiting E1 or examining LLPS of purified Rad23B with purified K48 polyUb chains?

3. On pg. 7, the authors wrote: “This was confirmed by transmission electronic microscopy, which also revealed that nuclear proteasome foci are membrane-less protein condensates (Fig. 1j).” The term “membraneless condensates” provokes a dynamic, liquid-like compartment. A TEM image does not show this. The data in the remainder of the paper supports that these are indeed condensates, so I would suggest rewording the sentence to just highlight that these foci are membrane-less.

4. Extended Fig 6 e) was named d) in the legend.

5. The following work explores how different domains of UBQLN2 affect its LLPS and localization to stress granules and parallels the results from the domain deletion studies in this manuscript. “Dao, T.P., Kolaitis, R.-M., Kim, H.J., O’Donovan, K., Martyniak, B., Colicino, E., Hehnly, H., Taylor, J.P., Castañeda, C.A., 2018. Ubiquitin Modulates Liquid-Liquid Phase Separation of UBQLN2 via Disruption of Multivalent Interactions. *Mol. Cell* 69, 965-978.e6. <https://doi.org/10.1016/j.molcel.2018.02.004>”

6. This work, in which the authors briefly looked at the domains that drive p62 LLPS, is also relevant. “Sun, D., Wu, R., Zheng, J., Li, P., Yu, L., 2018. Polyubiquitin chain-induced p62 phase separation drives autophagic cargo segregation. *Cell Res.* 28, 405–415. <https://doi.org/10.1038/s41422-018-0017-7>”

7. siRNA results show knocking down other shuttle factors having an effect on the number of cells with foci, but Rad23A, which is quite similar to Rad23B, doesn’t seem to affect foci formation. What is a possible explanation? What about expression levels of the two in the nucleus? Also, could the authors include information regarding the efficiencies of siRNA knockdowns in their manuscript?

8. It’s interesting that knocking down Ubqln3, which is typically only found in testes, had an effect on formation of foci. Why did the authors look at this protein?

9. The author believe that hydrophobic interactions is responsible for Rad23B LLPS based on hexanediol data as well as the observation that the UBAs and UBL have hydrophobic patches. Recent literature has shown that the mechanism of how hexanediol works is unclear. As for the UBA and UBL domains, wouldn’t a more likely explanation as to why they are needed for formation of foci is that their interactions with ubiquitinated substrates and the proteasome, respectively, are needed to bring these

components together? Moreover, the UBL and UBA have been shown to interact with one another (Walters, K. J., Lech, P. J., Goh, A. M., Wang, Q. & Howley, P. M. (2003). DNA-repair protein hHR23a alters its protein structure upon binding proteasomal subunit S5a. Proc. Natl Acad. Sci. USA, 100, 12694–12699.). That can also increase the multivalency needed for LLPS.

10. The authors hypothesized that the presence of NEAA can lead to intermediates that might inhibit LLPS. Could it be that NEAA induce deubiquitination of certain substrates in SIPAN, which in turn causes SIPAN disassembly. Although this is not in the scope of this study, it would be interesting to find out if/what specific proteins undergo enhanced ubiquitination upon nutrient deprivation.

Reviewer #3 (Remarks to the Author):

Review

This is an interesting paper reporting on the observation of nuclear proteasome assemblies forming upon starvation, named SIPAN (Starvation induced Proteasome Assemblies in the Nucleus) by the authors. They show that these structures are dynamic, that non-essential amino-acids are important in this process. Importantly, they also show that ubiquitination is critical in their formation. Last but not least, they show that SIPAN formation doesn't occur in starved cells depleted of RAD23. The findings are interesting and worthy of publication but the manuscript needs serious revision before it can be publishable. The data is sound but the paper suffers from many overinterpretations which need to be rectified.

Major issues

1. The authors did a superficial job in citing the literature in the field. They need to revise and cite relevant work adequately.

a. In the introduction, they mention that protein degradation and amino acid recycling are fundamental and go on with autophagy, ignoring relevant work on proteasome and amino acid homeostasis. This statement needs revising "However, how proteasome-mediated protein degradation is coordinated with amino acid supply and protein synthesis has remained largely elusive". There is published work on this topic, particularly from the Bertolotti lab. The authors need to read and cite relevant literature.

b. They go on introducing proteasome assemblies, ignoring work of the Enenkel lab in this area, highly relevant to this manuscript.

c. They ignore work from Saeki on the nuclear proteasome assemblies.

2. In the text and in the discussion, they compare proteasome granules (yeast) to their mammalian SIPAN. I don't see why they believe the 2 granules are related but they should at least consider the possibility that they may be different structures. The whole discussion is on comparing similarities and differences between the yeast and mammalian structures whilst they may be unrelated. This needs to be revised.

3. The paper is difficult to read and the figures need to be improved. There are many figures that need to move from the extended figure section to the main to render the paper enjoyable. Why hiding interesting findings in the extended section? ((Extended Data Fig. 4a, top panel). Extended Data Fig. 4b). Extended Data Fig. 5a. (Extended Data Fig. 8a). Extended Data Fig. 9c

4. It is hard to find when the experiments are done with endogenous versus GFP tagged proteins. I would like to see a characterization of the properties of the overexpressed tagged proteasome subunit to appreciate the data. For example, they could show some immunostaining of the endogenous and compare with overexpressed GFP tagged PSMD4 to help assessing the relevance of the data acquired overexpressing GFP-PSMD4. Do we know if PSMD4 overexpression perturbs the proteasome assembly or function? This needs testing.

5. Why did they use IMR90 cells? Can they test different cell lines to assess the cell-specificity or broad relevance of the findings?

6. The whole idea of LLPS is massively overinterpreted. The physicochemical changes applied to cells have pleiotropic effects. The authors cannot tease apart direct effects -physicochemical changes- to their indirect consequences ie cellular adaptation to these changes. The authors conclusively report the formation of proteasome foci but I don't see why these should be LLPS unless they provide hard evidence for it. At present, there is none. The text needs to be revised. The findings are interesting. There is no need to claim that the proteasome phase separate when there is no evidence for this. The text needs to be revised accordingly.

7. "We expressed RAD23B by lentiviral transduction in IMR90 and found that this factor localizes in SIPAN (Fig. 3f)." The authors need to assess whether the endogenous RAD23B co-localize with SIPAN.

8. "UBL and UBA domains are required for SIPAN formation (Fig. 3h and Extended Data Fig. 9c)." This is a very important finding that gives mechanistic insights. However, the authors are so blinded by their LLPS theory that they ignore the mechanistic significance of this. The next need to be revised. UBA and UBL domain contribute function not just hydrophobicity !!!!

9. "Notably, live imaging indicates that RAD23B droplets undergo fusion events in vitro." The observation here is used to support the LLPS theory. This is a massive overinterpretation: RAD23 may phase separate in a test tube, it is a big leap of faith to imagine from that the proteasome may phase separate! Please revise.

10. "Conversely, preventing amino acid recycling by blocking autophagy with chloroquine or 3-methyladenine accelerates SIPAN formation (Fig. 4b, Video 11). Of note, blocking mTOR pathway with rapamycin or torin, does not affect SIPAN formation following nutrient deprivation (Fig. 4b)." These 2 findings are inconsistent. The authors should comment adequately on these discrepancies.

11. Figures need more clarity. The authors need to indicate the treatment and the additive /or washout in the figure.

12. "Following proteasome inhibition, we observed increased levels of several short-lived nuclear stress-associated transcription factors, including p53, c-fos and c-Jun, suggesting that SIPANs are not associated with proteasome inhibition (Fig. 5a). " This is non-sensical: This experiment just shows that proteasome inhibitors still work when proteasome foci are formed. One wants to know whether proteasome is functional in these granules. How about monitoring the degradation of short lived and proteasome reporters ? It would help adding mechanistic insights.

13. Panel 5h panel is missing.

14. "Based on these results altogether, we concluded that SIPAN formation is associated with amino acid starvation-induced cell death" This is not convincing at all: they apply treatments that kill cells, see proteasome granules and conclude that the granules are associated with cell death. I am also not comfortable with the following conclusions: RAD23B depletions protect cells. The link with apoptosis is overstated and needs revision.

15. In the discussion: "Genetic manipulation of signaling pathways that link proteasome LLPS and amino acid sensing is expected to provide additional insights into the role of SIPAN in human disease." This conclusion is unrealistic. If there was a way to genetically manipulate these proteasome granules, I would like to see this done in this paper! There are interesting findings in this manuscript. I would like to encourage the authors to connect their discussion to their interesting findings and discuss their potential significance and implications.

Rebuttal letter

Liquid phase separation of the mammalian nuclear proteasome links amino acid supply to apoptosis by
Uriarte et al.

Response to the Reviewer's comments

Reviewer #1

The proteasome plays a pivotal role in cellular protein quality systems by degrading injured harmful proteins, thereby proteasome regulations are critically important for the survival of stressed cells. However, the spatiotemporal regulation of proteasomes, especially upon stress conditions, is not fully understood. In this study, Uriarte et al. discovered that the formation of novel proteasome-positive nuclear foci (termed SIPAN by authors) that have liquid droplets-like properties upon amino acid deprivation. Also, the authors showed that SIPAN colocalizes with ubiquitin conjugates and the formation of SIPAN requires RAD23B, a proteasome shuttling factor, and PSME3, an alternative proteasome activator in the nuclear proteasome. Interestingly, in vitro experiments showed recombinant RAD23B itself can form condensates by liquid-liquid phase separation (LLPS). As one of the biological consequences of SIPAN formation, it seems that SIPAN facilitates cell death induction upon amino acid depletion. The findings are novel and will be of wide interest for those in the fields of protein degradation as well as LLPS. However, there are several technical concerns and unanswered questions.

Response R1-1: We thank the reviewer and appreciate his comment. We believe that the discovery of SIPAN will indeed open new avenues of research on LLPS as well as on cell signaling, especially on the link between amino acid metabolism, protein synthesis and degradation.

First, the LLPS of RAD23B may be an artifact of using a high concentration of crowding reagents (Figure 4), and may mislead the interpretations of other results. Second, although RAD23B and PSME3 were identified, the current manuscript lacks mechanistic insight for the SIPAN formation. The authors should address these main issues to make this suitable for publication in Nature Communications.

Response R1-2: We appreciate these comments. We have performed additional experiments. As detailed immediately below, we have now provided significant insights into LLPS in vitro, in vivo and on the mechanism of cell death induced by amino-acid deprivation.

Major points:

1. I have concerns about the in vitro LLPS of RAD23B (Figure 4). First, the condition of the experiment

so limited and the concentration of crowding reagents is very high. Perhaps, 15% Ficoll would induce LLPS of most proteins. Can you make phase diagrams showing how easily RAD23B can form droplets, e.g. concentrations of RAD23B vs crowding reagent?

Response R1-3: As suggested by the reviewer, we performed new LLPS experiments with increasing concentrations of Ficoll, as the crowding agent, as well as increasing the concentrations of purified RAD23B and made the phase separation diagram (**Fig. 8g**). We found that, as expected, phase separation is increased with the concentration of the crowding agent. Moreover, reducing Ficoll concentration by three times (at 5%), comparatively to what we previously used, can still result in RAD23B phase separation. We can also observe RAD23B phase separation using three times less RAD23B concentration, comparatively to what we previously used. We note that with the highest concentration of Ficoll used, BSA does not undergo LLPS at protein concentration similar to the highest concentration of RAD23B used. Moreover, when deleting critical domains of RAD23B, notably UBA2, highly reduced phase separation was observed, again arguing against a non-specific phase separation (**Fig. 8k,l**). We note that, although Yasuda et al (Nature volume 578, pages296–300(2020)) have used a different crowding agent and different conditions, they also observed that RAD23B undergo LLPS in vitro.

On the other hand, while it is difficult to make strict comparisons as LLPS experimental conditions differ between studies (pH, temperature, nature of the protein, protein purification conditions, buffer conditions), we note that similar concentrations (as ours) of Ficoll have been used by others for comparable ranges of protein concentrations (e.g., Dao et al., Molecular Cell, 69 (6), 965-978, 2018; Molliex et al., Cell 163 (1), 123-133, 2015; Mitrea et al, Nature Communications, 9(1):842, 2018; Lu et al., Nature Cell Biology 22, 453–464, 2020). Finally, we can also observe phase separation with PEG and Dextran at concentrations used by others (e.g., Lu et al., Nature Cell Biology 22, 453–464, 2020; Gu et al., PNAS, 117 (49) 31123-31133, 2020).

Second, although His6 tag is small, His tag may affect chemical features of the protein of interest in many cases. So, His tags would be removed prior to any in vitro LLPS experiment. Third, domain analysis of RAD23B is also necessary for in vitro LLPS as in Figure 2. Also, given that SIPAN includes ubiquitylated proteins and that the UBA domains are required for SIPAN formation, I wonder if ubiquitin conjugates are required for the phase separation. Can the authors test co-phase separation of K48-linked ubiquitin chain and RAD23B.

Response R1-4: We appreciate these excellent suggestions. We have produced and purified RAD23B protein with and without His tag and these undergo phase separation (**Suppl. Fig. 8e**). Yasuda et al (Nature volume 578, pages296–300(2020)) used RAD23B without tag and observed that this protein undergoes LLPS. Thus, in both studies, the tag is not a major determinant in RAD23B phase separation. We have also performed phase separation studies by conducting protein domain analysis of

RAD23B following purification of several deletion mutants (**Fig. 8k,l**). We found that deletion of UBL, UBA1 or STI does not impact RAD23B LLPS in vitro, while deletion of UBA2 or combined deletion of UBA1/UBA2 strongly impacts RAD23B LLPS. These results are overall consistent with those reported by Yasuda et al (Nature volume 578, pages296–300(2020)) showing that combined deletion of UBA1/UBA2 abolishes RAD23B-mediated LLPS in vitro.

However, in vivo, the situation is different as deletion of UBL, UBA1 or UBA2 strongly affects SIPAN formation. What explains why certain domains are required for SIPAN formation in the cells, but not for in vitro LLPS with purified RAD23B is currently unclear. However, it is important to note that the in vitro system is very minimal and does not contain the proteasome and other ingredients that might orchestrate SIPAN formation. It is also possible that in vitro LLPS with purified proteins could be somewhat promiscuous as pointed out by the reviewer. Nonetheless, overall, the results show that at least UBA2 domain is critical for RAD23B LLPS in vitro as observed in cells.

We also note that, in our hands, K48 ubiquitin chain reduces RAD23B LLPS in vitro (**see Fig. immediately below**). In our study, the buffer used as the solvent of ubiquitin has been changed multiple times to obtain ubiquitin chains in the same buffer used for LLPS. However, our result is different from those obtained by Yasuda et al. (Nature volume 578, pages296–300(2020)), as they found that K48 ubiquitin chains increases RAD23B LLPS in vitro. We also note that in another study on a protein highly similar to RAD23B, UBQLN2 (which contain UBL, STI1-like, and UBA domains), ubiquitin was also found to inhibit LLPS (Dao et al. Mol Cell, 69(6):965-978.e6, 2018). We prefer not to show these data at this point and we believe that these studies need a deep characterization, as the system used in vitro is minimal and does not contain the proteasome.

In contrast, ubiquitin is required for SIPAN formation in vivo as : **i)** inhibition of UAE1 abolishes SIPAN formation, **ii)** ubiquitin and conjugated ubiquitin are found in SIPAN, **iii)** MG132 and Bortezomib accelerate SIPAN formation, **iv)** DUB inhibition prevents SIPAN dissolution and **v)** RAD23B proteasome shuttling factor is required for SIPAN formation (**Figs. 7, 8 and Suppl. Figs. 7, 8**).

Clearly, additional studies on the mechanisms of SIPAN formation in vivo as well as on LLPS in vitro using all factors (including purified proteasome) are needed to definitively determine the exact role of ubiquitin in proteasome phase separation.

2. *The study lacks mechanistic insights for SIPAN formation. Amino acid depletion induces LLPS of proteasomes in a RAD23B-dependent manner, but what is the direct trigger of the formation of SIPAN? In my view, it is much likely that ubiquitin chains are also involved in the SIPAN formation. To clarify this, the authors need to test the effect of TAK-243, a highly effective UAE1 inhibitor, on the SIPAN formation. If the TAK-243 inhibits the SIPAN formation or induces the SIPAN dissolution, the authors need to*

Purified His-RAD23B (30 μ M final concentration) in 50 mM HEPES pH 7.2 and 100 mM NaCl was mixed with K48-UBIQUITIN chain tetramers (5 μ M final concentration) and then Ficoll 400 (10 % final concentration) was added. Phase separation was immediately observed using crystal violet staining and light microscopy.

investigate the possible ubiquitylated substrates that trigger the LLPS of the proteasome. Although there is no apparent change in cellular ubiquitylation levels, amino acid depletion might induce ubiquitylation of certain nuclear substrates. Another possibility is the ubiquitylation of proteasome itself as reported by Cohen-Kaplan et al (PNAS 113, E7490-E7499, 2016). Ideally, the authors should perform an MS-based ubiquitylome analysis between normal and amino-acid depleted conditions.

Response R1-5: we appreciate these insightful comments. As suggested, we performed experiments with the UAE1 inhibitor TAK243 and show that UAE1-mediated ubiquitination is required for SIPAN formation (**Fig. 7g,h**). In addition, we found that inhibition of UAE1 promotes SIPAN dissolution (**Fig. 7g,i**). Thus, ubiquitination seems to be continually required for SIPAN formation and maintenance. Consistently, inhibition of deubiquitination prevents SIPAN dissolution following addition of normal culture medium (**Fig. 7g,k**). Of note, treatment with the E1 inhibitor does not affect SIPAN dissolution following addition of normal culture medium (**Fig. 7g,j**). These results are overall consistent with our data showing the presence of conjugated ubiquitin in SIPAN. Moreover, while proteasome inhibitors accelerate SIPAN formation (**Fig. 7d-f**), they do not influence SIPAN dissolution (**Fig. 7g,k**).

Furthermore, consistent with our data, we also now found that K48-linked ubiquitination is detected in SIPAN (**Fig. 7c**).

We also conducted immunoprecipitation from HBSS treated cells using FK2 antibody and failed to detect an increased signal of ubiquitinated proteins during conditions of nutrient deprivation (**see Fig. immediately below**). Moreover, we did not detect changes in ubiquitination states after immunoprecipitation of the proteasome protein PSMD4 or by direct analysis of protein band shifts by western blotting. Thus, overall, the results suggest that ubiquitination is very discrete, and probably reflects states of active ubiquitination and deubiquitination of specific substrates at SIPAN. MS studies for ubiquitome can eventually help, but due to the discrete ubiquitination events, this will require large-scale cultures and purification of nuclear ubiquitinated proteins. So far, this has been very challenging with IMR90 primary cells, as SIPAN formation become compromised with cell expansion, probably due to alteration of cell metabolism in cells approaching senescence.

IMR90 cells were treated with HBSS for 8 h, and then harvested for denaturing immunoprecipitation of ubiquitinated proteins (IP FK2) or for native immunoprecipitation of proteasome (IP PSMD4). The IgG antibody was used as a negative control.

3. *SIPAN facilitates apoptosis induced by amino acid depletion, but its molecular mechanism was not described. Is apoptosis induction a consequence of the sequestration of the proteasome? Or is SIPAN a site for degradation of apoptosis-related factors? The authors showed a link between SIPAN and apoptosis, but it is too descriptive and lacks the physiological function of SIPAN.*

Response R1-6: We appreciate this excellent comment. We found that SIPAN are formed very early after nutrient deprivation and reach a maximum (80-90 % of cells with SIPAN) at 6-10 hours, time at which cells do not show signs of apoptosis or cell death (normal appearance of cells by phase contrast microscopy, absence of typical apoptotic PARP1 cleavage, absence of SubG1 cell population). Thus, SIPAN formation is associated with initial signaling that triggers cell death. Indeed, SIPAN formation precedes p53 induction and expression of p53-target genes (**Fig. 10f**). Moreover, inhibition of PSME3 or RAD23B, both of which localize at SIPAN, inhibits cell death. We also now show that the p53 target gene NOXA is induced during nutrient deprivation and that inhibition of RAD23B or PSME3 prevents the induction of this pro-apoptotic molecule. Thus, our data suggest that SIPAN formation is an early signaling event, associated with p53 mediated cell death. Of note, we did not observe p53 in SIPAN, we also did not observe the p53 regulators, MDM2, USP7 or USP10 in SIPAN. Thus, other protein substrates of the proteasome are likely to link SIPAN to the p53 pathway.

On the other hand, we also provide additional evidence that proteasome is active in SIPAN (**Fig. 10b**). It will be interesting to further determine which protein substrates are targeted by the proteasome and how their degradation results in p53/NOXA-mediated apoptosis.

4. *The authors showed that PSME3 is a component of SIPAN required for the SIPAN formation. PSME3 is a ubiquitin-independent activator of the CP, and the authors are discussing ubiquitin-independent proteasomal degradation in SIPAN. However, this alternative activator can form the RP-CP- PSME3 hybrid proteasome. To clarify this, the authors need to test whether the PSME3-positive foci can form in the RP subunit knockdown cells. With the same approach, it would be interesting to analyze the functional relationship between RAD23B and PSME3, because a single depletion of either RAD23B or PSME3 is sufficient to suppress apoptosis. These analyses would add the formation mechanism and function of SIPAN.*

Response R1-7: As suggested by the reviewer, we depleted RP or CP subunits and found that these are important for PSME3 foci formation (**Fig. 2d-f**). However, depletion of PSME3 had only a marginal effect on SIPAN formation (**Fig. 2d-f**). Our data suggest that both RAD23B and PSME3 proteins are important for the function of SIPAN as their depletion prevent NOXA induction and cell death (**Fig. 10 and Suppl. Fig. 10**). While it will be interesting to carefully investigate the relationship between PSME3 and 19S RP (and in relation with the 20S CP) in SIPAN formation and function, we believe that this could be a subject of further studies.

Minor points:

1. 1,6- hexanediol (1,6-HD) is a problematic molecule as it is known to lead to membrane integrity problems (Kroschwald et al, *Matters* 3, e201702000010, 2017). The authors use 10% 1,6-HD, a very high concentration, to disrupt SIPAN (Figure 2k). The authors need to retest at 1% or lower concentration.

Response R1-8: We agree with the reviewer and realize the limitations using high concentrations of 1,6-hexanediol (1,6-HD). As shown in the study indicated by the reviewer (Kroschwald et al, *Matters* 3, e201702000010, 2017) and others (e.g., Wheeler et al., *Elife* 2016 Sep 7;5:e18413. doi: 10.7554/eLife.18413), 1,6-HD can have detrimental effects on cells, particularly with high concentration and extended exposure times. Indeed, to control that membrane integrity is maintained, we have previously incubated cells with 1,6-HD, along with propidium iodide which rapidly enter the cells and stain the DNA. This allowed us to ensure that cell membrane integrity is not compromised during our imaging. We have also reduced the concentration of this compound and found that only 7.5-10% can strongly reduce SIPAN within 2 min of incubation only (**Fig. 6b**). For in vitro LLPS with purified RAD23B, we observed a strong reduction of phase separation with 1% of 1,6-HD (**Fig. 8i**). Of note comparable concentrations of 1,6-HD have been used for in vitro or in vivo LLPS by others (*Nature Cell Biology* volume 22, 453–464, 2020; Kroschwald et al, *Matters* 3, e201702000010, 2017, Boehning et al., *Nature Structural & Molecular Biology* volume 25, 833–840, 2018). Finally, we present ample evidence to demonstrate LLPS of proteasome in vivo (**Fig. 5, Fig. 6, Fig. 8, Suppl. Fig. 6 and Suppl. Fig. 8**).

2. Can the authors show the diameter and circularity of SIPAN? This may be useful to think of LLPS.

Response R1-9: Our data show that SIPAN are circular. We have now estimated the apparent diameter of SIPAN (see Material and methods). As shown in **Fig. 5d**, SIPAN area range between 0.1 and 0.7 μm^2 with an average 0.3 μm^2 . Their diameter is around 0.6 μm , which is comparable to that of DNA damage foci (Schumann et al., *Sci Rep*;8(1):2286. 2018). However, more precise measurements of foci size would require super resolution microscopy. On the other hand, we also analyzed SIPAN mobility and intensity after fusion and this corroborated our previous data on the highly dynamic nature of SIPAN (**Fig. 5e-j**).

3. Figure 3c. Although bAP-15 is originally reported as an inhibitor of proteasomal deubiquitinases, b-AP15 inhibits degradation of proteasome substrates broadly. To evaluate correctly, the authors need to test general proteasome inhibitors and compare them with that of b-AP15. Also, I am curious if proteasome activity is required for SIPAN resolution upon amino acid replenishment.

Response R1-10: We used the proteasome inhibitors MG132 and Bortezomib and found that proteasome inhibition does not prevent SIPAN resolution following medium replenishment (**Fig. 7g,k**). However, while we recognize that b-AP15 is a broad-spectrum inhibitor, this compound indeed inhibits

SIPAN dissolution (**Fig. 7g, k**). Thus, deubiquitination rather than proteasome activity is required for SIPAN dissolution.

4. *Figure 3d. The authors showed a mini siRNA screening to pick up the essential factor for the SIPAN formation. Curiously, although RAD23A has the same domain structure with RAD23B, only the RAD23B depletion drastically reduced the SIPAN formation. Can you explain why RAD23B, but not RAD23A, is essential for the SIPAN formation? Also, the authors discuss “SHFM1 and UBLN3 provide a link between ubiquitin signaling and...”, but, SHFM1 (also known as DSS1 or SEM1) is a component of a lid subcomplex of the proteasome.*

Response R1-11: We found that both RAD23A and RAD23B form foci following amino acid depletion. However, only RAD23B depletion dramatically reduces SIPAN formation (**Fig. 7I**). Similar results were obtained by Yasuda et al. (Nature volume 578, pages296–300(2020)), indicating that osmotic stress-mediated proteasome foci formation is inhibited following siRNA-mediated of RAD23B, but not RAD23A. In their study, they found that expression of RAD23A is almost 10 times lower than that of RAD23B which might account for the difference in their ability to promote foci formation. We also now show that in contrast to RAD23B depletion, RAD23A depletion does not protect from cell death induced by nutrient deprivation (**Suppl. Fig. 10e,f**). Thus RAD23B, but not RAD23A, appears to be the key determinant in SIPAN assembly and function.

In respect to DSS1, we agree with the reviewer that this is a component of the proteasome. However, we have included DSS1 in our screening as this protein has been also found in several other nuclear and chromatin-associated complexes (Trends Biochem Sci. 2016 May;41(5):446-459. DSS1/Sem1, a Multifunctional and Intrinsically Disordered Protein.). In any case, depletion of DSS1 significantly reduced SIPAN formation which is consistent with our data shown in **Fig. 2b,c**.

5. *Figure 3h. WB is needed to confirm the proper expression of RAD23B mutants, because previous studies showed UBA deletion causes RAD23 degradation by the proteasome (Heinen et al, Nat Commun 2, 191, 2011). Also, the authors discuss that the RAD23B LLPS is induced by multivalent interactions between UBL and two UBA domains, but it is unlikely because the UBL domain is used for the recruitment of the proteasome.*

Response R1-12: We have shown the immunoblotting data of RAD23B mutant in HEK293T cells, as these cells amplify expression plasmids with T antigen replication origin. This ensures that RAD23B proteins are indeed expressed and at the correct molecular weight. However, the high protein expression levels in this cell type often preclude the appreciation of protein stability defects. We now performed immunoblotting in HCT116 cells (the cell type used for immunostaining of RAD23 mutants) and observed that, indeed, UBA deletion, particularly UBA2, destabilizes the protein mutants, as previously observed by others (**Suppl. Fig. 8b**). However, this does not constitute an issue for evaluation

of RAD23B mutants ability to form SIPAN, as using immunostaining, we can consider cells that express adequate levels of mutants RAD23B, as they received higher amounts of the plasmids (**Suppl. Fig. 8c**). On the other hand, we agree with the reviewer about previous studies on the importance and known mechanism of action of RAD23B domains. However, at this point, we do not know how RAD23B interactions occur in SIPAN. It is possible that different modes of interaction involving RAD23B occur in SIPAN (relative to known modes of interactions) or perhaps a mixture of known and new interactions. Clearly, we know that protein domains are important, but more studies are needed to determine the composition of SIPAN and how RAD23B orchestrates SIPAN formation.

6. I cannot see some WB panels (Figures 5a and 5h). Please fix it.

Response R1-13: Our apologies for this oversight. These have been lost during file conversion. We now included the panels and ensured that it remains during conversion. These panels show that proteasome inhibition in cells (with preformed SIPAN) results in the accumulation of short-lived nuclear proteins (**Fig. 10a**) and that depletion of PSME3 prevents PARP1 cleavage associated with apoptosis (**Fig. 10d**).

Reviewer #2

In this manuscript, Uriarte and coworkers investigate the formation and function of SIPAN, starvation-induced proteasome-containing nuclear bodies, in mammalian cells. SIPAN forms under nutrient deprivation, but not other cellular stresses, and requires components of both proteasomal catalytic and regulatory particles. SIPAN have characteristics of membraneless compartments that form via LLPS. They assemble and dissipate relatively quickly following introduction and removal of stress, appear spherical in shape, and undergo fusion and rapid exchange with the surroundings. Since the authors found conjugated Ub in SIPAN, they hypothesized that Ub-binding shuttle factors might affect SIPAN assembly and showed that knocking down Rad23B, among others to a lesser effect, inhibits SIPAN formation. Importantly, the UBL and UBA domains of Rad23B are essential for SIPAN assembly, suggesting that Rad23B's recruitment of ubiquitinated substrates to the proteasome leads to SIPAN (my interpretation, not the authors). Interestingly, Rad23B can undergo LLPS in vitro. The authors further found the presence of non-essential amino acids to inhibit SIPAN formation and correlated SIPAN with cell death from amino acid-induced starvation. These bodies operate differently than PSGs, proteasome storage granules. Still, the function of these SIPAN bodies is not entirely clear, and the state of the proteasome (active or not) remains to be examined in SIPAN bodies. This work is important for the LLPS field, suggesting that proteasome LLPS is important for cell function under certain stress conditions.

The authors did a tremendous amount of work with great controls. The data are good. The manuscript is well written, although the introduction seemed to have been rushed. The manuscript is suitable for a publication in Nature Comm after the comments below are addressed.

Response R2-1: We thank the reviewer and appreciate his comments and enthusiasm about our study.

The state of the proteasome (active or not) remains to be examined in SIPAN bodies.

Response R2-2: We initially showed that following treatment of cells with preformed SIPAN (at time points while the large majority of cells form SIPAN), addition of proteasome inhibitors (which further enhance SIPAN formation) causes an accumulation of short-lived nuclear proteins (**Fig. 10a**). This suggested that proteasome is still active in the nucleus. We now used a proteasome activity probe and show that proteasome is still active in SIPAN (**Fig. 10b**). Our data are consistent with Yasuda et al (Nature volume 578, pages 296–300(2020)), showing that nuclear proteasome foci formed following osmotic stress are sites of active protein degradation.

1. Please include scale bars in all Figures, especially Figure 3i-j (and also a timescale for Figure 3j). Additionally, please include in vitro LLPS buffer conditions in Figure 3 legend.

Response R2-3: The scale bars are now included in **Fig. 8** and other **Figures**

2. The authors have some data that might imply that ubiquitinated substrates are needed for the formation of these SIPAN bodies. It would be great if they can show definitively that polyub chains are needed. How about inhibiting E1 or examining LLPS of purified Rad23B with purified K48 polyUb chains?

Response R2-4: We appreciate these excellent comments and suggestions, that has been also raised by Reviewer 1. We have conducted additional experiments with new interesting results. Please see our response above (**Responses R1-4 and R1-5**).

3. On pg. 7, the authors wrote: "This was confirmed by transmission electronic microscopy, which also revealed that nuclear proteasome foci are membrane-less protein condensates (Figure 1j)." The term "membraneless condensates" provokes a dynamic, liquid-like compartment. A TEM image does not show this. The data in the remainder of the paper supports that these are indeed condensates, so I would suggest rewording the sentence to just highlight that these foci are membrane-less.

Response R2-5. We agree with the reviewer. This has been changed. The sentence now reads: This was confirmed by transmission electronic microscopy, which also revealed that proteasome foci are membrane-less nuclear structures.

4. Extended Figure 6 e) was named d) in the legend.

Response R2-6: This has been changed

5. *The following work explores how different domains of UBQLN2 affect its LLPS and localization to stress granules and parallels the results from the domain deletion studies in this manuscript. “Dao, T.P., Kolaitis, R.-M., Kim, H.J., O’Donovan, K., Martyniak, B., Colicino, E., Hehnly, H., Taylor, J.P., Castañeda, C.A., 2018. Ubiquitin Modulates Liquid-Liquid Phase Separation of UBQLN2 via Disruption of Multivalent Interactions. Mol. Cell 69, 965–978.e6.*

<https://doi.org/10.1016/j.molcel.2018.02.004>;

Response R2-7: This work has been discussed as follows: It is noteworthy that another proteasome shuttling factor, UBQLN2, with similarities to RAD23B, also uses its multiple domains to drive LLPS of stress granules in the cytoplasm (Ref). However, in this case, ubiquitin was found to inhibit LLPS mediated by UBQLN2. Further studies are required to determine how ubiquitin promotes or inhibits LLPS.

6. *This work, in which the authors briefly looked at the domains that drive p62 LLPS, is also relevant. “Sun, D., Wu, R., Zheng, J., Li, P., Yu, L., 2018. Polyubiquitin chain-induced p62 phase separation drives autophagic cargo segregation. Cell Res. 28, 405–415. <https://doi.org/10.1038/s41422-018-0017-7>;*

Response R2-8: We thank the reviewer for the note. This has also been discussed as follows: Interestingly, a parallel can be made with K63-linked polyubiquitin chains which drives LLPS of the p62 scaffold protein resulting in autophagosome formation and selective autophagy.

7. *siRNA results show knocking down other shuttle factors having an effect on the number of cells with foci, but Rad23A, which is quite similar to Rad23B, doesn’t seem to affect foci formation. What is a possible explanation? What about expression levels of the two in the nucleus? Also, could the authors include information regarding the efficiencies of siRNA knockdowns in their manuscript?*

Response R2-9. We thank the reviewer for these excellent points. See response above (**Response R1-11**). We also now performed RT-qPCR to ensure the efficacy of siRNAs (**Suppl. Fig. 7e**).

8. *It’s interesting that knocking down Ubqln3, which is typically only found in testes, had an effect on formation of foci. Why did the authors look at this protein?*

Response R2-10. We have targeted several proteasome shuttling proteins that can link substrate ubiquitination to the proteasome. We agree that UBQLN3 is typically expressed in the testis. The effects observed on SIPAN might be associated with off-target effects of siRNAs. Our qRT-PCR in IMR90 cells indicate that this factor is indeed marginally expressed in IMR90 cells. Thus, we removed UBQLN3 from the screen. We also note that the key factor, RAD23B, was validated by multiple siRNAs that target

different regions of the mRNA (**Fig. 7l and m**).

9. The author believe that hydrophobic interactions is responsible for Rad23B LLPS based on hexanediol data as well as the observation that the UBAs and UBL have hydrophobic patches. Recent literature has shown that the mechanism of how hexanediol works is unclear. As for the UBA and UBL domains, wouldn't a more likely explanation as to why they are needed for formation of foci is that their interactions with ubiquitinated substrates and the proteasome, respectively, are needed to bring these components together? Moreover, the UBL and UBA have been shown to interact with one another (Walters, K. J., Lech, P. J., Goh, A. M., Wang, Q. & Howley, P. M. (2003). DNA-repair protein hHR23a alters its protein structure upon binding proteasomal subunit S5a. *Proc. Natl Acad. Sci. USA*, 100, 12694–12699.). That can also increase the multivalency needed for LLPS.

Response R2-11: We appreciate these comments. We do not know what is the exact mechanism of interactions that govern RAD23B-mediated LLPS. But we agree that multivalent interactions ensured by UBL and UBA domains might link ubiquitinated substrates to the proteasome and drive LLPS. Nonetheless, we cannot exclude that weak interactions ensured by other regions of RAD23B (possibly other proteins) might also be involved. A combination of domain-mediated interactions with weak interactions ensured by non-organized regions might act in a concerted manner to drive proteasome LLPS. We now included this point in the discussion section. Please see page 24.

10. The authors hypothesized that the presence of NEAA can lead to intermediates that might inhibit LLPS. Could it be that NEAA induce deubiquitination of certain substrates in SIPAN, which in turn causes SIPAN disassembly. Although this is not in the scope of this study, it would be interesting to find out if/what specific proteins undergo enhanced ubiquitination upon nutrient deprivation.

Response R2-12: We appreciate this excellent comment. It will be indeed very interesting to determine the exact link between NEAA and ubiquitination. This would be carefully investigated in the future.

Reviewer #3

This is an interesting paper reporting on the observation of nuclear proteasome assemblies forming upon starvation, named SIPAN (Starvation induced Proteasome Assemblies in the Nucleus) by the authors. They show that these structures are dynamic, that non-essential amino-acids are important in the this process. Importantly, they also show that ubiquitination is critical in their formation. Last but not least, they show that SIPAN formation doesn't occur in starved cells depleted of RAD23. The findings are interesting and worthy of publication but the manuscript needs serious revision before it can be

publishable. The data is sound but the paper suffers from many over interpretations which need to be rectified.

Response R3-1: We appreciate the comments. We believe that we have improved our manuscript by conducting additional studies. We improved the text and removed or corrected statements that can be perceived as over interpretation.

Major issues

1. The authors did a superficial job in citing the literature in the field. They need to revise and cite relevant work adequately.

a. In the introduction, they mention that protein degradation and amino acid recycling are fundamental and go on with autophagy, ignoring relevant work on proteasome and amino acid homeostasis. This statement needs revising “However, how proteasome-mediated protein degradation is coordinated with amino acid supply and protein synthesis has remained largely elusive”. There is published work on this topic, particularly from the Bertolotti lab. The authors need to read and cite relevant literature.

b. They go on introducing proteasome assemblies, ignoring work of the Enenkel lab in this area, highly relevant to this manuscript.

c. They ignore work from Saeki on the nuclear proteasome assemblies.

Response R3-2. Our apologies for the oversight. The manuscript has been initially written with the attempt to have a concise manuscript for potential publication in *Nature*. We initially wanted to focus on studies that addressed proteasome localization. However, we recognize that we could have better explained and cited the literature. As now outlined in the revised introduction, we expanded to better cover the relevant literature. We hope that the reviewer is now satisfied. In respect to the work reported by Yasuda et al. (Stress- and ubiquitylation-dependent phase separation of the proteasome, *Nature* volume 578, pages296–300(2020)), we emphasize that our study was under review when this paper was published and that we were invited to resubmit a revised form. Thus, we believe that this study (Yasuda et al.,) should be rather discussed (in results and discussion sections) rather than listed in the introduction.

2. In the text and in the discussion, they compare proteasome granules (yeast) to their mammalian SIPAN. I don't see why they believe the 2 granules are related but they should at least consider the possibility that they may be different structures. The whole discussion is on comparing similarities and differences between the yeast and mammalian structures whilst they may be unrelated. This needs to be revised.

Response R3-3: We made these comparisons as: i) proteasome is highly conserved. ii) PSG and SIPAN form following metabolic stress, iii) we show that RAD23 is required for the formation of PSG in yeast and that its human homolog, RAD23B, is also required for the formation of SIPAN in mammalian

cells. Moreover, in Arabidopsis, proteasome granules also form following metabolic stress (Elife 7, doi:10.7554/eLife.34532 (2018)). Perhaps, there is a common and highly conserved mechanism of proteasome adaptation following stress. However, we agree that we should be cautious that these structures might not be necessarily related. We mitigated our statements accordingly.

We stated in the discussion: Nonetheless, we cannot exclude at this point that PSG and SIPAN are unrelated structures in terms of composition, dynamics and functional significance.

3. The paper is difficult to read and the Figures need to be improved. There are many Figures that need to move from the extended Figure section to the main to render the paper enjoyable. Why hiding interesting findings in the extended section? ((Extended Data Figure 4a, top panel). Extended Data Figure 4b). Extended Data Figure 5a. (Extended Data Figure 8a). Extended Data Figure 9c.

Response R3-4: We acknowledge this concern. The manuscript has been originally submitted for consideration by *Nature*. We had to keep the manuscript very short and display only 5 Figures. The manuscript has been transferred to Nature Communications which allows more Figures. We now extended the text to add additional clarifications and also moved several panels of supplementary data as main Figures. We hope the reviewer is now satisfied.

4. It is hard to find when the experiments are done with endogenous versus GFP tagged proteins. I would like to see a characterization of the properties of the overexpressed tagged proteasome subunit to appreciate the data. For example, they could show some immunostaining of the endogenous and compare with overexpressed GFP tagged PSMD4 to help assessing the relevance of the data acquired overexpressing GFP-PSMD4. Do we know if PSMD4 overexpression perturbs the proteasome assembly or function? This needs testing.

Response R3-5: Our apologies that this was not very clear in the previous version. In fact, we have conducted immunostaining for endogenous expression and localization for many proteasome subunits and this for the large majority of our experiments (see **Fig. 1c; Fig. 2a-f; Fig. 3a, b; part of 3d, 3e; Fig. 4a-c; Fig. 5a-c, m, n, o; Fig. 6a; Fig. 7b, c, d, e, f, h, i, j, k, l, m, n; Part of Fig. 8c (PSMD7); Fig. 9a, b, c, d, e, f, g, h; Fig. 10, b, l**). Please, also see (**Suppl. Fig. 2a; Suppl. Fig. 3a, b, c; Suppl. Fig. 4; Suppl. Fig. 5a, c; Suppl. Fig. 6a, b, c, d, part of panel g (PML); Suppl. Fig. 7b, c, h; Part of Suppl. Fig. 8c (PSMD7); Suppl. Fig. 9a, c, d; Suppl. Fig. 10l**). When GFP-tagged forms are used, this was indicated as Protein X-GFP. We also note that IMR90 cells are stably expressing GFP with lentivirus vectors ensuring moderate expression of exogenous proteins. Finally, we did not perform function assays with GFP-fused proteasome subunits.

5. Why did they use IMR90 cells? Can they test different cell lines to assess the cell-specificity or broad relevance of the findings?

Response R3-6: We originally used IMR90 as these primary human fibroblasts are normal diploid cells. We initially wanted to avoid potentially confounding results that can be caused by oncogenic transformation. We also now used another human primary fibroblastic cell type, HDFL, and found similar results in term of SIPAN assembly (**Fig. 4 and Suppl. 4**) and impact of RAD23B and PSME3 depletion on cell death, following amino acid depletion (**Fig. 10 and Suppl. Fig. 10**). We also note that we tested nutrient deprivation and found that SIPAN form in many cell types (**Fig. 4 and Suppl. Fig. 4**). Interestingly, we found that certain cancer cells have reduced ability to form SIPAN upon nutrient deprivation (**Suppl. Fig. 4**). We also found that oncogenic transformation of IMR90 results in reduced ability of transformed cells to form SIPAN (**Fig. 10 and Suppl. Fig. 10**). This interesting data suggest that deregulation of SIPAN might be associated with tumor development, but this will require additional investigations.

6. The whole idea of LLPS is massively overinterpreted. The physicochemical changes applied to cells have pleiotropic effects. The authors cannot tease apart direct effects -physicochemical changes- to their indirect consequences ie cellular adaptation to these changes. The authors conclusively report the formation of proteasome foci but I don't see why these should be LLPS unless they provide hard evidence for it. At present, there is none. The text needs to be revised. The findings are interesting. There is no need to claim that the proteasome phase separate when there is no evidence for this. The text needs to be revised accordingly.

Response R3-7: We appreciate this concern, but we believe we provide with the new data ample evidence to indicate that the mammalian proteasome undergoes LLPS in the nucleus upon nutrient deprivation (**Figs. 3, 5, 6, 7, 8 and Suppl. Figs. 5, 6, 8**). We found that: **i)** SIPAN are spherical membrane-less structures and we measured their diameter and mobility. **ii)** SIPAN undergo fusion events and we quantified their fusion frequencies. **iii)** Our FRAP studies indicated that SIPAN rapidly exchange proteasome particles with the surrounding milieu. **iv)** SIPAN are induced by amino acid deprivation and quickly dissolve following amino acid replenishment. **v)** SIPAN are highly sensitive to physico-chemical conditions and can dissipate and recover in the original locations. **vi)** The major driver of SIPAN RAD23B in the cells undergo LLPS in vitro. Our data are consistent with the recent studies by Yasuda et al., Nature volume 578, pages296–300(2020)).

7. “We expressed RAD23B by lentiviral transduction in IMR90 and found that this factor localizes in SIPAN (Figure 3f). “ The authors need to assess whether the endogenous RAD23B co-localize with SIPAN.

Response R3-8: We performed experiment on endogenous RAD23B and this is assembled in SIPAN (**Fig. 7n**). Thus, endogenous or overexpressed RAD23B is assembled in SIPAN upon amino acid deprivation (**Fig. 7n and Suppl. Fig. 7g**).

8. *“UBL and UBA domains are required for SIPAN formation (Figure 3h and Extended Data Figure 9c).” This is a very important finding that gives mechanistic insights. However, the authors are so blinded by their LLPS theory that they ignore the mechanistic significance of this. The next need to be revised. UBA and UBL domain contribute function not just hydrophobicity !!!!*

Response R3-9: We agree with the reviewer. This has been changed. Please see page 24.

9. *“Notably, live imaging indicates that RAD23B droplets undergo fusion events in vitro. “ The observation here is used to support the LLPS theory. This is a massive overinterpretation: RAD23 may phase separate in a test tube, it is a big leap of faith to imagine from that the proteasome may phase separate! Please revise.*

Response R3-10: Please see our response above (**Response R3-7**).

10. *“Conversely, preventing amino acid recycling by blocking autophagy with chloroquine or 3-methyladenine accelerates SIPAN formation (Figure 4b, Video 11). Of note, blocking mTOR pathway with rapamycin or torin, does not affect SIPAN formation following nutrient deprivation (Figure 4b). “ These 2 findings are inconsistent. The authors should comment adequately on these discrepancies.*

Response R3-11: We now added more explanations. We have blocked mTOR with and without nutrient deprivation and found that the absence of mTOR activity does not induce the formation of SIPAN. We also note, during nutrient deprivation, mTOR activity is already greatly reduced as shown by phosphorylation of 4EBP1 in **Fig. 1b**. We believe that our data suggest that either SIPAN are independent of mTOR inhibition or that inhibition of this kinase might not be sufficient to induce SIPAN formation.

11. *Figures need more clarity. The authors need to indicate the treatment and the additive /or washout in the Figures*

Response R3-12: We added additional description to enhance clarity of Figure legends.

12. *“Following proteasome inhibition, we observed increased levels of several short-lived nuclear stress-associated transcription factors, including p53, C-FOS and C-JUN, suggesting that SIPANs are not associated with proteasome inhibition (Figure 5a). “This is non-sensical: This experiment just shows that proteasome inhibitors still work when proteasome foci are formed. One wants to know whether proteasome is functional in these granules. How about monitoring the degradation of short-lived and proteasome reporters? It would help adding mechanistic insights.*

Response R3-13: We appreciate this comment. First, we believe that the experiment could have been better explained. We initially showed that, following treatment of cells with already preformed SIPAN (time at which the large majority of cells form SIPAN), addition of proteasome inhibitors for a short period of time (1 hour) causes an accumulation of short-lived nuclear proteins (**Fig. 10a**). We also note that proteasome inhibitors further enhance SIPAN formation. This suggested that proteasome is still active in the nucleus. As described above, we now provide additional evidence using an activity probe that the proteasome is still active in SIPAN (**Fig. 10b**). Our data are consistent with the study by Yasuda et al. (Nature volume 578, pages296–300(2020)) showing that proteasome foci are sites of active protein degradation.

13. *Panel 5h panel is missing.*

Response R3-14: Our apologies. This panel was lost during file conversion. We now included this panel (**Fig. 10d**), which also shows that inhibition of PSME3 protects against cell death following amino acid deprivation.

14. *“Based on these results altogether, we concluded that SIPAN formation is associated with amino acid starvation-induced cell death” This is not convincing at all: they apply treatments that kill cells, see proteasome granules and conclude that the granules are associated with cell death. I am also not comfortable with the following conclusions: RAD23B depletions protect cells. The link with apoptosis is overstated and needs revision.*

Response R3-15: We appreciate this comment and agree that the link between SIPAN and cell death can be further consolidated. We initially determined that absence of amino acids causes SIPAN formation and cell death. In addition, absence of non-essential amino acids (NEAA) is a major determinant of SIPAN formation, relative to essential amino acids (EAA) (**Fig. 9 and Suppl. Fig. 9**); and replenishment of the cells with NEAA induces SIPAN dissipation and protects from cell death, relative to EAA (**Suppl. Fig. 9**). Moreover, cycloheximide which inhibits mRNA translation and amino acid utilization prevents SIPAN formation and protect against cell death, while chloroquine which prevents amino acid recycling increases SIPAN formation and cell death (**Fig. 9 and Suppl. Fig. 9**). Importantly, depletion of RAD23B or PSME3, both of which localize in SIPAN, inhibits cell death (**Fig. 10 and Suppl. Fig. 10**). In particular, RAD23B is required for SIPAN formation. In addition, we now show that the p53 tumor suppressor is induced in IMR90 cells, upon amino acid depletion, to promote the expression of its target genes. In particular, the p53 target gene NOXA plays an important role in promoting cell death following amino acid depletion. Moreover, depletion of RAD23B or PSME3 prevents NOXA induction, further consolidating the link between SIPAN-associated factors and cell death (**Fig. 10 and Suppl. Fig. 10**). Thus, overall, SIPAN appear to regulate the p53 pathway in response to amino acid deprivation. We also

analyzed several factors that regulate p53 including USP7, USP10 and MDM2, but no apparent fluorescence signal was observed in SIPAN for these factors. Finally, we conducted oncogenic cellular transformation and found that the ability of IMR90 cells to form SIPAN become greatly reduced in transformed cells and this is also accompanied by strongly reduced cell death in response to amino acid depletion. Clearly these studies further support the link between SIPAN and cell death. Nonetheless, further studies are needed to determine the mechanism by which SIPAN regulate cell death in response to amino acid availability, but this could be another study in its own right.

15. In the discussion: "Genetic manipulation of signaling pathways that link proteasome LLPS and amino acid sensing is expected to provide additional insights into the role of SIPAN in human disease." This conclusion is unrealistic. If there was a way to genetically manipulate these proteasome granules, I would like to see this done in this paper! There are interesting findings in this manuscript. I would like to encourage the authors to connect their discussion to their interesting findings and discuss their potential significance and implications

Response R3-16: We appreciate this comment. We meant by genetic manipulation that we could conduct loss-of-function of relevant genes and determine the impact on SIPAN formation or resolution. Indeed, we found that RAD23B and PSME3 localize in SIPAN. We also provided evidence that loss-of-function of PSME3 or RAD23B using siRNAs affects SIPAN assembly or function and inhibits cell death induced by amino acid depletion (**Figs. 7, 10**). We also now show that depletion of NOXA prevents cell death induced by amino acid depletion (**Fig. 10 and Suppl. Fig. 10**). Of note, depletion of RAD23B or PSME3 prevents NOXA upregulation upon amino acid deprivation. Moreover, our data with E1 and DUB inhibition also support the notion that signaling pathways are involved in the regulation SIPAN assembly and function (**Fig. 7**). Nonetheless, we mitigated our statement and focused the discussion on the findings of the paper. The sentence to end the manuscript now reads as: Clearly, further manipulation of metabolic and signaling pathways that link proteasome LLPS and amino acid sensing is expected to provide additional insights into the role of SIPAN in physiology and human disease.

REVIEWERS' COMMENTS

Reviewer #1 (Remarks to the Author):

In the revised manuscript, the authors have added a great deal of experimental data, and the manuscript has been very much strengthened regarding the characterization of proteasome condensates as droplets, the role of RAD23B and PSME3, and the mechanism of cell death induction. The authors have addressed almost all of my concerns. I enjoyed reading the manuscript again and support its publication in Nature Communication.

Reviewer #2 (Remarks to the Author):

The authors have completed an additional round of substantial experiments to tease out the underlying mechanisms of SIPAN formation. While the trigger is not fully identified, the authors have observed the importance of continuous ubiquitination to maintain SIPAN foci, further clarified the role of RAD23B in SIPAN formation, and compared/contrasted mammalian SIPAN formation with yeast PSG (proteasome storage granule) formation. The authors propose that SIPAN is an upstream signal of nutrient starvation-induced cell death. This is an extensive body of work with many interesting questions for the protein quality control field, particularly in the role of proteasome condensates/assemblies in nutrient stress response. This work complements recent work by Yasuda et al. (Nature 2020) that also described the role of RAD23B in the formation of proteasome foci, highlighting a new role of ubiquitin-binding shuttle proteins in assembling stress-induced condensates. There are a few comments and textual edits, but the work is ready for publication in Nature Communications.

In the abstract and in the beginning of the discussion, I would suggest revising the text that says that the “proteasome itself undergoes LLPS”. This is misleading, as there are a number of components (RAD23B, potentially ubiquitinated substrates, etc.) that together recruit proteasomes into nuclear condensates, and this work doesn’t explicitly state that the proteasome phase separates on its own. The authors provide evidence that RAD23B phase separates on its own. The intriguing question of how K48 polyUb chains affect these RAD23B droplets is addressed in the rebuttal letter, but requires further examination as the authors state.

I may have missed this comment but it is interesting to see how nutrient-deprived cells appear to move towards each other in Supplementary Video 1 and Video 11. This may be coincidence or related to SIPAN functionality?

On pg. 17, I would disagree with that the statement that UBL, UBA1 or UBA2 almost completely abrogated SIPAN formation based on Fig 8k,l. This figure is referring to purified RAD23B constructs, not RAD23B in cells where SIPAN formation is being tested. Please reword and recheck this paragraph for accuracy. Additionally, please note that it is difficult to assess what the RAD23B deletion constructs do when only a single experimental condition is tested, so I caution overinterpretation in that regard. Full phase diagrams would need to be elucidated which is outside the scope of the current manuscript.

In the discussion on pg. 24, the UBQLN2 statement is misleading – I believe the work shows that UBQLN2 uses its multiple domains to drive UBQLN2 LLPS, which may contribute towards its association with stress granules or other stress-induced condensates.

Pg. 9 line 211 – should be PSMD14?

Pg. 14 line 322 – “over time”

Pg. 16 line 358 – should be Fig. 7n, correct?

Pg. 20 line 447 – which has ‘no’ impact

Reviewer #3 (Remarks to the Author):

Some of my original major concerns remain unaddressed. They need to be addressed in full before publication. The authors present interesting findings but some faulty interpretations cause serious damage.

Comment 6, Response R3-7: I don't find evidence supporting the idea that the proteasome phase separate. The author show evidence of foci formation, the phase separation argument is circumstantial and not demonstrated. The claims need to be revised, and so does the title.

Comment 12, Response R3-13.

The authors have not addressed the issue. They show that proteasome inhibition cause accumulation of short-lived proteins. This is evidence that there is some active proteasome in cells. The authors claim that this demonstrate that SIPAN contains active proteasome. The claim is not substantiated and needs to be removed. Citing another paper as evidence isn't right. If the authors want to claim that the proteasome is active in SIPAN, they need evidence. Without evidence, the claim needs to be removed.

Comment 13, Response 3-14.

Here as well, there is no evidence support the role of SIPAN in cell death. The reasoning is a syllogism and the conclusion invalid. Of course absence of amino acids cause death and blocking translation prevents this. In the process, SIPAN are induced. Yet, there is no direct evidence supporting the idea that SIPAN is involved. There is no way to demonstrate that either at this point because the SIPAN components have many other functions. The claim that SIPAN regulates cell death needs to be deleted.

Comment 15, Response 3-16. Same as above. The proteins the authors propose to knock down have major functions unrelated to SIPAN. Inactivating them will not address the importance of SIPAN. This part needs to be revised.

Rebuttal letter

Liquid phase separation of the mammalian nuclear proteasome links amino acid supply to apoptosis by

Uriarte et al.

Reviewer #1 (Remarks to the Author): In the revised manuscript, the authors have added a great deal of experimental data, and the manuscript has been very much strengthened regarding the characterization of proteasome condensates as droplets, the role of RAD23B and PSME3, and the mechanism of cell death induction. The authors have addressed almost all of my concerns. I enjoyed reading the manuscript again and support its publication in Nature Communication.

Response: We thank the reviewer for the enthusiasm and comments.

Reviewer #2 (Remarks to the Author): The authors have completed an additional round of substantial experiments to tease out the underlying mechanisms of SIPAN formation. While the trigger is not fully identified, the authors have observed the importance of continuous ubiquitination to maintain SIPAN foci, further clarified the role of RAD23B in SIPAN formation, and compared/contrasted mammalian SIPAN formation with yeast PSG (proteasome storage granule) formation. The authors propose that SIPAN is an upstream signal of nutrient starvation-induced cell death. This is an extensive body of work with many interesting questions for the protein quality control field, particularly in the role of proteasome condensates/assemblies in nutrient stress response. This work complements recent work by Yasuda et al. (Nature 2020) that also described the role of RAD23B in the formation of proteasome foci, highlighting a new role of ubiquitin-binding shuttle proteins in assembling stress-induced condensates. There are a few comments and textual edits, but the work is ready for publication in Nature Communications.

In the abstract and in the beginning of the discussion, I would suggest revising the text that says that the “proteasome itself undergoes LLPS”. This is misleading, as there are a number of components (RAD23B, potentially ubiquitinated substrates, etc.) that together recruit proteasomes into nuclear condensates, and this work doesn’t explicitly state that the proteasome phase separates on its own. The authors provide evidence that RAD23B phase separates on its own. The intriguing question of how K48 polyUb chains affect these RAD23B droplets is addressed in the rebuttal letter, but requires further examination as the authors state.

Response: We tried to think about another title and did not come up with a better one. We believe our current title is not incorrect, and that we could say that the proteasome undergo phase separation. Proteasome phase separation has been demonstrated in our paper in vivo. We agree that we did not demonstrate that the proteasome itself undergoes phase separation in vitro, but we never claimed that the proteasome is sufficient for liquid phase separation. In all cases, under conditions of nutrient deprivation, we observe a distinct phase separation of the proteasome in the cells and we extensively characterized these condensates. Our title is arguably comparable to the recent paper by Yasuda et al., Nature volume 578, pages296–300 (2020), Stress- and ubiquitylation-dependent phase separation of the proteasome. Thus, we would like to keep the original title.

I may have missed this comment but it is interesting to see how nutrient-deprived cells appear to move towards each other in Supplementary Video 1 and Video 11. This may be coincidence or related to SIPAN functionality?

Response: Thank you to the reviewer for the careful observation and suggestion. We do not know at this point whether this is a coincidence or whether this might reflect a cellular response associated with SIPAN. We will be carefully looking at this in our next studies.

On pg. 17, I would disagree with that the statement that UBL, UBA1 or UBA2 almost completely abrogated SIPAN formation based on Fig 8k,l. This figure is referring to purified RAD23B constructs, not RAD23B in cells where SIPAN formation is being tested. Please reword and recheck this paragraph for accuracy.

Response: Our apologies for this mistake. We agree with the reviewer. The phase separation in Fig 8k,l. is with purified RAD23B. The sentence is now stated as: *Finally, we found that deletion of several RAD23B domains, notably UBL, UBA1 or UBA2 reduced RAD23B ability to undergo phase separation.*

Additionally, please note that it is difficult to assess what the RAD23B deletion constructs do when only a single experimental condition is tested, so I caution overinterpretation in that regard. Full phase diagrams would need to be elucidated which is outside the scope of the current manuscript.

Response: We agree with the reviewer and indeed additional studies are needed to elucidate how RAD23B undergo phase separation in vitro and in cells.

In the discussion on pg. 24, the UBQLN2 statement is misleading – I believe the work shows that UBQLN2 uses its multiple domains to drive UBQLN2 LLPS, which may contribute towards its association with stress granules or other stress-induced condensates.

Response: We agree and now restated as : *It is noteworthy that another proteasome shuttling factor, UBQLN2, with similarities to RAD23B, also uses its multiple domains to undergo LLPS and association with stress granules in the cytoplasm*

Pg. 9 line 211 – should be PSMD14?

Pg. 14 line 322 – “over time”

Pg. 16 line 358 – should be Fig. 7n, correct?

Pg. 20 line 447 – which has ‘no’ impact

Response: Thank you. We have made these corrections.

Reviewer #3 (Remarks to Author): Some of my original major concerns remain unaddressed. They need to be addressed in full before publication. The authors present interesting findings but some faulty interpretations cause serious damage.

Comment 6, Response R3-7: I don't find evidence supporting the idea that the proteasome phase

separate. The author show evidence of foci formation, the phase separation argument is circumstantial and not demonstrated. The claims need to be revised, and so does the title.

Response: We respectfully disagree with the reviewer and we hope that our interpretation and conclusion will be considered. We believe that we have provided sufficient evidence to claim that the proteasome undergoes a liquid-liquid phase separation. Unfortunately, the reviewer does not provide specific comments as to why we cannot describe our proteasome foci as a liquid-liquid phase separation (LLPS) and he/she does not provide specific details/interpretations that we can agree or disagree with. Here we outline the major findings on which we based our conclusion. **(1)** Many components of the proteasome are found in distinct nuclear foci upon nutrient deprivation, and depletion of components of 19S and 20S subunits abolish proteasome foci formation (Figure 2a-e; Supplementary Figure 2a,c). This ability of the proteasome to form small foci is reminiscent of other subnuclear structures that result from LLPS including PML bodies (Corpet A, et al. *Nucleic Acids Res.* 2020;48(21):11890-11912). We note that our electronic microscopy studies indicated that proteasome foci are membrane less, an important criterion for foci to be candidates for LLPS (Figure 3c), **(2)** These proteasome foci are transient and dissipate quickly following amino acid replenishment (Figure 5k-n; Supplementary Figure 5a-c). This indicates the dynamic nature of proteasome foci, which is reminiscent of other cellular condensates that originate as a result of LLPS, such as stress granules (Protter DSW and Parker R. *Trends Cell Biol.* 2016;26(9):668-679). **(3)** The proteasome foci detected by staining for endogenous proteins including PSMB4, PSMB5, PSMB6, PSMB7, PSMD2, PSMD4, PSMD7, PSMD11, PSMD14 as well as exogenous proteins (PSMB4-GFP, PSMB5-GFP, PSMD12-GFP, PSMD14-GFP) show a spherical morphology (Figure 1c,f; Figure 2a; Supplementary figure 2a,b). This characteristic is one of the criteria that is also necessary to qualify protein foci as LLPS candidates (Alberti Set al. *Cell.* 2019;176(3):419-434). **(4)** Live-cell imaging indicates that proteasome foci undergo fusion events, another criteria for considering proteasome foci formation as a result of LLPS (Figure 5e) (Alberti Set al. *Cell.* 2019;176(3):419-434). **(5)** Importantly, FRAP experiments using an obligate component of the proteasome (PSMB4), indicated the highly dynamic exchange between SIPAN and the surrounding cellular environment in the nucleus (Figure 6e). This is another important criteria that supports our conclusion (Alberti Set al. *Cell.* 2019;176(3):419-434). **(6)** Perturbations of the intracellular environment is known to influence the stability of phase separations, and we show that proteasome foci are highly sensitive to the physico-chemical conditions of the cell environment (Figure 6a,c,d; Supplementary Figure 6) (Alberti Set al. *Cell.* 2019;176(3):419-434). **(7)** Further, we used the 1,6-hexanediol, a well-known inhibitor of LLPS (Alberti Set al. *Cell.* 2019;176(3):419-434), and we show that the proteasome foci dissipate rapidly after treatment of cells (Figure 6b). **(8)** We

show that RAD23B, a multivalent protein, is required for proteasome foci formation and this factor undergoes LLPS in vitro (Figure 7l,m; Figure 8; Supplementary Figure 8). RAD23B is a key protein in phase separation, likely acting as the scaffold to promote the recruitment of additional proteins or complexes (the clients, e.g., the proteasome), thus ensuring functionality to the phase separation. The literature highlights examples of scaffold/client partnership which adds a new level of complexity in phase separation (Banani, et al. Cell. 2016;166(3):651-663; Kilic S, et al. EMBO J. 2019;38(16):e101379; Ditlev JA et al. J Mol Biol. 2018;430(23):4666-4684; Bracha D, et al. Nat Biotechnol. 2019 Dec;37(12):1435-1445). Taken our data altogether and taking into consideration the current criteria for LLPS (Alberti Set al. Cell. 2019;176(3):419-434), our data indicate clearly that the proteasome undergoes a phase separation. Again, we do not currently believe that the proteasome by itself phase separate and our study does not conclude on whether the proteasome is sufficient for its own LLPS. Finally, supporting our results, another study recently published showed the formation of proteasome foci as a result of LLPS in response to osmotic stress (Yasuda et al. Nature volume 578, pages296–300, 2020).

Comment 12, Response R3-13.

The authors have not addressed the issue. They show that proteasome inhibition cause accumulation of short-lived proteins. This is evidence that there is some active proteasome in cells. The authors claim that this demonstrate that SIPAN contains active proteasome. The claim is not substantiated and needs to be removed. Citing another paper as evidence isn't right. If the authors want to claim that the proteasome is active in SIPAN, they need evidence. Without evidence, the claim needs to be removed.

Response: We followed short-lived nuclear proteins to show that these proteins still accumulate after proteasome inhibition with MG132 (and during conditions of nutrient deprivation). Importantly, our MG132 treatment was done for only one hour when the SIPAN are already formed (about 90-100 % of cells with proteasome foci). In addition, we used a proteasome activity probe (Me4BodipyFL-Ahx3Leu3VS), which also provided another evidence that the proteasome present in the foci is still capable of proteolysis (Figure 10b). This probe was used by Yasuda et al. (Nature volume 578, pages296–300, 2020) to also show that the proteasome is active in the foci induced by osmotic stress. Nonetheless, we now mitigated our statement as follows:

In the Results section:

Next, we used a Me4BodipyFL-Ahx3Leu3VS (Proteasome Activity Probe) 43, which accumulates in SIPAN after hydrolysis, and this also ~~revealed~~ suggested that the proteasome is active in SIPAN (Fig. 10b).

In the Discussion section:

SIPAN assemble in nuclear regions of low chromatin density and the proteasome ~~is active~~ within these structures ~~is labelled with a fluorescent activity probe~~, suggesting that SIPAN are subnuclear sites of active proteolysis.

Comment 13, Response 3-14.

Here as well, there is no evidence support the role of SIPAN in cell death. The reasoning is a syllogism and the conclusion invalid. Of course absence of amino acids cause death and blocking translation prevents this. In the process, SIPAN are induced. Yet, there is no direct evidence supporting the idea that SIPAN is involved. There is no way to demonstrate that either at this point because the SIPAN components have many other functions. The claim that SIPAN regulates cell death needs to be deleted.

Response: We did not identify factors that directly link proteasome foci to cell death. In our paper, we do not claim a direct physical link between proteasome foci and cell death, but we provide several pieces of evidence to support our conclusion that SIPANs promote cell death. **(1)** Inhibition of protein synthesis by cycloheximide prevents SIPAN formation by preventing the consumption of the intracellular amino acid pool. This also promoted cell survival (Figure 9d-f; Supplementary Figure 9e-f). We note that the cycloheximide treatment is performed under nutrient starvation, a stress condition known to inhibit general protein synthesis (Morita M. et al. Cell Cycle. 2015;14(4):473-480; Ma XM and Blenis J. Nat Rev Mol Cell Biol. 2009 May;10(5):307-18). Thus, cycloheximide-mediated translation inhibition, during starvation, is likely to act on a limited subset of stress-associated proteins. **(2)** Preventing recycling of amino acids by treating cells with autophagy inhibitors, chloroquine or 3-methyladenine, promote proteasome foci formation and decrease cell survival (Figure 9d-f; Supplementary Figure 9e,f). **(3)** Non-essential amino acids (NEAA) do not induce proteasome foci formation and prevent cell death, while essential amino acids (EAA) do not dissipate proteasome foci and do not rescue from cell death (Figure 9g,h; Supplementary Figure 9c,d,g,h). **(4)** The proteasome shuttle factors RAD23B and RAD23A are both present in the proteasome foci. Depletion of RAD23B, but not RAD23A, inhibits SIPAN formation (Figure 7l-n; Supplementary Figure 7g,h). Moreover, Depletion of RAD23B, but not RAD23A, promotes cell survival (Figure 10c,d,e; Supplementary Figure 10a,b,d,f). **(5)** Furthermore, depletion of PSME3, another component of SIPAN, also promotes cell survival (Figure 2d; Figure 10c,d,e; Supplementary Figure

10a,c,d). **(6)** We also showed that p53/NOXA is induced during nutrient deprivation, and depletion of NOXA promotes cell survival. In addition, the depletion of RAD23B or PSME3 prevents the induction of NOXA (Figure 10f-j, Supplementary Figure 10d,i-k). **(7)** We also inactivated the p53 pathway, which is important for cell death in response to nutrient stress, to induce oncogenic transformation of primary fibroblasts. We found that transformed cells have a highly reduced ability to form SIPAN and are resistant to cell death following nutrient deprivation (Figure 10l,m,n, Supplementary Figure 10l). Overall, while we agree with the reviewer that more investigations are required to identify the direct mediators that link SIPAN and cell death, we believe that we provided sufficient evidence to support the role of SIPAN in cell death.

Comment 15, Response 3-16. Same as above. The proteins the authors propose to knock down have major functions unrelated to SIPAN. Inactivating them will not address the importance of SIPAN. This part needs to be revised.

Response: As indicated above, RAD23B and PSME3 form foci with the proteasome in the nucleus upon nutrient deprivation. In addition, the depletion of RAD23B, PSME3 or NOXA promotes cell survival in response to nutrient deprivation (Figure 10; Supplementary figure 10). We know that RAD23B and PSME3 could have different functions under normal conditions. However, we emphasize that we are investigating RAD23B and PSME3 functions under conditions of nutrient deprivation. Our localization data on RAD23B and PSME3, indicating that these factors form foci under conditions of nutrient deprivation, are compelling. We show that RAD23B is necessary for proteasome foci formation under conditions of nutrient deprivation (Figure 7). Our functional assays indicate that depletion of either RAD23B or PSME3 prevent NOXA upregulation and cell death in response to nutrient deprivation (Figure 10; Supplementary Figure 10). We believe that our data altogether are well integrated in a coherent model.

+++++

9. Please consider the revised title “Condensation of the mammalian nuclear proteasome links amino acid supply to apoptosis”.

-We thank the editor for the suggestion. We are somewhat concerned that the word condensation might not be well comprehended. We believe that our original title reflects our findings (as outlined above). In fact, we presented and discussed our data in proteostasis meetings and several experts very

much appreciated that the proteasome undergoes LLPS. Nonetheless, we now propose a new title as follows: **“Starvation-induced proteasome assemblies in the nucleus link amino acid supply to apoptosis”**. We hope that the editor will find this new title suitable.

-In the abstract, we removed Proteasome LLPS as follows:

(iv) RAD23B proteasome shuttling factor is required for proteasome ~~LLPS~~ SIPAN formation

10. Please note that the Supplementary Information does not contain any Supplementary Tables and please remove reference to the uncropped blots, which are in the Source Data. Please provide these Tables in the Supplementary Information file if these are intended to be included with the published manuscript.

The tables are now present in the Supplementary Information file and we removed the references to the uncropped blots and FACS.

We hope that our manuscript is now suitable for publication.

Sincere regards,

El Bachir Affar